

# A prevalence of *Arthropterygius* (Ichthyosauria: Ophthalmosauridae) in the Late Jurassic—earliest Cretaceous of the Boreal Realm

Nikolay G. Zverkov[1,2,3] and Natalya E. Prilepskaya[4]

[1] Geological Faculty, Lomonosov Moscow State University, Moscow, Russia
[2] Geological Institute of the Russian Academy of Sciences, Moscow, Russia
[3] Borissiak Paleontological Institute of the Russian Academy of Sciences, Moscow, Russia
[4] Severtsov Institute of Ecology and Evolution, Russian Academy of Sciences, Moscow, Russia

Corresponding author
Nikolay G. Zverkov,
zverkovnik@mail.ru

## ABSTRACT

The ichthyosaur genus *Arthropterygius Maxwell, 2010* is considered as rare and poorly known. However, considering the existing uncertainty regarding its position in respect to ophthalmosaurid subfamilies in recent phylogenies, it is among the key taxa for understanding the evolution of derived Late Jurassic and Early Cretaceous ichthyosaurs. Recently excavated unique material from the Berriassian of Franz Josef Land (Russian Extreme North) and examination of historical collections in Russian museums provided numerous specimens referable to *Arthropterygius*. The new data combined with personal examination of ichthyosaurs *Palvennia*, *Janusaurus*, and *Keilhauia* from Svalbard give us reason to refer all these taxa to *Arthropterygius*. Therefore, we recognize four species within the genus: *Arthropterigius chrisorum* (*Russell, 1994*), *A. volgensis* (*Kasansky, 1903*) comb. nov., *A. hoybergeti* (*Druckenmiller et al., 2012*) comb. nov., and *A. lundi* (*Roberts et al., 2014*) comb. nov. Three of the species are found both in the Arctic and in the European Russia. This allows the suggestion that *Arthropterygius* was common and widespread in the Boreal Realm during the Late Jurassic and earliest Cretaceous. The results of our multivariate analysis of ophthalmosaurid humeral morphology indicate that at least some ophthalmosaurid genera and species, including *Arthropterygius*, could be easily recognized based solely on humeral morphology. Our phylogenetic analyses place the clade of *Arthropterygius* close to the base of Ophthalmosauria as a sister group either to ophthalmosaurines or to platypterygiines. Although its position is still uncertain, this is the best supported clade of ophthalmosaurids (Bremer support value of 5, Bootstrap and Jackknife values exceeding 80) that further augments our taxonomic decision.

## INTRODUCTION

Ichthyosaurs were common components of marine herpetofauna in the Late Jurassic. We know this due to several Late Jurassic formations that yielded significant ichthyosaur

materials. These are primarily Kimmeridge Clay Formation of England and France (*Hulke, 1871*; *Mansell-Pleydell, 1890*; *Sauvage, 1911*; *Delair, 1960*, *1986*; *McGowan, 1976*, *1997*; *Grange et al., 1996*; *Etches & Clarke, 1999*; *Moon & Kirton, 2016*), the Solnhofen Formation of Germany (*Wagner, 1852*, *1853*; *Meyer, 1864*; *Bauer, 1898*; *Bardet & Fernández, 2000*), the Vaca Muerta Formation of Argentina (*Fernández, 1997*, *2000*, *2007a*, *2007b*; *Gasparini, Spalletti & De La Fuente, 1997*, *Gasparini et al., 2015*), the Agardhfjellet Formation of Svalbard, Norway (*Angst et al., 2010*; *Druckenmiller et al., 2012*; *Roberts et al., 2014*; *Delsett et al., 2016*, *2017*) and a number of formations of the Volgian (Tithonian) age in European Russia (*Kabanov, 1959*; *Efimov, 1998*, *1999a*, *1999b*; *Arkhangelsky, 1997*, *1998*, *2000*, *2001a*, *2001b*; *Zverkov, Arkhangelsky & Stenshin, 2015*; *Zverkov et al., 2015*; *Zverkov & Efimov, 2019*). Still our knowledge of the Late Jurassic ichthyosaurs is non-uniform: some taxa are quite well known owing to relatively complete and well-preserved specimens (*Ophthalmosaurus Seeley, 1874*; *Grendelius McGowan, 1976*; *Caypullisaurus Fernández, 1997*; *Aegirosaurus Bardet & Fernández, 2000*; *Undorosaurus Efimov, 1999b*), whereas others are poorly known from only a small number of largely incomplete and/or poorly preserved specimens (e.g., *Nannopterygius Huene, 1922*, *Brachypterygius Huene, 1922*, and *Arthropterygius Maxwell, 2010*). Being in the list of these puzzling ichthyosaurs, *Arthropterygius* is known by only fragmentary remains: its type and the only species is represented only by the holotype, an incomplete skeleton from Arctic Canada (*Maxwell, 2010*). Two more fragmentary specimens were subsequently referred to as *Arthropterygius*: one from Argentina (*Fernández & Maxwell, 2012*) and another from the Russian North (*Zverkov et al., 2015*), however, both of them were described in open nomenclature. Thereby the genus remains poorly known.

In recent years, the Slottsmøya Member of the Agardhfjellet Formation of Svalbard has yielded numerous marine reptile specimens including four monotypic ichthyosaur genera, for most of which only one specimen is known (*Druckenmiller et al., 2012*; *Roberts et al., 2014*; *Delsett et al., 2017*). Recently, it has been proposed and argued that one of the ichthyosaur genera from Svalbard, "*Cryopterygius*," is a junior subjective synonym of *Undorosaurus Efimov, 1999b* (*Zverkov & Efimov, 2019*). The other three genera from Svalbard are discussed in the present contribution and are all considered as junior subjective synonyms of *Arthropterygius*. Study of newly discovered materials from Franz-Josef Land (Russian Extreme North) combined with examination of ichthyosaurs in historical collections of several museums in Russia and in the Natural History Museum at the University of Oslo substantially expand the knowledge of ichthyosaurs of the *Arthropterygius* clade.

One of the most peculiar skeletal elements of *Arthropterygius* is its humerus that has a marked constriction between the radial and ulnar facets (ventral skew). This trait in combination with other features (distally faced radial facet and presence of a well-developed facet for the anterior accessory element) helps for easy recognition of humeri belonging to *Arthropterygius* among those of other ophthalmosaurids. However, the marked constriction between the ulnar and radial facet is not unique for *Arthropterygius* and is characteristic of a very poorly known Late Jurassic "*Macropterygius*" (*Moon & Kirton, 2018*). At the same time, "*Macropterygius*" do not have a facet for

an anterior accessory epipodial element, and, furthermore, it has slightly anteriorly deflected radial facet, which differs from distally faced radial facets in *Arthropterygius*, *Ophthalmosaurus* and some other opthalmosaurids. The variation of the humerus in ophthalmosaurids has not been previously assessed with implementation of morphometric techniques, although this skeletal element has a complex morphology that gives a number of phylogenetically informative characters (see characters in e.g., *Fischer et al., 2012*; *Moon, 2019*; *Zverkov & Efimov, 2019*). In order to highlight this, we gather a new dataset and run the principal component analysis (PCA) of ophthalmosaurid humeral morphology.

This research continues an ongoing project of taxonomic and phylogenetic revision of the Late Jurassic ichthyosaurs of the Boreal Realm. Here, we focus on ichthyosaurs of *Arthropterygius* clade (*Zverkov & Efimov, 2019*), their taxonomy, ontogenetic, intra- and interspecific variation along with their phylogenetic relations to other ophthalmosaurids.

## MATERIALS

During the fieldwork of A.P. Karpinsky Russian Geological Research Institute (VSEGEI) in Franz Josef Land, several ichthyosaur specimens were collected from the black shales of the Hofer Formation (Upper Jurassic to lowermost Cretaceous; *Kosteva, 2005*; *Rogov et al., 2016*). The first specimen (CCMGE 1–2/13328) represented by a medial fragment of the left scapula and proximal fragment of the right humerus of a big ichthyosaur was found by S. Yudin and P. Rekant in a scree of a slope formed by Kimmeridgian and Volgian sediments at Wilczek Land (Fig. 1A). NGZ had excavated two more relatively complete specimens at Berghaus Island (Fig. 1A). Both of them are referable to *Arthropterygius chrisorum* (see 'Results'). When found, skulls and some portions of postcranial skeleton of both CCMGE 3–16/13328 and CCMGE 17–44/13328 were already exposed and weathered, thereby a number of cranial elements are too fragmental for description and even more parts are missing. The specimens were collected and prepared by NGZ, and scanned by NEP using Artec Spider 3D scanner.

Furthermore, studying the collections in museums of Russia, we found out several specimens referable to *Arthropterygius*. Four of them are from the Middle Volgian of the Volga Region (Ulyanovsk and Samara regions), the fifth, originating from the Russian North, was described in previous work (*Zverkov et al., 2015*). Two of the specimens, deposited in Vernadsky State Geological Museum (SGM, Moscow), were excavated at the beginning of the last century. One (SGM 1573) was discovered by A.P. Pavlov and subsequently described by *Bogolubov (1910)* as *Ophthalmosaurus* cf. *thyreospondylus*, another specimen (SGM 1731-01–15), found in 1937 by an unknown collector, remained hitherto undescribed. A partial skeleton of a juvenile (KSU 982/P-213), described by *Kasansky (1903)* as a new species, *Ichthyosaurus volgensis Kasansky, 1903*, is deposited in the Museum of Geology and Mineralogy of Kazan State University (KSU). Since the original descriptions of SGM 1573 and KSU 982/P-213 a number of skeletal elements were lost in both specimens. The vertebral column (except for several small caudal centra) is now lost in KSU 982/P-213. Initially, the specimen excavated by A.P. Pavlov (SGM 1573) included 13 vertebrae, several neural arches, rib fragments, left coracoid, complete right

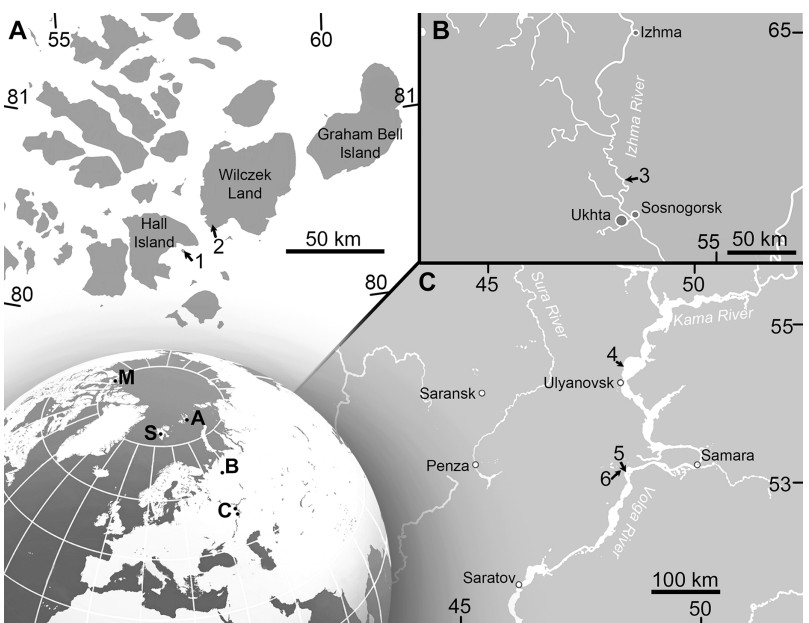

**Figure 1 Maps showing the discovery sites of *Arthropterygius* in Russia and globally.** (A) Map of Franz-Joseph Land with localities on Berghaus Island (1), and on Wilczek Land (2). (B) Map of a part of Timan-Pechora Basin, with the locality near Porozhsk Village (3). (C) Map of the middle Volga Region with the locatities near Gorodischi Village (4), Kashpir Village (5), and Novaya Racheyka Village (6). (M) The locality on Melville Island, Arctic Canada. (S) Localities on Svalbard, Norway.

scapula, interclavicle, left humerus, anterior accessory epipodial, and several autopodial elements (*Bogolubov, 1910*). Currently, 10 vertebrae, interclavicle, broken distal portion of the scapula and left humerus are deposited in SGM, the remaining elements either decayed or misplaced (I.A. Starodubtseva, 2016, personal communication). However, we suggest that the available remains are sufficient for attributing SGM 1573 to *A. chrisorum*.

Three more specimens referable to *Arthropterygius* were found in recent decades at the right bank of the Volga River near Gorodischi Village, Ulyanovsk Region. An incomplete postcranial skeleton (YKM 63548) was found by V. M. Efimov and donated to YKM; an isolated humerus UPM 2442 was found by I.M. Stenshin (UPM); an isolated basisphenoid was obtained by NGZ from an anonymous fossil dealer and donated to SGM, where it deposited now under the number SGM 1743-2.

The specimens referable to *Arthropterygius* and examined as part of the present study are summarized in Table 1.

For the PCA, NGZ has additionally collected data on ophthalmosaurid humeri stored in the following institutions: CAMSM, NHMUK, PMO, SGM, UPM, YKM (see Tables S6 and S7).

## GEOLOGICAL SETTING

During the latest Jurassic and earliest Cretaceous, a high faunal provincialism is observed in many basins of the Northern hemisphere (*Rogov, 2012*; *Rogov & Zakharov, 2009*). These basins constitute the so-called Pan-Boreal Superrealm, where Volgian and Ryazanian

**Table 1** Specimens referable to *Arthropterygius* and examined as part of the present study.

| Specimen no | Material | Locality | Formation/bed and ammonite zone | Taxonomic identification in previous works | Taxonomic identification in this work | Reference to Figure |
|---|---|---|---|---|---|---|
| CMGE 1–2/13328 | Medial fragment of the left scapula and proximal fragment of the right humerus | Cape Hansa, Wilczek Land | Hofer Formation, middle part (collected ex situ); Kimmeridgian to Volgian | – | Ophthalmosauridae gen. indet. cf. *Arthropterygius* | S3 |
| CMGE 3–16/13328 | Incomplete skeleton of a juvenile individual: left quadrate, partial basisphenoid, incomplete supratemporals, fragmentary parietal, and several other indeterminate cranial fragments, incomplete vertebral column (69 vertebrae from anterior dorsal to tailfin centra); rib fragments, right forefin, right scapula, coracoids | Berghaus Island, Franz Josef Land | Upper part of the Hofer Fm., early Berriassian | – | *Arthropterygius chrisorum* | 3E–3I; 3N–3V; 6L–6AA; 7F–7R; 18A–18D; 23B, 23F, 23L, 23N, 23S, 23W, 23CC, 23DD; S2 |
| CMGE 17–44/13328 | Incomplete skeleton of a young adult individual: right nasal, prefrontals, right postfrontal, fragmentary parietal, basisphenoid, left quadrate; fragments of palate bones and other indeterminate cranial remains; mandible, including articulated left surangular, angular, splenial and prearticular, isolated presacral and anterior caudal centra (31 fragment), multiple rib fragments, fragments of pectoral girdle (coracoids, scapulae, interclavicle and clavicle), incomplete right forefin, proximal part of the left humerus, left radius, partial ischiopubis, left femur | Berghaus Island, Franz Josef Land | Upper part of the Hofer Fm., early Berriassian | – | *Arthropterygius chrisorum* | 2A–2J; 3A–3D, 3J–3M; 5; 7A–7E, 7X–7BB; 8; 18E–18H; 23C, 23G, 23K, 23O, 23T, 23AA; S7E |

(Continued)

| Specimen no | Material | Locality | Formation/bed and ammonite zone | Taxonomic identification in previous works | Taxonomic identification in this work | Reference to Figure |
|---|---|---|---|---|---|---|
| SGM 1573 | 10 vertebrae, interclavicle, broken distal portion of the scapula, left humerus | Right bank of the Volga River between Ulyanovsk and Gorodischi, Ulyanovsk Region | Promza Fm., *Dorsoplanites panderi* Ammonite Biozone; early middle Volgian | *Ophthalmosaurus* cf. *thyreospondylus* Bogolubov (1910) | *Arthropterygius chrisorum* | 6A–6K; 7S–7W, 7CC–7EE; 23I, 23EE, 23FF |
| SGM 1731-01–15 | 10 anterior presacral vertebrae with articulated neural arches; scapulae; left coracoid; left humerus with articulated epipodial and proximal autopodial elements | Bank of the Volga River near Kashpir, Samara Region | Promza Fm., *Dorsoplanites panderi* Ammonite Biozone; early middle Volgian | – | *Arthropterygius lundi* | 15A–15C, 15H–15M; S8 |
| KSU 982/P-213 (holotype) | Incomplete skeleton of a juvenile represented by cranial remains (including basisphenoid, opisthotics, quadrates, parietals, right supratemporal, and articular), three posterior caudal and tailfin vertebrae; neural arches and rib fragments, coracoids; fragments of the interclavicle, scapula, and clavicles, distal portion of the femur | Berezoviy Dol Ravine near Novaya Racheika Village, Syzran District, Samara Region | Promza Fm., *Dorsoplanites panderi* Ammonite Biozone; early middle Volgian | *Ichthyosaurus volgensis* Kasansky, 1903 | *Arthropterygius volgensis* (Kasansky, 1903) comb. nov. | 16, 17, 18Q–18T |
| SGM 1743-2 | Isolated basisphenoid | Right bank of the Volga River near Gorodischi Village, Ulyanovsk Region | Promza Fm., *Dorsoplanites panderi* Zone; early middle Volgian | – | *Arthropterygius* cf. *chrisorum* | 18I–18L |
| SGM 1502 | Basisphenoid, fragmental rostrum, vertebra, scapula, humerus (see Zverkov et al., 2015) | Right bank of the Volga River near Gorodischi Village, Ulyanovsk Region | Paromes Fm., *Dorsoplanites panderi* Zone; early middle Volgian | *Arthropterygius* sp. (Zverkov et al., 2015) | *Arthropterygius lundi* | 11B, 11C; 18U–18X; S7G |

| Specimen no | Material | Locality | Formation/bed and ammonite zone | Taxonomic identification in previous works | Taxonomic identification in this work | Reference to Figure |
|---|---|---|---|---|---|---|
| PMO 222.669 | A partially articulated and almost complete anterior half of the skeleton of a moderately large ichthyosaur (for details see *Delsett et al. (2018)*) | Janusfjellet, Svalbard, Norway | Slottsmøya Member, Agardhfjellet Formation, Janusfjellet Subgroup, "15.5 m above the echinoderm marker bed" ?*Dorsoplanites maximus* Ammonite Biozone; middle Volgian | *Palvennia hoybergeti Delsett et al. (2018)* | *Arthropterygius chrisorum* | 9; 10; 11A; 12A–12F; S4, S5, S7A |
| SVB 1451 (holotype) | A nearly complete skull, atlas/axis complex and fragmentary vertebra, right clavicle, fragments of left and right scapulae, proximal and distal portions of a humerus, limb elements and several disarticulated dorsal ribs | Janusfjellet, Svalbard, Norway | Slottsmøya Member of the Agardhfjellet Formation; "15.2 metres below the *Dorsoplanites* bed," most likely *Dorsoplanites ilovaiskii—D. maximus* ammonite biozones; middle Volgian | *Palvennia hoybergeti* (*Druckenmiller et al., 2012*) | *Arthropterygius hoybergeti* | 2L, 2M; 4; 23D, 23H, 23J, 23P, 23U, 23X; S7D |
| YKM 63548 | A slab containing a series of 19 presacral vertebrae with articulated neural arches and ribs, right humerus, a cast of the left humerus with associated radius, ulna and intermedium (original forelimb was lost because of pyrite decay) | Right bank of the Volga River near Gorodischi Village, Ulyanovsk Region | Promza Fm., *Dorsoplanites panderi* Ammonite Biozone (early middle Volgian) | – | *Arthropterygius* cf. *hoybergeti* | 12L–12Q; S6, S7B |
| UPM 2442 | Isolated humerus | Volga River near Gorodischi Village, Ulyanovsk Region | Promza Fm., *Dorsoplanites panderi* Ammonite Biozone (early middle Volgian) | – | *Arthropterygius* cf. *hoybergeti* | 12G–12K; S7C |

(Continued)

| Specimen no | Material | Locality | Formation/bed and ammonite zone | Taxonomic identification in previous works | Taxonomic identification in this work | Reference to Figure |
|---|---|---|---|---|---|---|
| PMO 222.654 (holotype) | Partial skeleton of a moderately large individual (for details see *Roberts et al., 2014*) | Janusfjellet, Svalbard, Norway | Slottsmøya Member, Agardhfjellet Fm., "31 m below the Dorsoplanites Bed, 4 m below the echinoderm bed," ? *Pavlovia rugosa* to *Dorsoplanites ilovaiskii* ammonite biozones | *Janusaurus lundi* (*Roberts et al., 2014*) | *Arthropterygius lundi* | 13; 14; 15D–15G, 15N–15Q; S7F |
| PMO 222.655 (holotype) | Incomplete skeleton of a small individual (for details see *Delsett et al., 2017*) | Janusfjellet, Svalbard, Norway | Slottsmøya Member, Agardhfjellet Formation; upper Volgian, ?early Berriasian | *Keilhauia nui* (*Delsett et al., 2017*) | *Arthropterygius* sp. juv. cf. *A. chrisorum*. | 23M, 23R |
| PMO 224.250 | Incomplete basioccipital and basisphenoid, indet. cranial remains; incomplete pectoral girdle and forelimbs | Wimanfjellet, Svalbard, Norway. | Slottsmøya Member, Agardhfjellet Formation; "19 m above the echinoderm marker bed" ? *Crendonites anguinus* Ammonite Biozone; middle Volgian | Ophthalmosauridae indet. (*Delsett et al., 2018*) | *Arthropterygius* cf. *chrisorum* | – |

stages (as well as Bolonian and Portlandian stages in Northwest Europe) are used instead of the Tithonian and Berriasian international units. For detail on the Boreal–Tethyan correlation of the Volgian–Ryazanian and Tithonian–Berriasian we direct the reader to: (*Houša et al., 2007*; *Rogov & Zakharov, 2009*; *Rogov, 2012*, *2014*; *Bragin et al., 2013*). Most of the Upper Jurassic to lowermost Cretaceous marine reptile localities of the Northern Hemisphere (in European Russia, England, and Norway) belong to the Pan-Boreal Superrealm and their stratigraphic volume correspond to the Kimmeridgian, Volgian, and Ryazanian or equivalents.

*Stratigraphic position of specimens from European Russia.* All *Arthropterygius* specimens from European Russia originate from black shales of the Upper Jurassic (Middle Volgian) formations: Paromes Formation of the Timan-Pechora Basin (*Kravets, Mesezhnikov & Slonimsky, 1976*) and Promza Formation of the Volga Region (*Yakovleva, 1993*; *Mitta et al., 2012*). These formations correspond to the *Dorsoplanites panderi* Ammonite Biozone.

*Stratigraphic position of specimens from Franz-Josef Land.* Two ichthyosaur skeletons (CCMGE 3–16/13328 and CCMGE 17–44/13328) were found very close to each other, on the northeast slope of Berghaus Island, 150 m above sea level, in the uppermost part of a sequence of black shale and siltstone of the Hofer Formation (*Kosteva, 2005*). CCMGE 3–16/13328 was collected five m higher stratigraphically than CCMGE 17–44/13328. The layers with ichthyosaurs were filled with bivalves *Buchia unschensis*, *Buchia fischeriana*, and *B.* cf. *volgensis* (identifications are made by V.A. Zakharov, GIN) characteristic of the Jurassic/Cretaceous transitional interval of the Boreal Realm (*Zakharov, 1987*). On the adjacent slope, at a slightly higher level, ammonites *Surites* cf. *praeanalogus* were collected, indicating *Hecteroceras kochi* Ammonite Biozone of the Ryazanian (Lower Cretaceous) age (this and all subsequent ammonite identifications are made by M.A. Rogov, GIN); 20 m below, ammonites *Chetaites chetae*, index of the uppermost Ammonite Biozone of the Volgian of Arctic were collected; and finally, 50 m below the level of CCMGE 17–44/13328 on the same slope *Laugeites lambecki* and *Praechetaites* cf. *exoticus* were collected, indicating *Laugeites groenlandicus* Ammonite Biozone of the upper Middle Volgian (*Rogov & Zakharov, 2009*; *Rogov et al., 2016*). Absence of ammonite finds in the layers with ichthyosaurs do not allow to conclude with confidence whether they are from the uppermost Volgian or whether Ryazanian part of the section; however, it is unambiguous that the ichthyosaurs are of early Berriassian age (for comments on Jurassic–Cretaceous Boreal–Tethyan correlation as well as correlation of Boreal sections see, e.g., geological setting section in *Zverkov & Efimov, 2019*; a separate paper with details on the stratigraphy of Berghaus Island is currently in preparation).

*Comment on stratigraphic position of CMN 40608.* In the locality, Cape Grassy, Melville Island, shale, and siltstone of the Ringnes Formation are conformably overlain by soft, clay shales of the Deer Bay Formation (*Embry, 1994*). Elsewhere these lithologically similar formations are separated by sandstones of the Awingak Formation (*Embry, 1994*; *Poulton, 1994*). According to *Embry (1994)* the thickness of the Ringnes Formation in Cape

Grassy is *c*. 20 m (*Embry, 1994*: fig. 6). Taking this into consideration, the fact that CMN 40608 was found 51 m above the base of the Ringnes Formation, withal weathered out on the surface of the outcrop and slightly scattered (*Russell, 1994*), indicates that CMN 40608 was actually found within the Deer Bay Formation, but not Ringnes Formation as recorded by *Russell (1994)*. Considering that not much data is published on Late Jurassic invertebrates and biostratigraphy of Cape Grassy, it is uncertain what is the stratigraphic volume of the Ringnes and Deer Bay formations in this locality, as it varies significantly across the archipelago (*Embry, 1994*). In general, the stratigraphic span of the Ringnes Formation is considered as Oxfordian to Kimmeridgian and the span of the Deer Bay Formation is considered as Volgian to Valanginian (*Jeletzky, 1965*, *1973*; *Embry, 1994*; *Poulton, 1994*). In this regard, CMN 40608 is most likely Volgian if not even Ryazanian (Tithonian or Berriassian) in age.

## METHODS

### Phylogenetic analysis

For the phylogenetic analysis, we used recent matrix focused on ophthalmosaurids, presented by *Zverkov & Efimov (2019)*. One unit, *Keilhauia nui*, was removed, as the specimen it is based on is considered undiagnostic in the present contribution, see 'Discussion'. Two other units, *Arthropterygius volgensis* and *A. chrisorum* PMO 222.669 were added to the dataset. The scores for species of *Arthropterygius* were extended and partially changed based on new data (see Supplemental Materials for details).

Six new characters related to the morphology of the supratemporal, parietal, quadrate, coracoid, and humerus were added to the dataset (for details see Table S10). The new characters were coded from the literature for taxa that we have not personally examined (Table S11; *Gilmore, 1905*; *Broili, 1907*; *Andrews, 1910*; *Fraas, 1913*; *Sollas, 1916*; *Romer, 1968*; *McGowan, 1972*, *1973a*; *Johnson, 1979*; *Kirton, 1983*; *Wade, 1984*, *1990*; *Godefroit, 1993*; *Fernández, 1994*, *1997*, *1999*, *2007a*; *Bardet & Fernández, 2000*; *Maisch & Matzke, 2000*; *McGowan & Motani, 2003*; *Kear, 2005*; *Motani, 2005*; *Maxwell & Caldwell, 2006*; *Druckenmiller & Maxwell, 2010*; *Kolb & Sander, 2009*; *Zammit, Norris & Kear, 2010*; *Fischer et al., 2011*, *2012*, *2014a*, *2014b*; *Maxwell, Fernández & Schoch, 2012*; *Fernández & Talevi, 2014*; *Marek et al., 2015*; *Paparella et al., 2017*). The analysis was performed using TNT 1.5 (*Goloboff & Catalano, 2016*), applying traditional search with 10,000 replicates and tree bisection and reconnection with 100 trees saved per replication. The RAM allocation was extended to 1,024 MB and the memory to 10,000 trees. Decay indices (Bremer support, "suboptimal" = 5) and resampling methods to estimate the robustness of nodes (standard bootstrapping and jackknifing, 1,000 iterations) were also computed in TNT 1.5.

In order to eliminate problematic "wildcard" taxa, we used a posteriori approach of *Pol & Escapa (2009)*, that is, directly implemented in TNT 1.5. The two taxa (*Athabascasaurus bitumineus Druckenmiller & Maxwell, 2010* and *Platypterygius platydactulus Broili, 1907*) were identified as unstable and pruned from the second analysis. The pruned dataset was analysed using the exact same procedures as was used for the full dataset.

## Principal component analysis

To compare humeri of ophthalmosaurids we gathered a series of metrics and ratios that collectively summarize morphology of the humerus (Tables S6 and S7). The metrics are: proximodistal length of the humerus, anteroposterior width of humeral proximal and distal ends, thickness of humeral proximal end; dorsoventral width of humeral distal end; anteroposterior width at midshaft, anteroposterior and dorsoventral width of the distal facets, and the angle between the ulnar and radial facets (for details see Fig. S1). Based on the metrics the following ratios were calculated (Table S7):

(1) Humeral proximal expansion: anteroposterior width of humeral proximal end divided by the humeral proximodistal length.

(2) Humeral distal expansion: anteroposterior width of humeral distal end divided by the humeral proximodistal length.

(3) Humeral stoutness: humeral minimal anteroposterior width at diaphysis divided by the humeral proximodistal length.

(4) Humeral proximodistal proportionality: anteroposterior width of humeral proximal end divided by the same measurement of its distal end. The character based on this ratio is used in current phylogenetic analyses and distinguish ophthalmosaurids, which commonly have nearly equal proximal and distal humeral ends or proximal end slightly wider than the distal end see, for example, *Fischer et al. (2011*: Character 32*)*.

(5) Isometry of the humeral proximal end (or "anteroposterior elongation" of the humeral proximal end): anteroposterior width of humeral proximal end divided by the thickness of humeral proximal end (see Fig. S1). This ratio has extremely high value in "*Grendelius*" *zhuravlevi* (2.587) for which strongly compressed humeral proximal end is considered as autapomorphic (*Zverkov, Arkhangelsky & Stenshin, 2015*); the standard values for ophthalmosaurids are 1.8–1.5; for taxa with "isometric" humeral proximal end this value could be close to one (e.g., *Undorosaurus nessovi* and *Platypterygius platydactylus* see Table S7).

(6) Humeral distal compression: anteroposterior width of humeral distal end relative to the maximal dorsoventral width of humeral distal end.

(7) Relative anteroposterior width of facet for preaxial accessory epipodial element and radial facet.

(8) Relative anteroposterior width of ulnar and radial facets. As well as for ratio 4, there is a character based on similar ratios in current phylogenetic analyses, see, e.g., *Motani (1999*: Character 52*)* and *Moon (2019*: Character 209*)*. However, the referred character use "relative size" of ulnar and radial facets, which is not always clear as ulnar facet could be longer than radial facet but the same time, less wide dorsoventrally (as in most specimens of *Arthropterygius*). In this regard, it is better to consider separately relative anteroposterior width of ulnar and radial facets and relative dorsoventral width of ulnar and radial facets.

(9) Relative dorsoventral width of ulnar and radial facets.

The dataset is resolved at the specimen level with left and right humeri considered separately in order to reveal the existing humeral asymmetry within an individual and to assess its possible effects on the results. Data (see Tables S6 and S7) were collected based on personal observations of NGZ and completed by measurements and in rare cases analysis of pictures of the following references: (*Broili, 1907*; *Nace, 1939*; *Kuhn, 1946*; *Wade, 1984*; *Delair, 1986*; *McGowan, 1972*; *Arkhangelsky, 1998*; *Kolb & Sander, 2009*; *Maxwell, 2010*; *Maxwell & Kear, 2010*; *Moon & Kirton, 2016*). Only humeri with all documented ratios were considered, in rare cases, we completed our dataset by approximate ratios estimated based on oblique views (the case of *B. extremus* and *P. platydactylus*) or proportionally translated from other conspecific individuals (the case of *P. americanus*). The final dataset consisted of 55 humeri belonging to 45 individuals and 10 variables (Table S8). The ratios and angle between the ulnar and radial facets (in rad) were used as variables for the PCA. Data were scaled to equal variance by subtracting the mean value for each variable and then dividing each variable by the standard deviation. We then created a distance matrix with these data (Table S8). The dataset was analysed in PAST v. 3.20 (*Hammer, Harper & Ryan, 2001*).

## SYSTEMATIC PALAEONTOLOGY

**Ichthyosauria** *Blainville, 1835*

**Ophthalmosauridae** *Baur, 1887*

***Arthropterygius*** *Maxwell, 2010*

2010 *Arthropterygius Maxwell*: 403

2012 *Palvennia Druckenmiller, Hurum, Knutsen, Narkem*: 326

2014 *Janusaurus Roberts, Druckenmiller, Sætre & Hurum*: 4

2017 *Keilhauia Delsett, Roberts, Druckenmiller & Hurum*: 7

2018 *Palvennia Druckenmiller, Hurum, Knutsen, Narkem 2012*; *Delsett, Druckenmiller, Roberts, Hurum*: 8

**Type species:** *Ophthalmosaurus chrisorum Russell, 1994*

**Other valid species:** *Arthropterygius volgensis* (*Kasansky, 1903*) comb. nov., *A. hoybergeti* (*Druckenmiller et al., 2012*) comb. nov., *A. lundi* (*Roberts et al., 2014*) comb. nov.

**Emended diagnosis:** Moderate to large (three to five m) ichthyosaurs with following unique combination of features (autapomorphies are marked with "*"): relatively short and anteriorly pointed snout with snout ratio of *c.* 0.55 (the precise length of the snout is known exclusively for SVB 1451; pointed tip of the snout is known for PMO 222.669, PMO 222.655, SVB 1451, SGM 1502; the snout is relatively longer in all other ophthalmosaurids; it is not tapered in *Acamptonectes* and platypterygiines); strongly ventrally bowed jugal* (known for PMO 222.654, PMO 222.669, SVB 1451, CCMGE 17–44/13328); wide supratemporal anteromedial tongue covering the postfrontal (known for PMO 222.654, PMO 222.669, SVB 1451, CCMGE 17–44/13328; shared with *Athabascasaurus Druckenmiller & Maxwell, 2010*); extremely anteroposteriorly shortened

medial symphysis of parietals posteriorly restricted by a pronounced excavation and notch* (known for PMO 222.654, PMO 222.669, SVB 1451, KSU 982/P-213); large parietal foramen (known for SVB 1451, PMO 222.669, and PMO 222.654, although poorly preserved in the latter; this feature could be autapomorphic as it is currently unknown for other ophthalmosaurids, however, see Discussion); gracile quadrate with poorly developed "weak" condyle* (known for PMO 222.654, PMO 222.669, SVB 1451, CCMGE 3–16/13328, CCMGE 17–44/13328, KSU 982/P-213); basioccipital with extracondylar area wide in lateral view and practically unseen in posterior view (known for CMN 40608, PMO 222.654, PMO 222.669, SVB 1451); stapedial and opisthotic facets of the basioccipital shifted anteriorly and poorly visible in lateral view* (assessible for CMN 40608, PMO 222.654, PMO 222.669, SVB 1451; laterally exposed in other known ophthalmosaurids); basisphenoid with foramen for the internal carotid arteries opening posteriorly* (known for CMN 40608, PMO 222.669, SVB 1451, CCMGE 3–16/13328, CCMGE 17–44/13328, SGM 1502, SGM 1743-2, KSU 982/P-213); basioccipital facet of the basisphenoid facing posterodorsally, occupying in dorsal view area equal or even larger than that of dorsal plateau* (known for the same specimens as listed for previous character, except for PMO 222.669, in which this part is obscured); stapes with extremely gracile shaft (known for PMO 222.654, PMO 222.669, SVB 1451; among ophthalmosaurids shared with *Acamptonectes* Fischer et al., 2012); short and robust paraoccipital process of the opisthotic (known for PMO 222.654, PMO 222.669, SVB 1451, KSU 982/P-213; unlike that slender in *Ophthalmosaurus* and *Acamptonectes*); wide and extremely robust clavicles* (known for PMO 222.654, PMO 222.655, PMO 222.669, SVB 1451, CCMGE 17–44/13328); bulge in the middle of the interclavicle posterior median stem* (assessible only in SGM 1573 and PMO 222.654); large coracoids (proximodistal length of the scapula reduced in comparison to coracoid length) (known for CMN 40608, PMO 222.654, PMO 222.669, PMO 224.250, CCMGE 3–16/13328, SGM 1731-01–15, KSU 982/P-213); pronounced angle close to 90–100° between the articulated coracoids* (assessible for PMO 222.654, CCMGE 3–16/13328, SGM 1731-01–15, KSU 982/P-213); ventral skew between the radial and ulnar facets of the humerus (ulnar facet:radial facet dorsoventral width ratio less than 0.8; as in *Sisteronia* Fischer et al., 2014b) (known for CMN 40608, PMO 222.654, PMO 222.655, PMO 222.669, PMO 224.250, CCMGE 3–16/13328, CCMGE 17–44/13328, SGM 1502, SGM 1574, SGM 1731-01–15, YKM 63548, UPM 2442); three concave distal articular facets on humerus for a preaxial accessory element, radius and ulna (shared with most of other ophthalmosaurids except for *Nannopterygius*, *Brachypterygius*, *Aegirosaurus*, *Grendelius*, *Caypullisaurus*, and some *Platyptepterygius* spp.); ulna larger than the radius in dorsal view and lacking posterior perichondral ossification (ulna is known for CMN 40608, PMO 222.654, PMO 222.669, PMO 224.250, CCMGE 3–16/13328, CCMGE 17–44/13328, SGM 1731-01–15, YKM 63548; this condition is uncommon for ophthalmosaurines sensu Fischer et al., 2012); "latipinnate" forefin architecture with two distal carpals (four and three) contacting the intermedium, and distal ulnare/metacarpal five contact (among ophthalmosaurids shared with *Ophthalmosaurus*, *Brachypterygius*, and *Aegirosaurus*); autopodial elements circular in outline and loosely arranged (shared with *Ophthalmosaurus*); plate-like ishiopubis, lacking the obturator foramen (known for PMO

222.654, 222.655, and CCMGE 17–44/13328, although fragmental in the latter; absence of the obturator foramen shared with derived platypterygiines); ilium anteroposteriorly expanded at the dorsal end (known for PMO 222.654 and PMO 222.655; among other ophthalmosaurids shared with an ophthalmosaurid described by *Bauer (1898)*, however, it is not impossible that the latter represents *Arthropterygius*, in this case, the trait is autapomorphy of *Arthropterygius*).

**Occurrence:** Arctic Canada, Russian Extreme North (Franz Josef Land) and the European part of Russia, Norway (Svalbard) and Argentina (Neuquen Basin). Middle to Upper Volgian–Ryazanian (Tithonian–Berriassian) (see *Maxwell, 2010*; *Fernández & Maxwell, 2012*; *Druckenmiller et al., 2012*; *Roberts et al., 2014*; *Zverkov et al., 2015*; *Delsett et al., 2016*, *2017*).

### Remarks on the synonymy of *Arthropterygius*, *Palvennia*, *Janusaurus*, and *Keilhauia*

Based on the incomplete type specimen of *A. chrisorum* solely, the autapomorphic features of *Arthropterygius* are: basisphenoid with foramen for the internal carotid arteries opening posteriorly; basioccipital facet of the basisphenoid facing posterodorsally and occupying a half of the element in dorsal view; basioccipital with extracondylar area wide in lateral view and practically unseen in posterior view; shifted anteriorly stapedial and opisthotic facets of the basioccipital; presence of "ulnar torsion," with ulnar facet not as dorsoventrally wide as the radial facet, forming a distal skew of the humeral ventral surface (*Maxwell, 2010*; *Zverkov et al., 2015*; Nikolay G. Zverkov, personal observations, 2015–2016). All these features could be observed in the type specimens of genera that are here synonymized with *Arthropterygius*, except for cases where an element is unknown or obscured from observation: basisphenoid is mostly hidden in the holotype of *Janusaurus lundi*; humerus is incomplete in the holotype of *Palvennia hoybergeti*; basicranial elements are not preserved in the holotype of *Keilhauia nui*, and both the basioccipital and humerus are absent in the holotype of *Ichthyosaurus volgensis*.

The epipodial elements angular in outline for articulation with humerus in the holotype of *Arthropterygius chrisorum* were also considered as an autapomorphy of *Arthropterygius* (*Maxwell, 2010*). However, this condition was later reported for another ophthalmosaurid (*Zverkov et al., 2015*) and, in this regard, should be further considered with caution.

From our observations on the holotype of *Keilhauia nui* (PMO 222.655), we are unable to confirm any of the evidence proposed by *Delsett et al. (2017)* as supporting the maturity of the specimen. The specimen is too fragmentary and poorly preserved, thus we consider it referable to *Arthropterygius* only in open nomenclature. For details on this decision see Discussion.

In previous phylogenetic analyses, the position and relations of *Arthropterygius*, *Janusaurus*, *Palvennia*, and *Keilhauia* varied sufficiently and these taxa never formed a clade (*Roberts et al., 2014*; *Delsett et al., 2017*; *Fischer et al., 2016*; *Maxwell et al., 2016*; *Motani et al., 2017*; *Paparella et al., 2017*; *Moon, 2019*). In some phylogenetic hypotheses, *Arthropterygius*, *Janusaurus*, and *Palvennia* were recovered as sister taxa to each other,

but also to a clearly distinct ophthalmosaurid "*Cryopterygius*" (*Roberts et al., 2014*; Maxwell et al., 2016; *Delsett et al., 2016*). These results could have been considered as an argument for the validity of all these genera, however, the recent results of *Zverkov & Efimov (2019)* recovered *Arthropterygius*, *Janusaurus*, *Palvennia*, and *Keilhauia* in a clade distantly related to "*Cryopterygius*." This clade, called the "*Arthropterygius* clade" in *Zverkov & Efimov (2019)*, has relatively good support and its potential genus rank was announced in that paper. The phylogenetic results of the present contribution (see below) helps to further develop this idea. In this research, *Arthropterygius* clade is supported by nine autapomorphies: posterior position of the foramen for internal carotid arteries; dorsally facing basioccipital facet of the basisphenoid; raised opisthotic facet of the basioccipital; anteriorly shifted stapedial and opisthotic facets of the basioccipital; gracile stapedial shaft; weak quadrate condyle; robust clavicles; ulnar facet/radial facet ratio less than 0.83; pronounced angle between the articulated coracoids. For the taxa within the *Arthropterygius* clade, no unambiguous autapomorphies are found so that their generic independency could hardly be justified.

### *Arthropterygius chrisorum* (*Russell, 1994*)

(Figs. 2–8)

v.1910? *Ophthalmosaurus thyreospondylus* Owen; *Bogolubov*: 474

*1994 *Ophthalmosaurus chrisorum* Russell*: 198, fig. 3

2010 *Arthropterygius chrisorum* (*Russell, 1994*); *Maxwell*: 404, figs. 2–5

v.2018 *Palvennia hoybergeti* Druckenmiller et al., 2012; *Delsett, Druckenmiller, Roberts, Hurum*: 8, figs. 5–13 [*pars.*]

**Holotype:** CMN 40608, fragmentary skeleton of a large mature individual (for details see *Maxwell, 2010*).

**Referred specimens:** PMO 222.669, SGM 1573, CCMGE 3–16/13328, CCMGE 17–44/13328. See Table 1

**Emended diagnosis:** A moderately large (four to five m) ichthyosaur, diagnosed relative to other species of *Arthropterygius* by the following unique characters: quadrate with strongly ventrally shifted articular boss, V-shaped in posteromedial view; absence of pronounced angular protrusion of the quadrate (known for PMO 222.669, CCMGE 3–16/13328, CCMGE 17–44/13328); basisphenoid trapezoidal in outline with maximum mediolateral width in its anterior part; posterior foramen for the internal carotid arteries not visible in ventral view in adults, separated from the ventral surface by a thin shelf (known for CMN 40608, PMO 222.669, CCMGE 3–16/13328, CCMGE 17–44/13328); dorsoventrally high opisthotic with extremely reduced and robust paraoccipital process (hitherto found only in PMO 222.669); blunt termination of the lateral extremities of the interclavicle (this complete element is found only in SGM 1573); strongly anteroposteriorly elongated proximal end of the humerus with reduced deltopectoral crest shifted to its anterior edge (humeri are known for all the referred specimens); extremely pronounced ventral skew

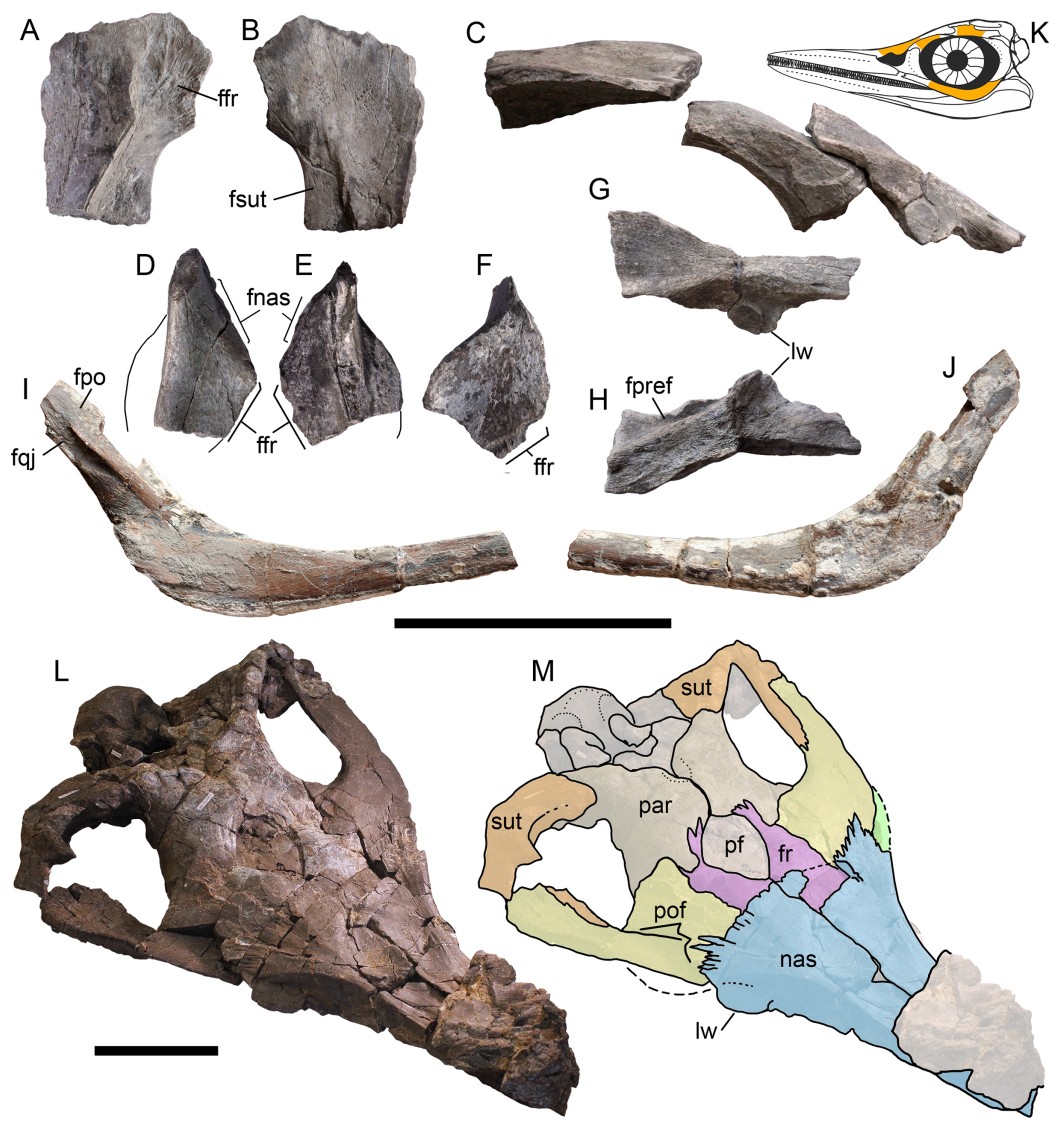

**Figure 2 Cranial remains of *Arthropterygius chrisorum* CCMGE 17–44/13328 (A–J) and PMO 222.669 (L, M).** (A, B) Right postfrontal in ventral (A) and dorsal (B) views. (C) Left lateral view on articulated postfrontal, prefrontal and nasal. (D) Left prefrontal in ventral view. (E, F) Right prefrontal in ventral (E) and dorsal (F) views. (G, H) left nasal in dorsal (G) and ventral (H) views. (I, J) Left jugal in medial (I) and lateral (J) views. (K) Cranial reconstruction, showing the depicted elements (colored). (L, M) oblique dorsal view and interpretation of sutures of the skull roof of PMO 222.669. Abbreviations: ffr, facet for the frontal; fnas, facet of the nasal; fpo, facet for the postorbital; fpref, facet for the prefrontal; fqj, facet for the quadratojugal; fsut, facet for the supratemporal; lw, lateral wing of the nasal lamella; nas, nasal; par, parietal; pf, parietal foramen; pref, prefrontal; sut, supratemporal. Both scale bars represent 10 cm.

between the ulnar and radial facets of the humerus; facet for the anterior accessory epipodial element of the humerus as wide as, and equal in size to the radial facet.

**Occurrence:** Upper Jurassic, Deer Bay Formation (likely its Volgian part) of Melville Island, Northwest Territories, Canada (type locality); Middle Volgian Promza Formation (*Dorsoplanites panderi* Ammonite Biozone) of Ulyanovsk Region, Russia; upper part of the

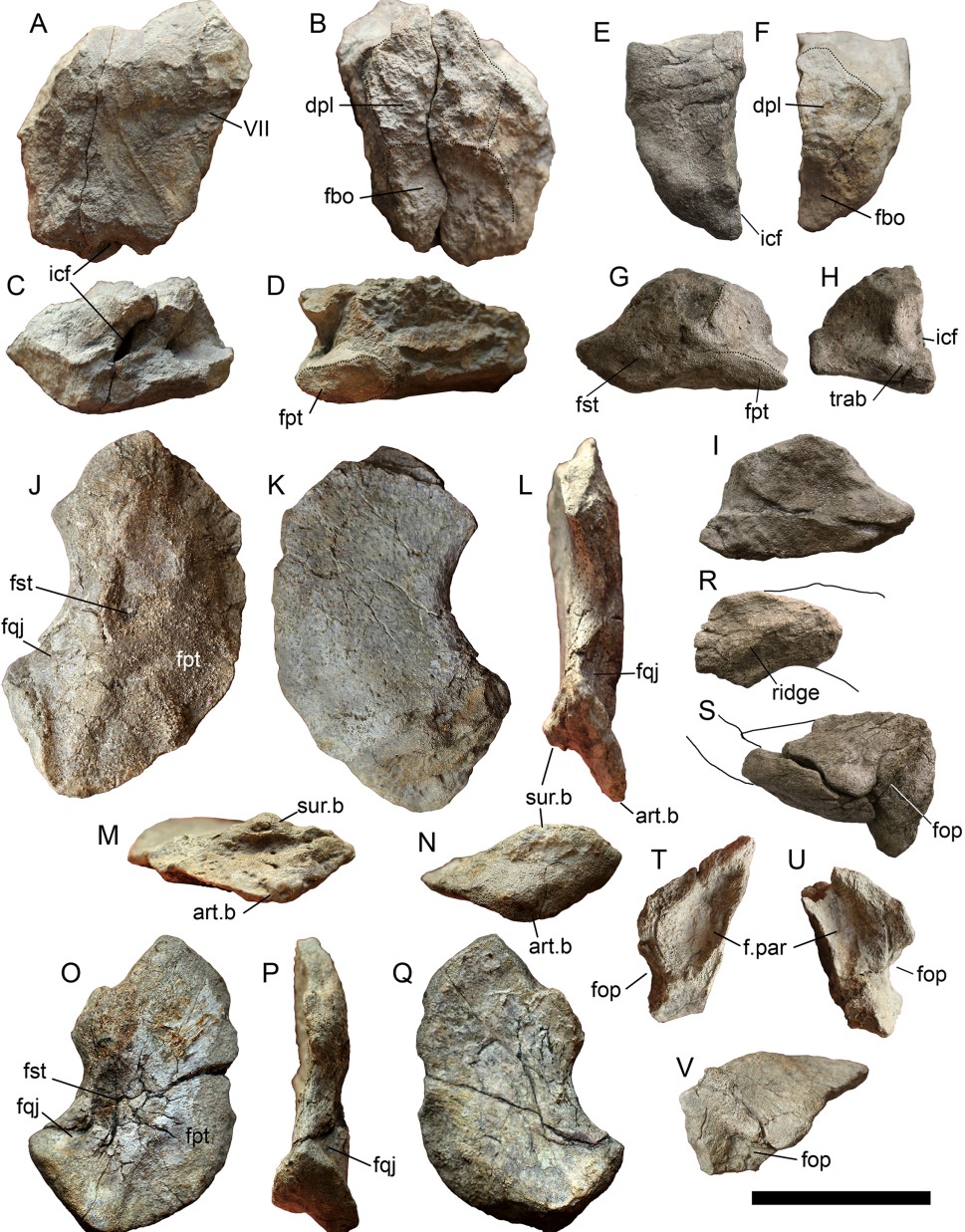

**Figure 3 Cranial elements of *Arthropterygius chrisorum* CCMGE 3–16/13328 and 17–44/13328.**
(A–I) Basisphenoids of CCMGE 17–44/13328 (A–D) and CCMGE 3–16/13328 (E–I) in ventral
(A, E), dorsal (B, F), anterior (C, H) and lateral (D, G) views, and sagittal section of the basisphenoid (I).
(J–Q) Left quadrates of CCMGE 17–44/13328 (J–M) and CCMGE 3–16/13328 (N–Q) in posteromedial
(J, O), anterolateral (K, Q), posterolateral (L, P) and ventral (M, N) views. (R) Supratemporal process of
the right parietal of CCMGE 3–16/13328 in dorsal view. (S) Articulated fragments of the right supra-
temporal and parietal of CCMGE 3–16/13328 in posterior view. (T, V) Medial ramus of the left
supratemporal of CCMGE 3–16/13328 in medial (T) and posterior (V) views; (U) medial ramus of the
right supratemporal of CCMGE 3–16/13328 in medial view. Abbreviations: art.b, articular boss; dpl,
dorsal plateau of the basisphenoid; fbo, facet for the basioccipital; fop, facet for the opisthotic; fpt, facet for
the pterygoid; fqj, facet for the quadratojugal; fst, facet for the stapes; icf, foramen for the internal carotid
arteries; sur.b, surangular boss; trab, facets for cartilaginous continuation of the *cristae trabeculares*; VII,
groove of the palatine ramus of facial (VII) nerve. Scale bar represents five cm.

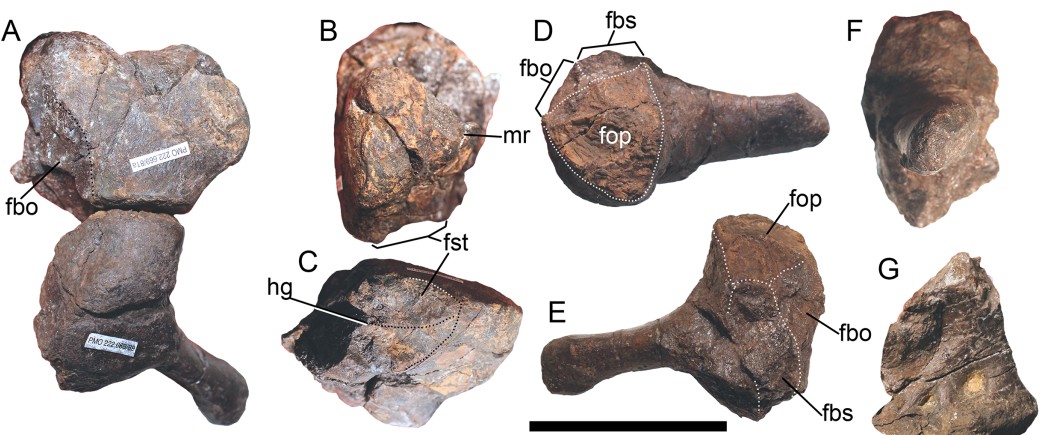

**Figure 4 Opisthotic, stapes, and exoccipital of *Arthropterygius chrisorum* PMO 222.669.** (A) Articulated right opisthotic and stapes in posterior view. (B, C) Right opisthotic in lateral (B) and ventral (C) views. (D–F) Right stapes in dorsal (D), anterior (E), and lateral (F) views; (G) right exoccipital in medial view. Abbreviations: fbo, facet for the basioccipital; fbs, facet for the basisphenoid; fst, facet for the stapes; hg, groove for transmission of hyomandibular branch of facial (VII) or glossopharyngeal (XI) nerve; mr, muscular ridge on the opisthotic. Scale bar represents five cm.

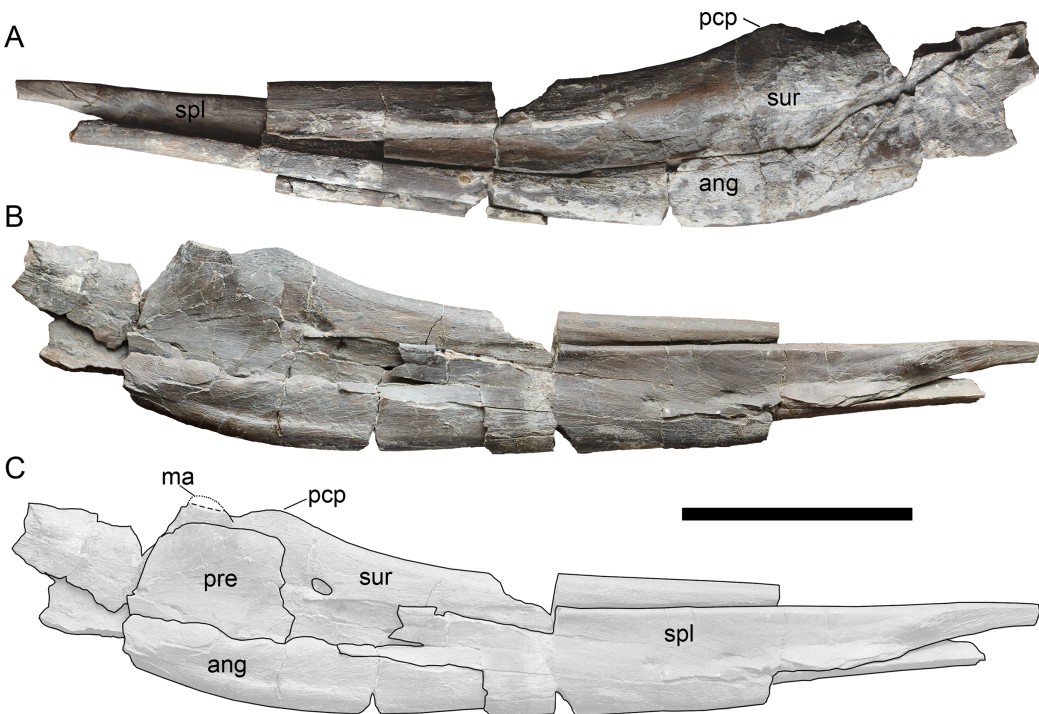

**Figure 5 Left mandibular ramus of *Arthropterygius chrisorum* CCMGE 17–44/13328 in lateral (A) and medial (B, C) views.** Abbreviations: ang, angular; ma, muscle (*M. adductor mandibulae externus*) attachment point; pcp, paracoronoid process; pre, prearticular; spl, splenial; sur, surangular. Scale bar represents 10 cm.

Hofer Formation (uppermost Volgian to lowermost Ryazanian, Berriasian) of Franz-Josef Land, Russian Extreme North; Slottsmøya Member of the Agardhfjellet Formation (Middle Volgian part of the section) of Svalbard, Norway.

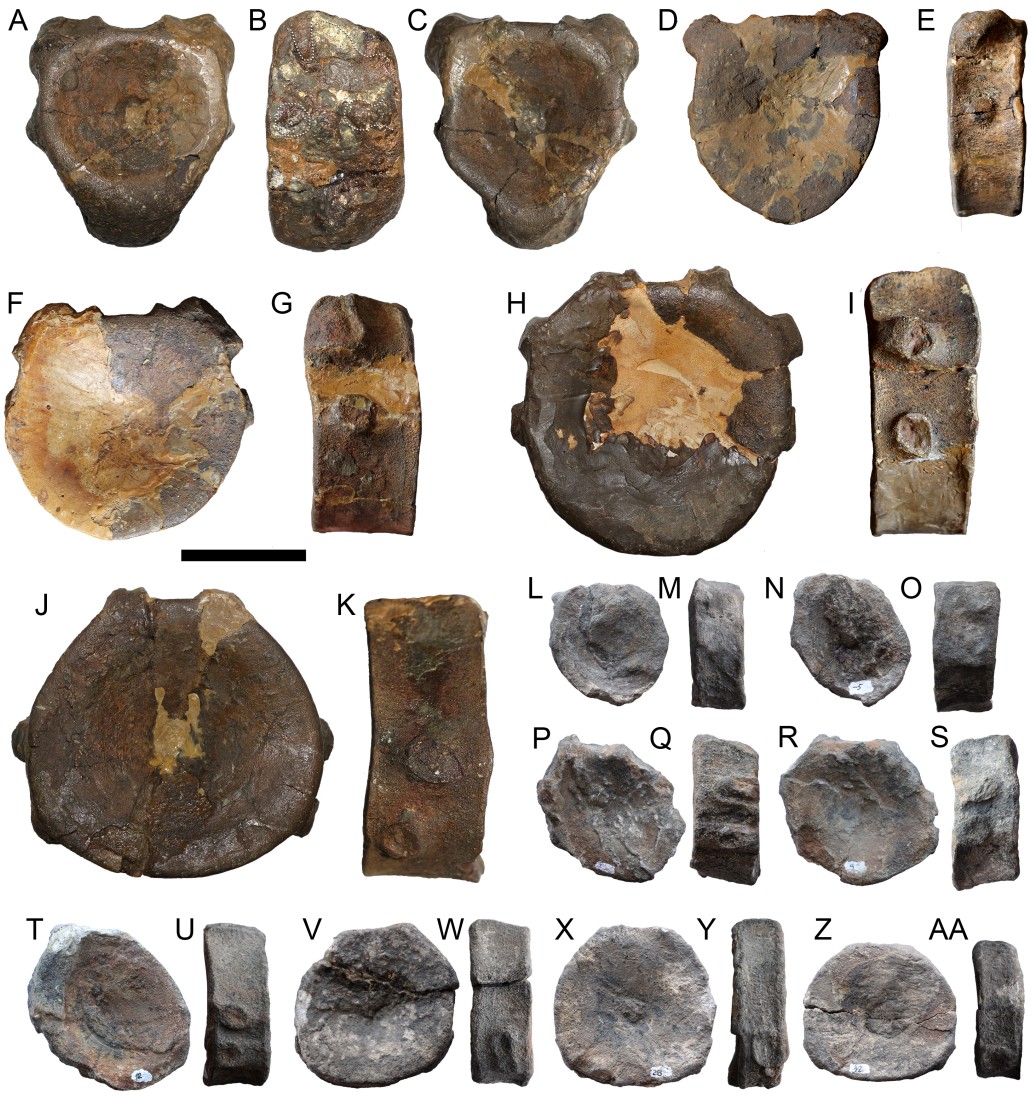

**Figure 6 Selected vertebral centra of *Arthropterygius chrisorum* SGM 1573 (A–K) and CCMGE 3–16/13328 (L–AA).** (A–C) Atlas-axis complex in anterior (A), right lateral (B), and posterior (C) views. (D–G, L–O) Anterior presacral vertebral centra. (H–K, P–U) Posterior presacral vertebral centra. (V–AA) Caudal centra. Each centrum depicted in articular and lateral views, respectively. Scale bar represents five cm.

## DESCRIPTION

**Skull.** The skull of *A. chrisorum* is now well-known due to a new find from Svalbard (PMO 222.669; *Delsett et al., 2018*). Thereby here we provide only some additional observations on the referred specimens, with special reference to new specimens from Franz Joseph Land. For more details on cranial morphology of *A. chrisorum* see the description of PMO 222.669 in *Delsett et al. (2018)*. Some additional observations and different interpretations of some sutures in PMO 222.669 could be found in sections with the description of the nasal, prefrontal, parietal, basioccipital, basisphenoid, opisthotic, and stapes.

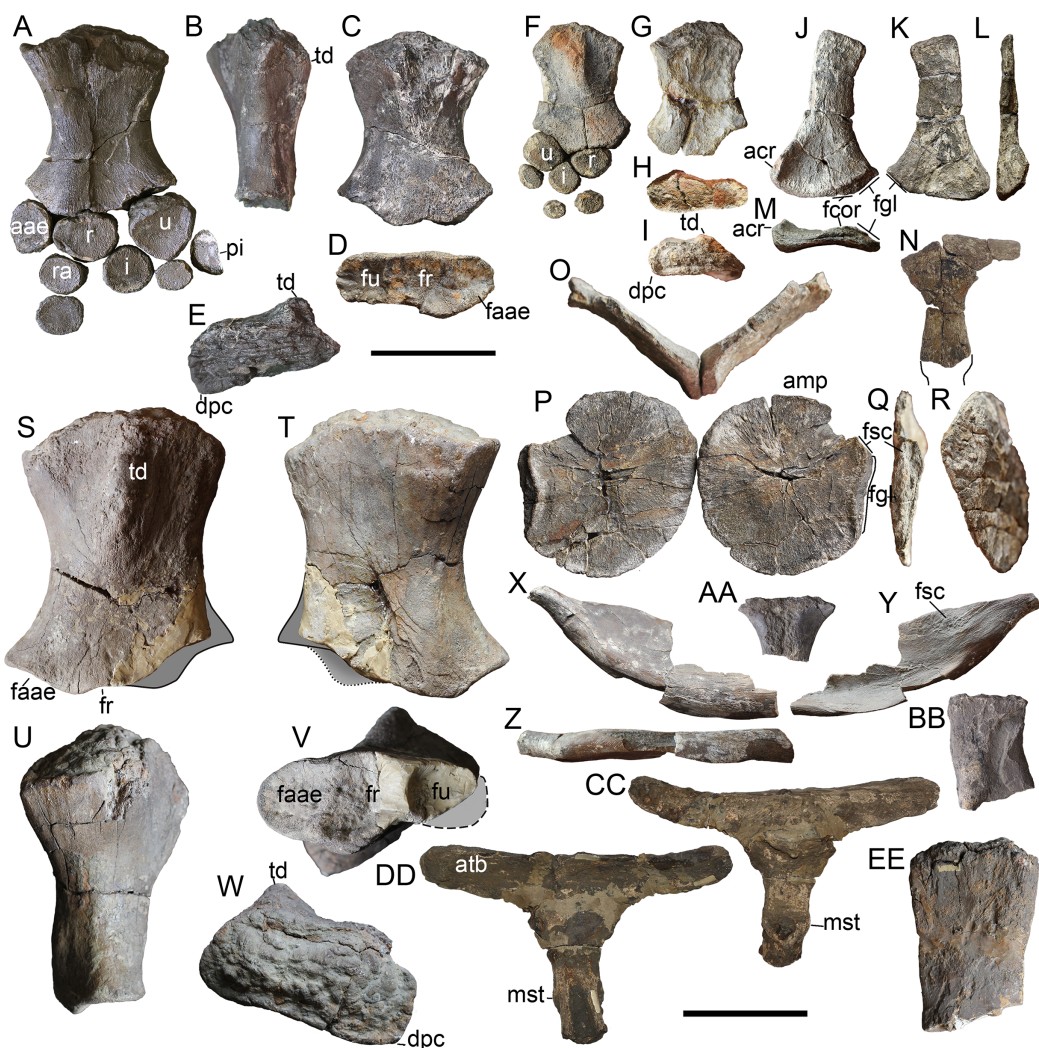

**Figure 7** Forelimb and pectoral girdle elements of *Arthropterygius chrisorum* CCMGE 17–44/13328 (A–E, X–BB), CCMGE 3–16/13328 (F–R), and SGM 1573 (S–W, CC–EE). (A) Right forelimb of CCMGE 17–44/13328 in ventral view. (B–F) Right humerus of CCMGE 17–44/13328 in posterior (B), dorsal (C), distal (D), and proximal (E) views. (F) right forelimb of CCMGE 3–16/13328 in dorsal view. (G–K) Right humerus of CCMGE 3–16/13328 in ventral (G), distal (H), anterior (I), posterior (J), and proximal (K) views. (J–M) Left scapula of CCMGE 3–16/13328 in lateral (J), medial (K), anterior (L), and proximal (M) views. (N) Interclavicle of CCMGE 3–16/13328; (O–R) coracoids of CCMGE 3–16/13328 in anterior (O) and ventral disarticulated (P) views, lateral (Q), and medial (R) views of the right coracoid. (S–W) Right humerus of SGM 1573 in dorsal (S), ventral (T), anterior (U), distal (V), and proximal (W) views. (X–Z) Right clavicle of CCMGE 17–44/13328 in anterior (X), posterior (Y), and ventral (Z) views. (AA) Fragmentary interclavicle of CCMGE 17–44/13328 in dorsal view. (BB) Dorsal ramus of the left scapula of CCMGE 17–44/13328 in lateral view. (CC, DD) interclavicle of SGM 1573 in ventral (CC) and dorsal (DD) views. (EE) fragmentary dorsal ramus of the scapula of SGM 1573. Abbreviations: aae, anterior accessory epipodial element; acr, acromial process; amp, anteromedial process of the coracoid; atb, anterior transverse bar of the iterclavicle; dpc, deltopectoral crest; faae, facet for the anterior accessory epipodial element; fcor, facet for the coracoid; fgl, glenoid contribution; fr, facet for the radius; fsc, facet for the scapula; fu, facet for the ulna; i, intermedium; mst, bulge in the middle of the interclavicle posterior median stem; pi, pisiform; r, radius; ra, radiale; td, dorsal process; u, ulna; ul, ulnare. Scale bar represents 10 cm.

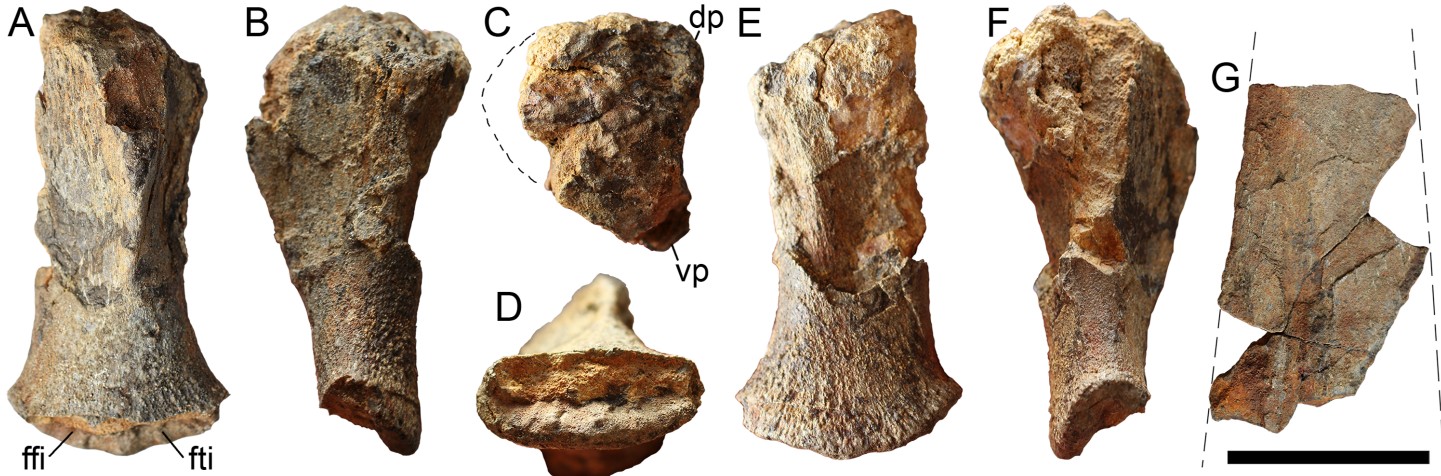

**Figure 8** Left femur (A–F) and partial ischiopubis (G) of *Arthropterygius chrisorum* CCMGE 17–44/13328. Femur in ventral (A), anterior (B), proximal (C), distal (D), dorsal (E), and posterior (F) views. Abbreviations: dp, dorsal process of the femur; ffi, facet for the fibula; fti, facet for the tibia; vp, ventral process of the femur. Scale bar represents five cm.

**Nasal.** The supranarial portion of the right nasal is preserved in CCMGE 17–44/13328 (Figs. 2C, 2G and 2H). It is too fragmentary for substantial description, however, from this fragment it could be said that the nasal lamella is well developed and forms a pronounced lateral "wing" overhanging the dorsal border of the external naris (Figs. 2G and 2H). Similar lateral "wing" is known for *Ophthalmosaurus* and *Acamptonectes* (*Fischer et al., 2012*; *Moon & Kirton, 2016*), but this structure is less pronounced or even completely reduced in platypterygiines (*Druckenmiller & Maxwell, 2010*; *Fischer et al., 2014a*, *2014c*; *Zverkov & Efimov, 2019*). In PMO 222.669 both nasals are preserved in articulation. To the description of these elements provided by *Delsett et al. (2018)*, we could add that the nasal bears a pronounced lateral "wings" over the external naris (Figs. 2L and 2M). The posterior portion of the nasal articulates with the postfrontal and frontal in a complex interdigitating suture, covering most of the frontal anteriorly (Fig. 2M). Posteriorly, the dorsal surface of the nasal is shallowly concave, forming an excavatio internasalis, that is, constricted laterally and medially by a raised areas.

**Prefrontal.** Although incomplete, both prefrontals are preserved in CCMGE 17–44/13328 (Figs. 2D–2F). These elements are composed of a dorsal sheet and robust, anteroventrally directed strut, forming the anterodorsal margin of the orbit (Figs. 2C and 2K). A straight ridge along the medial edge of the dorsal sheet meets a deep groove in the lateral margin of the overlapping nasal (Figs. 2D and 2E). Anterior to it, there is a facet for articulation with the frontal. When articulated with other elements, prefrontal had little dorsal exposure, being covered by the anterior plate of the postfrontal posteriorly and by the nasal anteromedially. In PMO 222.669, prefrontals are practically unseen dorsally, being covered by postfrontals and nasals (Figs. 2L and 2M).

**Parietal.** Only posterolateral processes of the parietal are preserved in both CCMGE 3–16/13328 and 17–44/13328. In this regard, the only observation that could be made is that

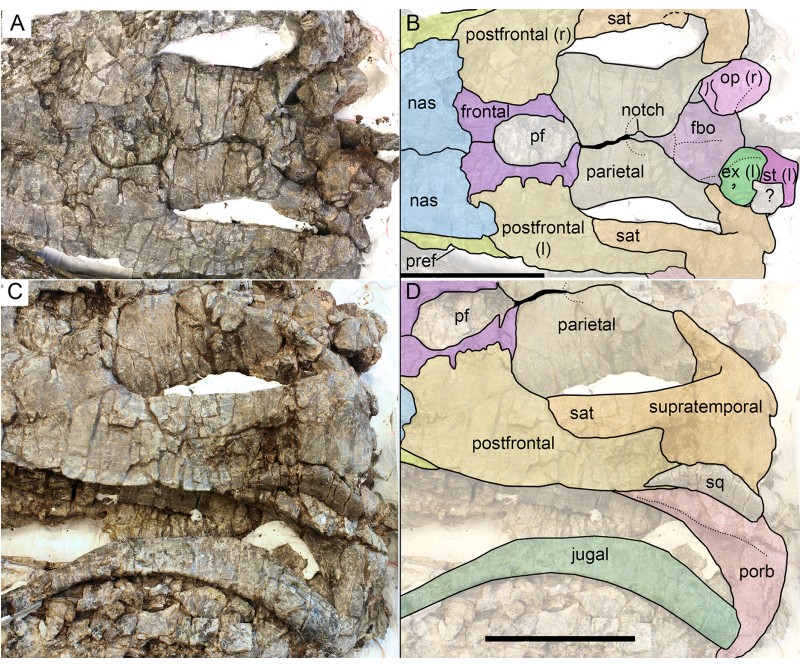

**Figure 9 Skull roof (A, B) and postorbital region (C, D) of *Arthropterygius hoybergeti* SVB 1451.** Abbreviations: ex(l), left exoccipital; fbo, facet of the basisphenoid for the basioccipital; nas, nasal; op(r), right opisthotic; pf, parietal foramen; porb, postorbital; pref, prefrontal; sat, supratemporal anteromedial tongue; sq, squamosal; st(l), left stapes. Scale bars represent 10 cm.

the process was slender but not robust as in *Undorosaurus* and some other platypterygiines (for comments on this character see *Zverkov & Efimov, 2019*). The parietals of PMO 222.669 are complete and articulated. In the original description (*Delsett et al., 2018*), the skull was not completely prepared of embedded rock, so that the posteromedial excavation and notch of the parietals were not seen. In general, the parietal of PMO 222.669 demonstrates characteristic morphology with the relatively slender posterolateral process and short but robust medial symphysis restricted posteriorly by a pronounced notch (Figs. 2L and 2M). This condition is currently known excusively for *Arthropterygius*, in other ophthalmosaurids the interparietal symphysis is anteroposteriorly long and completely occupies the medial edge of the parietal (*Kear, 2005*; *Druckenmiller & Maxwell, 2010*; *Fischer, 2012*; *Fischer et al., 2014a*; *Moon & Kirton, 2016*; *Zverkov & Efimov, 2019*).

**Postfrontal.** The partial right postfrontal is preserved in CCMGE 17–44/13328. An extensive facet of the supratemporal anteromedial tongue occupy nearly a half of the element mediolateral width dorsally and terminates right before the expansion of the anterior plate in an interdigitating suture (Figs. 2B, 2L and 2M). This condition is similar to that of *A. hoybergeti* (SVB 1451) and *A. lundi* (see Descriptions below), and among other ophthalmosaurids, it occurs only in not closely related *Athabascasaurus* (*Druckenmiller & Maxwell, 2010*); thus it could likely be considered as a non-unique autapomorphy of *Arthropterygius*. *Delsett et al. (2018)* described more short and gracile

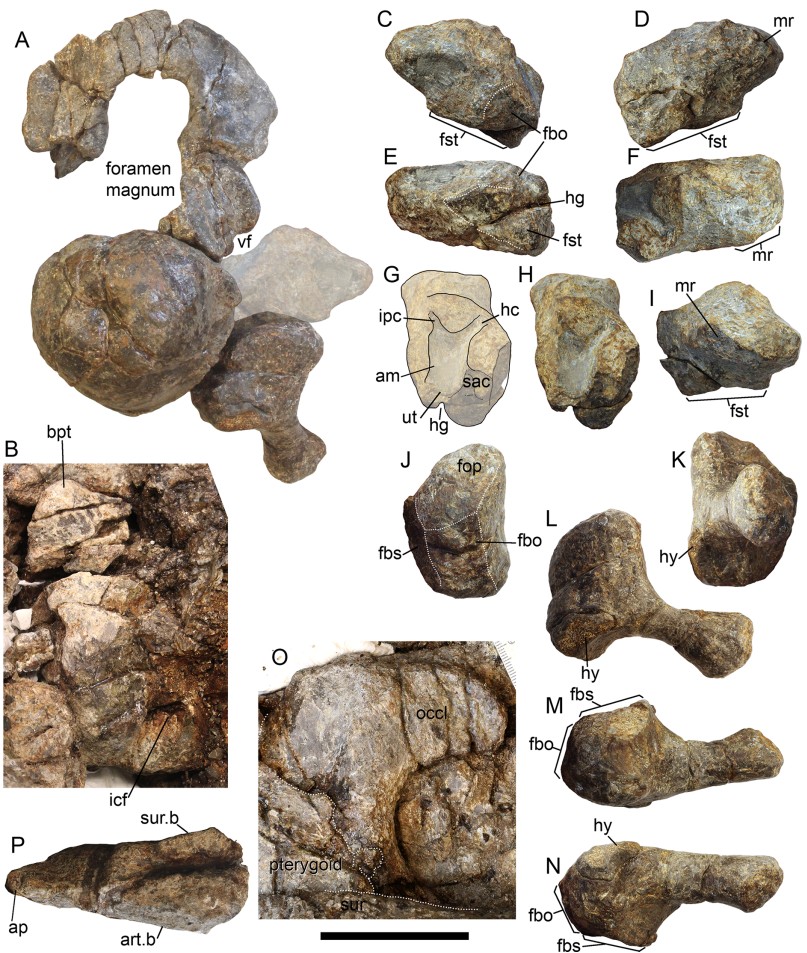

**Figure 10 Occipital region elements of *Arthropterygius hoybergeti* SVB 1451.** (A) Partially reconstructed occiput in oblique posterodorsal view (left opisthotic is mirrored and mounted as right in order to complement the picture). (B) Basisphenoid in ventral view. (C–I) Left opisthotic in posterior (C), anterior (D), ventral (E), dorsal (F), medial (G, H), and lateral (I) views. (J–N) Right stapes in medial (J), distal (K), posterolateral (L), dorsal (M), and ventral (N) views. (O) Right quadrate in posteromedial view. (P) Left quadrate in ventral view. Abbreviations: am, ampulla; ap, angular protrusion of the quadrate; art.b, articular boss; bpt, basipterygoid process; fbo, facet for the basioccipital; fbs, facet for the basisphenoid; fst, facet for the stapes; hc, impression of horizontal semicircular canal; hg, groove for transmission of hyomandibular branch of facial (VII) or glossopharyngeal (XI) nerve; hy, hyoid process; icf, foramen for the internal carotid arteries; ipc, impression of posterior vertical semicircular canal; mr, muscular ridge on the opisthotic; occl, occipital lamella; sac, sacculus; sur.b, surangular boss; ut, utriculus; vf, vagus foramen. Scale bar represents five cm.     

"supratemporal finger" = supratemporal anteromedial tongue, however, this is due to difficulties in tracing of sutures (see alternative interpretation on Figs. 2L and 2M).

**Supratemporal.** Medial rami of both supratemporals are preserved in CCMGE 3–16/13328. These portions are massive and quite short mediolaterally bearing triangular and excavated medial facets for articulation with the parietal (Figs. 3S–3U). Ventrolaterally to this facet, there is a small depression of the facet for the paroccipital process of the opisthotic (Figs. 3S–3V). It should be mentioned also that we were unable to verify the statement of *Delsett et al. (2018*: 22*)*, that the lateral process of the supratemporal in PMO

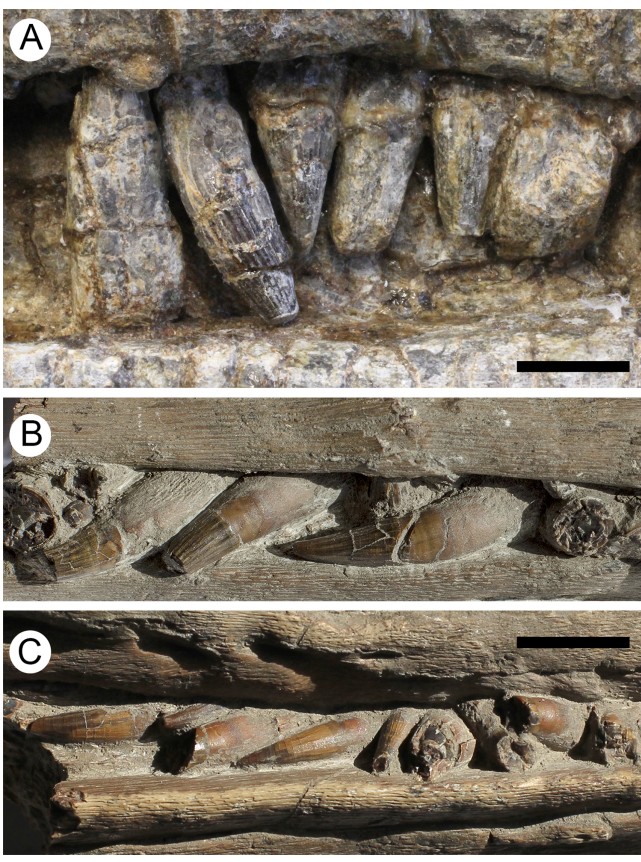

**Figure 11 Teeth of *Arthropterygius hoybergeti* SVB 1451 (A) and *A. lundi* SGM 1502 (B, C).** Scale bars represent 10 mm.

222.669 contacts the stapes. The occipital region in that specimen is strongly dorsoventrally compressed, thus hampering the assessment of this character.

**Jugal.** The jugal of CCMGE 17–44/13328 is a slender, strongly bowed J-shaped element (Figs. 3I and 3J). Its posterior part is mediolaterally compressed, ascending dorsally as a slender process and forming the posterior part of the orbit (Fig. 2K). On its medial surface, the process bears facets for the postorbital and quadratojugal (Fig. 2I). The suborbital portion of the jugal is strongly bowed, greater than that of *Ophthalmosaurus icenicus* (*Moon & Kirton, 2016*) but in similar degree to those of *Arthropterygius hoybergeti* and *A. lundi*.

**Quadrate.** The quadrate is known for both CCMGE 3–16/13328 and 17–44/13328 (strongly compressed). It is a relatively gracile ear-shaped element. The posterodorsal part of the occipital lamella is broken in both CCMGE specimens so it is hard to say anything regarding its natural shape. Owing to its complete preservation in PMO 222.669, we know that the occipital lamella is well developed. It is even more protruding than those of *Ophthalmosaurus* and *Acamptonectes* (*Fischer et al., 2012*; *Moon & Kirton, 2016*); whereas in most other ophthalmosaurids the occipital lamella is relatively reduced (*Broili, 1907*; *Kear, 2005*; *Kolb & Sander, 2009*; *Fischer et al., 2014b*; *Zverkov & Efimov, 2019*).

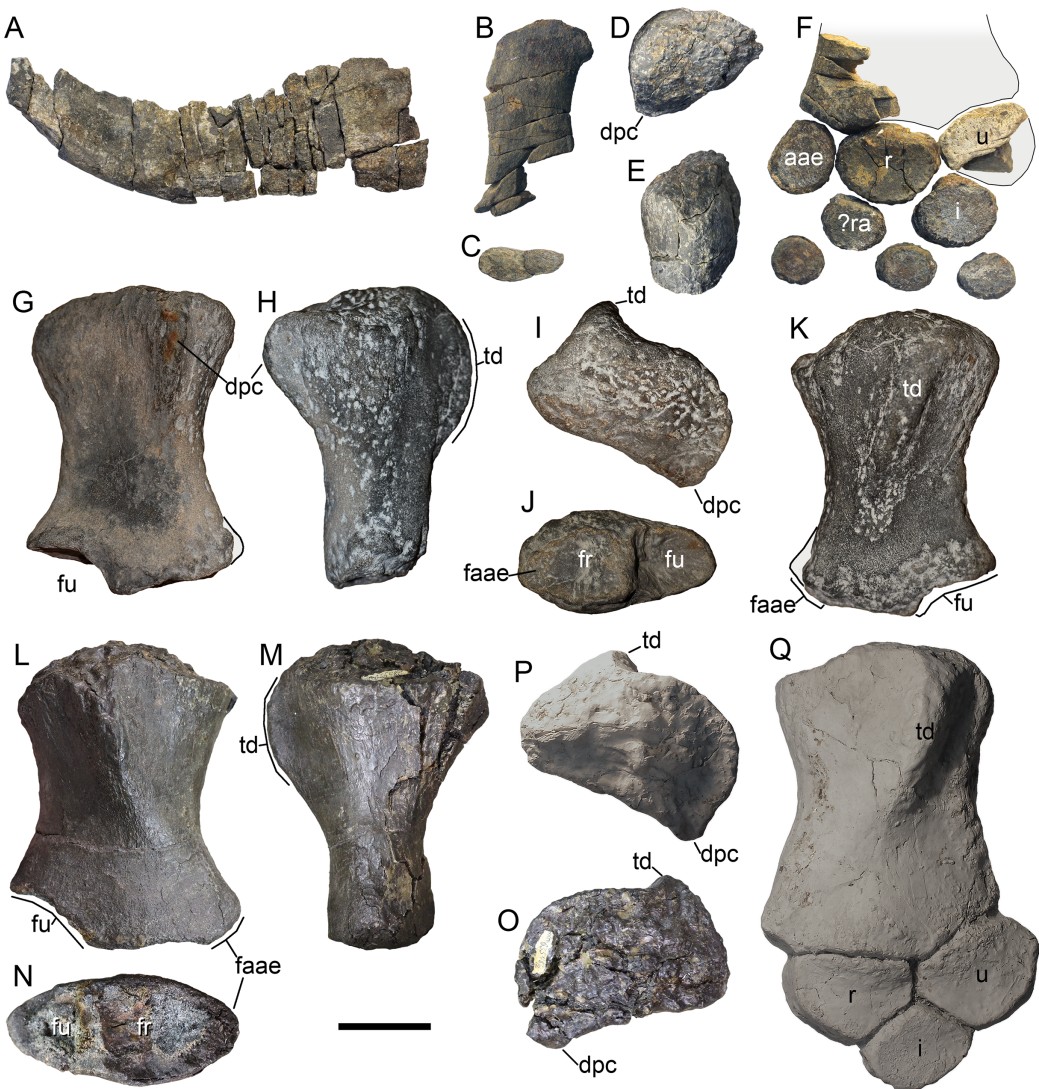

**Figure 12 Forelimb and pectoral girdle elements of *Arthropterygius hoybergeti*.** (A) Right clavicle of SVB 1451 in external view. (B, C) Dorsal ramus of the left scapula of SVB 1451 in external view (B) and cross-section (C). (D, E) Proximal portion of the right humerus of SVB 1451 in proximal (D) and anteroventral (E) views; (F) partially reconstructed forelimb of SVB 1451. (G–K) Left humerus *A.* cf. *hoybergeti* UPM 2442 in ventral (G), anterior (H), proximal (I), distal (J), and dorsal (K) views. (L–O) Right humerus *A.* cf. *hoybergeti* YKM 63548 in dorsal (L), anterior (M), distal (N), and proximal (O) views. (P, Q) A cast of the partial left forelimb of *A.* cf. *hoybergeti* YKM 63548 in proximal (P) and dorsal (Q) views. Abbreviations: aae, anterior accessory epipodial element; dpc, deltopectoral crest; faae, facet for the anterior accessory epipodial element; fr, facet for the radius; fu, facet for the ulna; i, intermedium; r, radius; ra, radiale; td, dorsal process; u, ulna; ul, ulnare. Scale bar represents five cm.

A shallow notch of the quadrate foramen restricts the posterolateral edge of the quadrate. The anterior edge of the pterygoid lamella is convex (Figs. 3J, 3K, 3O and 3Q). There is no marked angular protrusion ("antero-internal angle" of *Andrews, 1910*) on the quadrate. The articular condyle is weak and mediolaterally compressed. Its ventral surface is divided by the smooth groove into two bosses: large ventrally protruding medial boss for

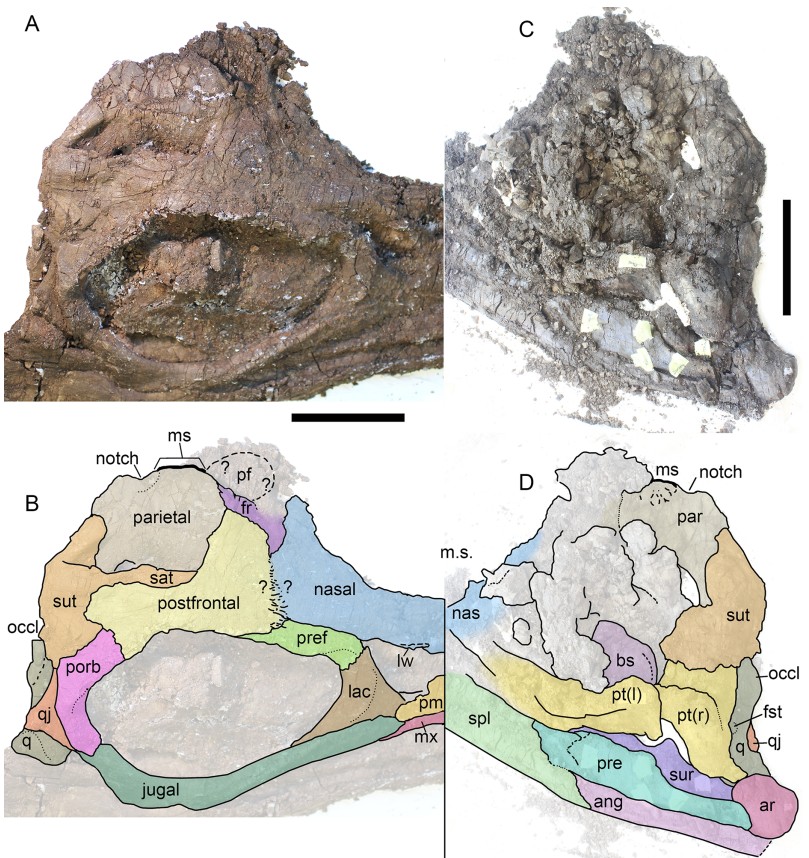

**Figure 13 Skull of *Arthropterygius lundi* PMO 222.654 in right dorsolateral view (A, B) and its inner side in posteromedial view (C, D).** Abbreviations: ang, angular; ar, articular; bs, basisphenoid; fro, frontal; fst, facet for the stapes; lac, lacrimal; lw, lateral wing of the nasal lamella; ms, medial symphysis; mx, maxilla; nas, nasal; occl, occipital lamella; pf, parietal foramen; pm, premaxilla; porb, postorbital; pre, prearticular; pref, prefrontal; pt, pterygoid; sat, supratemporal anteromedial tongue; spl, splenial; sur, surangular; sut, supratemporal; q, quadrate; qj, quadratojugal. Scale bar represents 10 cm.

the articulation with the articular and reduced anteriorly shifted lateral boss for the articulation with the surangular (Figs. 3L–3N). The ventral edge of the articular boss is somewhat V-shaped (Fig. 3J). Above the condyle, there is a pronounced circular depression—a facet for the quadratojugal (Figs. 3L, 3O and 3P). The stapedial facet, situated in the middle of the medial surface of the quadrate, is circular in outline (Figs. 3J and 3O).

**Basisphenoid.** The basisphenoid is the most peculiar element in basicranium of *Arthropterygius* due to an uncommon position of the posterior opening for the internal carotid arteries, which pierce the basisphenoid at its posterior edge. The ventral surface of the basisphenoid is trapezoid in outline (Figs. 3A and 3E). It is longer anteroposteriorly than mediolaterally wide, having the width to length ratio of 1.14–1.26 (see Table S5). The mediolateral width of the anterior part is greater than the width of the posterior part. The basipterygoid processes are relatively reduced in comparison to *Undorosaurus*, *Grendelius*, and most of platypterygiines (see *Zverkov & Efimov, 2019*).

The lateral facet of the basipterygoid processes is elongated-oval, lenticular in outline (Figs. 3D and 3G). The dorsal surface of the basisphenoid is divided into two surfaces—a squared posterodorsally faced basioccipital facet and a pentagonal dorsally faced dorsal plateau (Figs. 3B and 3F). A median groove bisects the dorsal surface over the entire length. The basioccipital facet is faced posterodorsally and occupies nearly a half of the dorsal surface in dorsal view. This condition is unique for *Arthropterygius*; in other ophthalmosaurids this facet is strongly inclined posteriorly and poorly visible in dorsal view (*Kear, 2005*; *Fischer et al., 2011*, *2012*, *2014b*; *Zverkov, Arkhangelsky & Stenshin, 2015*; *Moon & Kirton, 2016*; *Zverkov & Efimov, 2019*). The high anterior wall is vertical, slightly curving posterodorsally on its lateral sides, lining the cranioquadrate passage. The anterior wall raises the dorsum sellae in the middle, which is ventrally bounded by the funnel-like anterior foramen for the internal carotid arteries (Figs. 3C and 3H). Laterally the dorsum sellae is bounded by the ridges (crista trabeculares), which ventrally form the surfaces for their cartilaginous continuation; these surfaces are poorly pronounced in all specimens referred to *A. chrisorum* (Figs. 3C and 3H). Lateral to the crista trabeculares deep pits for attachment of the ocular musculature (likely retractor bulbi group) are situated. The posterior foramen for the internal carotid arteries opens posteroventrally in juvenile specimen CCMGE 3–16/13328, and posteriorly in mature individuals CCMGE 17–44/13328 and CMN 40608.

**Opisthotic and stapes.** The opisthotic and stapes are known only for PMO 222.669 (Fig. 4). Compared to other species of *Arthropterygius*, in *A. chrisorum* opisthotic is markedly higher dorsoventrally, and has more short and robust paraoccipital process (Figs. 4A and 4B). The medial head of the stapes is more massive than in *A. hoybergeti* and *A. lundi* and the lateral extremity of the stapedial process is more straight and somewhat dorsoventrally compressed (Figs. 4D–4F): in other species, it is dorsoventrally expanded.

**Mandible.** In general, the mandible is well described for PMO 222.669 by *Delsett et al. (2018)*. From other specimens, it is relatively well preserved only in CCMGE 17–44/13328, however, lacking anterior and posterior portions, including the whole dentary and articular (Fig. 5).

**Axial skeleton.** A continuous series of 69 vertebral centra is preserved in CCMGE 3–16/13328, only a few fragmentary, severely deformed and weathered vertebrae are collected for CCMGE 3–16/13328, and 10 vertebrae including atlas-axis complex are available for SGM 1573. These materials provide additional information to that published by Maxwell for the holotype (*Maxwell, 2010*).

The atlas-axis complex preserved in SGM 1573 is very similar to that of the holotype, however, diapophyses and parapophyses are relatively more protruding (Figs. 6A and 6C). The vertebrae of *Arthropterygius chrisorum*, in general, are similar to those of *Ophthalmosaurus icenicus* (see *Moon & Kirton, 2016*). The middle and posterior dorsal vertebrae of the large mature specimen, SGM 1573, are characterized by strongly protruding diapophyses and parapophyses (Figs. 6F–6I), whereas in the juvenile CCMGE

3–16/13328 these apophyses are less well pronounced (Figs. 6L–6S). A continuous vertebral series of CCMGE 3–16/13328 allows making some observations on vertebral count (Fig. S2). As some anteriormost presacral centra are missing it is hard to assess accurately the number of presacral vertebrae. Only 13 anterior presacral vertebrae, in which diapophyses are fused with neural arch facets, are present in CCMGE 3–16/13328. A count of posterior presacral vertebrae is 17. Six anteriormost caudal vertebrae bear characteristic eight-shaped synapophyses that commonly mark "sacral" region in ophthalmosaurids (*Moon & Kirton, 2016*). The rest preflexural caudal centra bear typical oval to circular rib facets (Figs. 6Y and AA). The shape of the articular surfaces in the caudal vertebrae is circular with the height slightly exceeding width in some anteriormost caudal vertebrae (Figs. 6V and 6X; Fig. S2); in posterior caudal vertebrae, the width markedly exceeds their height (Fig. 6Z; Fig. S2). Several fluke centra preserved in CCMGE 3–16/13328 have circular articular surfaces with nearly equal width and length.

Both mature SGM 1573 and juvenile CCMGE 3–16/13328 individuals do not have such a high degree of regionalization in posterior dorsal to anterior caudal centra, which was observed by *Maxwell (2010)*. It is possible that this condition is quite variable both in ontogeny and intraspecifically, thereby it is hard to assess its potential taxonomic value to the moment.

Numerous rib fragments were collected for CCMGE 17–44/13328. The longest but incomplete rib is near 70 cm in preserved lengths. The ribs are from T-shaped to eight-shaped in cross-section in proximal part of their length and become circular in cross-section distally.

### Appendicular skeleton

**Scapula.** The left scapula is completely preserved in CCMGE 17–44/13328 (Figs. 7J–7M). The element is robust: its proximodistal length is shorter than the coracoid anteroposterior length. It is similar to that of *Ophthalmosaurus icenicus* in general morphology (*Seeley, 1874*; *Andrews, 1910*; *Moon & Kirton, 2016*). The scapular shaft is mediolaterally flattened and elongated-oval in cross-section. The glenoid contribution is well developed and equal in length to the coracoid facet. The acromial process is massive and well-prominent; it curves ventrolaterally, forming a nearly right angle with the lateral surface of the scapula (Fig. 7N).

**Coracoid.** The coracoid of CCMGE 3–16/13328 is slightly longer anteroposteriorly than wide mediolaterally (Fig. 7P). It is similar to that of *Ophthalmosaurus icenicus* and *Undorosaurus gorodischensis* (*Andrews, 1910*; *Moon & Kirton, 2016*; *Zverkov & Efimov, 2019*), but differs in relative size, being anteroposteriorly longer than scapular proximodistal length. The medial symphysis is lenticular in outline; it occupies anterior two-thirds of the medial surface. The anteromedial process is prominent, laterally limited by an extensive anterior notch (anterior notch is relatively smaller in CCMGE 3–16/13328 than in the holotype, most likely as a reason of immaturity). The posterior portion of the coracoid is strongly compressed and convex posteriorly (Fig. 7P). The most interesting trait is that articulated coracoids form a pronounced angle of 100° (Fig. 7O); this condition is unique for *Arthropterigius*. The scapular facet and glenoid contribution are offset

by an angle of *c.* 140°. Their surfaces are slightly convex and tuberous. The glenoid contribution surface is parallel to the medial symphysis of the coracoid, thus coracoid mediolateral length is constant, unlike caudally constricting coracoids of *Sveltonectes* (*Fischer et al., 2011*), *Nannopterygius* (*Hulke, 1871*; *Kirton, 1983*), and "*Paraophthalmosaurus*" (*Arkhangelsky, 1997*; *Efimov, 1999a*) and caudally expanding coracoids of *Undorosaurus* (*Efimov, 1999b*).

**Clavicle.** The clavicle of CCMGE 3–16/13328 (Figs. 7X–7Z) is a large and robust element. It is very similar to that of *A. lundi*, being dorsoventrally high and anteroposteriorly thick, compared to other known ophthalmosaurids. On its medial surface, there is a rugose circular facet for articulation with the acromial process of the scapula (Fig. 7Y). This facet is pronounced, but not as well developed as in *A. lundi* (see below).

**Interclavicle.** The interclavicle of SGM 1573 is a large and slender T-shaped element. The anterior transverse bar of the interclavicle is straight, with a high dorsally rising wall; its lateral extremities extend far laterally, and their ends are rounded (Figs. 7CC and 7DD). There is no ventral knob observed in *Undorosaurus gorodischensis* and *Grendelius alekseevi* (*Zverkov, Arkhangelsky & Stenshin, 2015*; *Zverkov & Efimov, 2019*). The posterior median stem is slender and bears a shallow trough along its dorsal surface. There is a prominent bulge in the middle of the ventral surface of the stem (Figs. 7CC and 7DD). In PMO 222.669 a displaced portion of the clavicle was erroneously interpreted as a wide interclavicle posterior median stem (*Delsett et al., 2018*). In fact, the interclavicle of PMO 222.669 is heavily distorted and broken into several disarticulated pieces due to a collapsing of pectoral girdle during the taphonomic process, but judging from the preserved fragments, its posterior median stem was quite slender.

**Humerus.** Humeri are known for all the referred specimens. The humerus is a large and robust bone with wide and dorsoventrally compressed midshaft. The humeral "torsion" (angle between the long axes of the proximal and distal ends of the humerus) is *c.* 70°. The dorsal process is prominent and plate-like, extending up to the half of the humeral midshaft (Figs. 7C, 7F and 7S). The deltopectoral crest is poorly developed and shifted to the anterior border of the humerus (Figs. 7A, 7E, 7G, 7I, 7T and 7W). The proximal end is semi-rectangular in outline, being anteroposteriorly longer than dorsoventrally thick (Figs. 7E, 7I and 7W). There are three distal concave facets for the preaxial accessory element, radius, and ulna. The facet for the preaxial accessory element is large and semicircular in outline; it occupies nearly equal space as the radial facet. The radial facet is irregularly pentagonal in outline; its ventral edge is angular, forming in posterior half an abrupt skew to the ulnar facet (Figs. 7D, 7H and 7V). A ratio of the dorsoventral width of the radial facet to ulnar facet is 0.7–0.8 (see Table S7). Among ophthalmosaurids, the pronounced constriction between the ulnar and radial facet with a ventral skew is also known for "*Macropterygius*" and *Sisteronia*, which at the same time lack a well-developed facet for an anterior accessory epipodial element characteristic of *Arthropterygius* (*Fischer et al., 2014b*; *Moon & Kirton, 2018*).

**Epipodial elements.** The articular surfaces of the epipodial elements are convex for a peg-and-socket articulation with the concave distal humeral facets; however, this condition varies even in mature specimens from extremely deep in CMN 40608 to more shallow in SGM 1573. The anterior accessory epipodial element is circular in dorsal view; its anterior edge lacks perichondral ossification as in *Ophthalmosaurus icenicus* (*Andrews, 1910*; *Moon & Kirton, 2016*). This element rapidly tapers anteriorly. The radius is pentagonal in dorsal and ventral views (Figs. 6A and 6F). The ulna is the largest element in the epipodial row, its dorsal and ventral cortical parts are roughly hexagonal in outline. The element gradually constricts in dorsoventral width posteriorly. A perichondral ossification of the posterior edge of the ulna is absent (Fig. 6A). The intermedium wedges between the radius and ulna, but not reach the humerus, however, a distance between the humerus and intermedium varies from relatively short in CCMGE 3–16/13328 and CMN 40608 to relatively long in CCMGE 17–44/13328. Distally intermedium bears two slightly demarcated facets for distal carpals three and four, indicating a "latipinnate" forefin architecture. Maxwell described the distal margin of the intermedium of CMN 40608 as "gently curved" but indicated that the distal edge of the intermedium forms a surface for the articulation of a single distal carpal' (*Maxwell, 2010*: 410), considering this uncertainty and the new data on other referred specimens (CCMGE 3–16/13328, CCMGE 17–44/13328, PMO 222.669), it is more likely to interpret the presence of the two poorly demarcated facets for distal carpals three and four in the holotype (CMN 40608) rather than a single convex facet.

**Distal limb elements.** All the mesopodial and autopodial elements in CCMGE 17–44/13328 and PMO 222.669 are strongly dorsoventrally thickened, circular in outline and loosely packed, indicating a large amount of cartilage in the forefin, which is most similar to the condition observed in *Ophthalmosaurus icenicus* (*Andrews, 1910*; *Moon & Kirton, 2016*). One of the elements in CCMGE 17–44/13328 has a semicircular outline in dorsal view and bears a perichondral ossification along one of its edges, this probably represents a pisiform (Fig. 6A). The pisiform of exact same morphology is present in the left limb of PMO 222.669 (Nikolay G. Zverkov, 2018, personal observation).

**Pelvic girdle.** The only central portion of the ischiopubis has been collected for CCMGE 17–44/13328, which complicates the description of the element. The ischiopubis is plate-like, mediolaterally compressed (eight mm at its thickest part). The obturator foramen is likely absent (Fig. 8G).

**Femur.** The femur of CCMGE 17–44/13328 is slender with proximal and distal ends only slightly expanded (Fig. 8A). Its proximodistal length comprises 0.74 of the humeral proximodistal length (0.67 in the holotype CMN 40608). The femur of CCMGE 17–44/13328 is very similar to that of the holotype, possessing flattened ventral process terminating proximal to the mid-point, and thereby being more prominent than that of *A. lundi* (*Roberts et al., 2014*). The dorsal process is less pronounced than the ventral process and shifted to the anterior edge of the femur. There are two distal facets, which are concave and poorly demarcated, forming a common distal groove for the epipodial

elements (Fig. 8D). The fibular facet is slightly inclined posterodistally, whereas the tibial facet faces nearly distally.

**Measurements:** See Tables S1 and S2.

***Arthropterygius hoybergeti*** (*Druckenmiller et al., 2012*) comb. nov.
(Figs. 9–12)
v*2012 *Palvennia hoybergeti Druckenmiller, Hurum, Knutsen & Narkem*: 326, figs. 12–21

**Holotype and only referred specimen:** SVB 1451, see Table 1.

**Emended diagnosis**. A moderately large ophthalmosaurid (up to four m) distinguished from other species of *Arthropterygius* by the following unique character combination: basisphenoid longer anteroposteriorly than mediolaterally wide, with the widest part in the region of basipterygoid processes (unlike in *A. volgensis*); posterior foramen for internal carotid arteries opening on the posteroventral edge of the basisphenoid and forming a notch as in *A. lundi* (referred specimen SGM 1502) and unlike *A. chrisorum*; small basioccipital facet of the opisthotic (large in other known species of *Arthropterygius*); relatively large teeth with circular in cross-section roots and robust ridged crowns as in *A. chrisorum* but unlike gracile subtly ridged crowns of *A. lundi*; well developed deltopectoral crest (unlike in other species of *Arthropterygius*); anterodistal facet for the anterior accessory epipodial element sufficiently smaller than the radial facet, being thus relatively smaller than that in *A. lundi* and *A. chrisorum*.

**Occurrence:** *Arthropterygius hoybergeti* is known from the Slottsmøya Member of the Agardhfjellet Formation of Svalbard (type locality), where it was found most likely within the *Dorsoplanites ilovaiskii* Ammonite Biozone (lower Middle Volgian).
Two specimens from the Volga Region (both found on the right bank of the Volga River near Gorodischi Village, Ulyanovsk Region) referred here to as *A.* cf. *hoybergeti* are corresponding to *Dorsoplanites panderi* Ammonite Biozone of Promza Formation.

## DESCRIPTION

Here, we provide some new observations on the holotype SVB 1451, which had been described in detail by *Druckenmiller et al. (2012)* and additionally by *Delsett et al. (2018)*; thereby we provide only some additional information, not reported before.

**Nasal.** The nasal of SVB 1451 bears a well-pronounced lamella, a "wing," overhanging the dorsal border of the naris.

**Parietal.** The parietal has a very short but robust medial symphysis and well-pronounced notch posterior to it (Figs. 9A and 9B). The element posseses a relatively elongated and slender supratemporal process (Figs. 9A and 9B).

**Squamosal.** Although reported as absent, the squamosal of SVB 1451 (Figs. 9C and 9D) was mentioned by *Druckenmiller et al. (2012)* as a "*small rib-like element*" of unclear identity, and even figured (*Druckenmiller et al., 2012*: 327, fig. 16E, F).
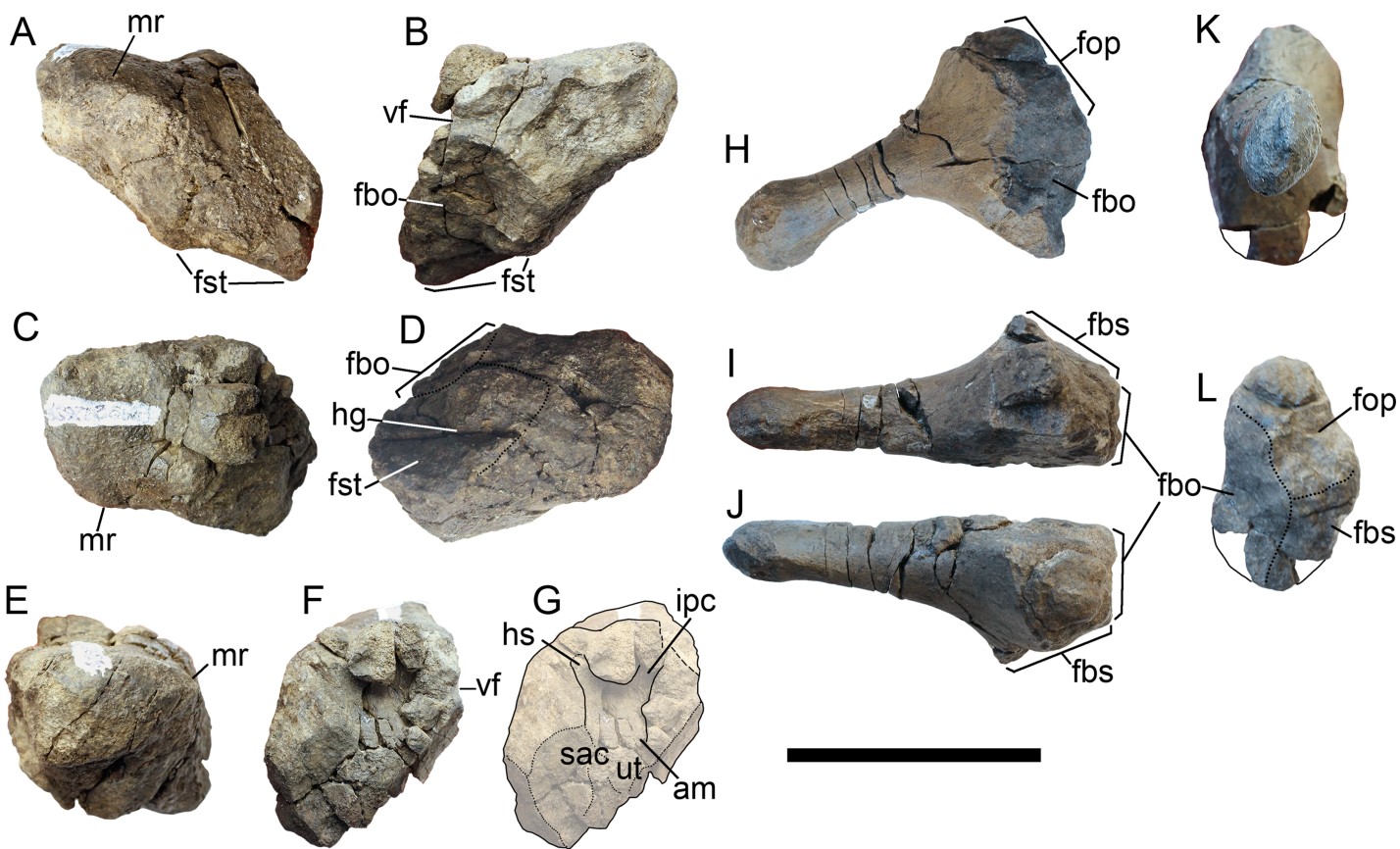

**Figure 14 Opisthotic and stapes of *Arthropterygius lundi* PMO 222.654.** (A–G) Opisthotic in anterior (A), posterior (B), dorsal (C), ventral (D), lateral (E), and medial (F, G) views. (H–L) Left stapes in, posterolateral (H), ventral (I), dorsal (J), distal (K), and medial (L) views. Abbreviations: am, ampulla; fbo, facet for the basioccipital; fbs, facet for the basisphenoid; fop, facet for the opisthotic; fst, facet for the stapes; hg, groove for transmission of hyomandibular branch of facial (VII) or glossopharyngeal (XI) nerve; hc, impression of horizontal semicircular canal; ipc, impression of posterior vertical semicircular canal; mr, muscular ridge on the opisthotic; sac, sacculus; ut, utriculus; vf, vagus foramen. Scale bar represents five cm.

**Quadrate.** Both quadrates of SVB 1451 are preserved, but only partially exposed, so that dorsal portion of the right quadrate and ventral portion of the left quadrate are available for observations. The occipital lamella of the quadrate is extremely well developed (Fig. 10O). The articular condyle is relatively weak; the articular boss is larger than the surangular boss and protrudes ventrally. There is a pronounced angular protrusion of the quadrate (absent in *A. chrisorum*).

**Basisphenoid.** The basisphenoid of SVB 1451 could be observed in ventral and dorsal views (Figs. 9A, 9B and 10B). It is longer anteroposteriorly than mediolaterally wide. The widest part is the region of basipterygoid processes that are directed anterolaterally. A posterior foramen for the internal carotid arteries opens on the posteroventral edge of the basisphenoid and forms a notch as in *A. lundi* and unlike *A. chrisorum* (Fig. 10B).

**Opisthotic.** The opisthotic was not described for SVB 1451 by *Druckenmiller et al. (2012)*, neither by *Delsett et al. (2018)*, however, both opisthotics are well-preserved.

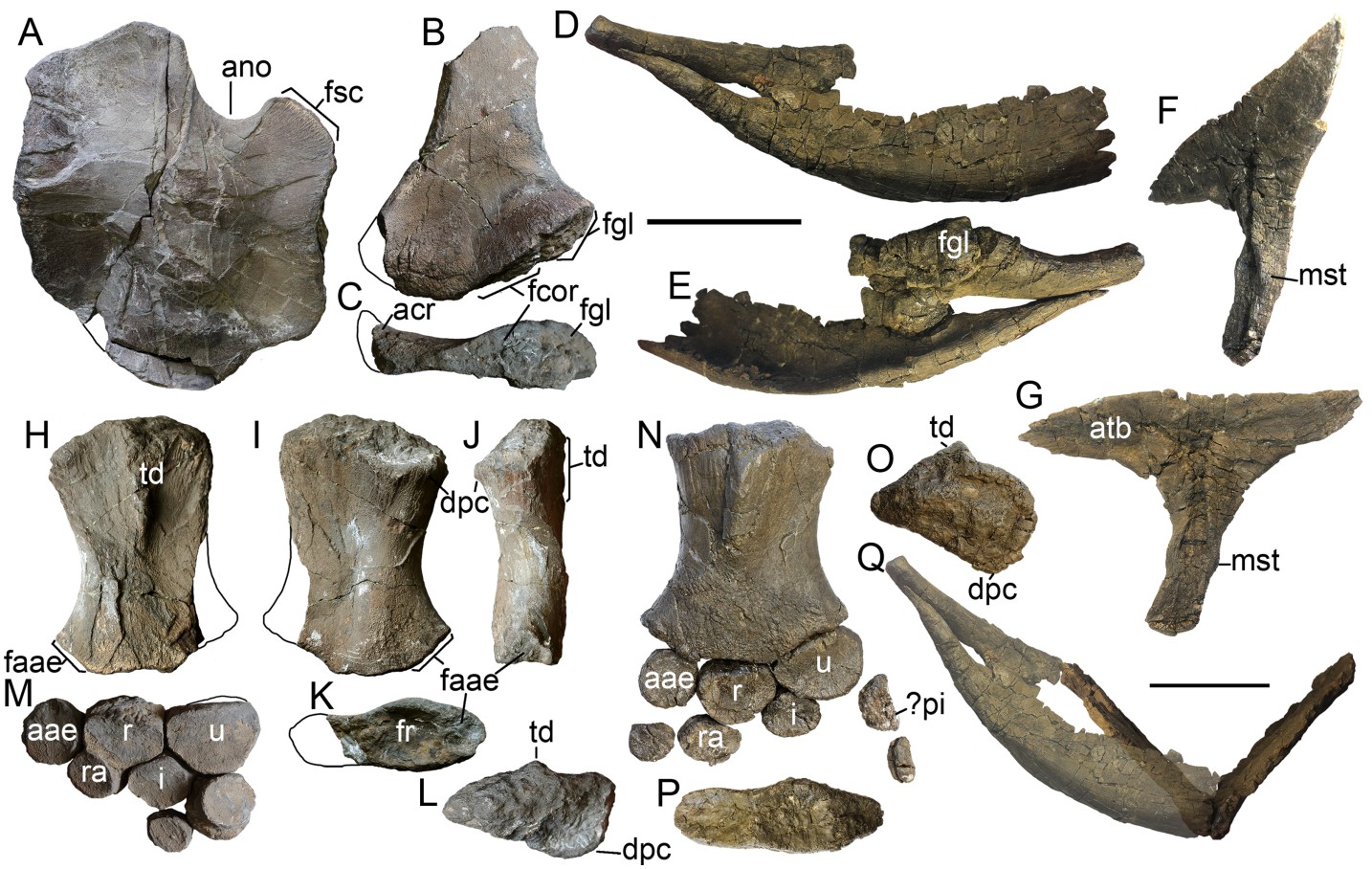

**Figure 15 Forelimb and pectoral girdle elements of *Arthropterygius lundi*.** (A) Left coracoid of SGM 1731-01–15 in ventral view. (B, C) Left scapula of SGM 1731-01–15 in lateral (B) and proximal (C) views. (D, E) Articulated right clavicle and scapula of PMO 222.654 in anterior (D) and posteromedial (E) views. (F, G) Interclavicle of PMO 222.654 in oblique posterolateral (F) and dorsal (G) views. (H–L) Left humerus of SGM 1731-01–15 in dorsal (H), ventral (I), anterior (J), distal (K), and proximal (L) views. (M) Articulated epipodial and autopodial elements of the left forelimb of SGM 1731-01–15. (N) Left forelimb of PMO 222.654 in dorsal view. (O, P) Left humerus of PMO 222.654 in proximal (O) and distal (P) views. (Q) Partially reconstructed pectoral girdle of PMO 222.654. Abbreviations: aae, anterior accessory epipodial element; acr, acromial process; atb, anterior transverse bar of the interclavicle; dpc, deltopectoral crest; faae, facet for the anterior accessory epipodial element; fcor, facet for the coracoid; fgl, glenoid contribution; fr, facet for the radius; fsc, facet for the scapula; fu, facet for the ulna; i, intermedium; mst, bulge in the middle of the interclavicle posterior median stem; pi, pisiform; r, radius; ra, radiale; td, dorsal process; u, ulna. Scale bars represent 10 cm.

The paraoccipital process of the opisthotic is short and robust, which is a common condition for ophthalmosaurids except for *Ophthalmosaurus* and *Acamptonectes* (*Fischer et al., 2012*). The facet for the supratemporal is oval in outline (Fig. 10I), being strongly dorsoventrally compressed unlike that dorsoventrally high in *Arthropterygius chrisorum* and *A. volgensis*. The lateral muscular ridge is well developed (Figs. 10D and 10I). The stapedial facet is somewhat triangular in outline and bisected by a straight mediolateral canal for either VII or for IX nerve, as was interpreted by *Kirton (1983)* (see also *Kear, 2005*; *Moon & Kirton, 2016*). The facet for the basioccipital is relatively small and quadrant in outline with convex margin directed dorsolaterally, it is sufficiently smaller than the stapedial facet (Fig. 10C). The impression of semicircular

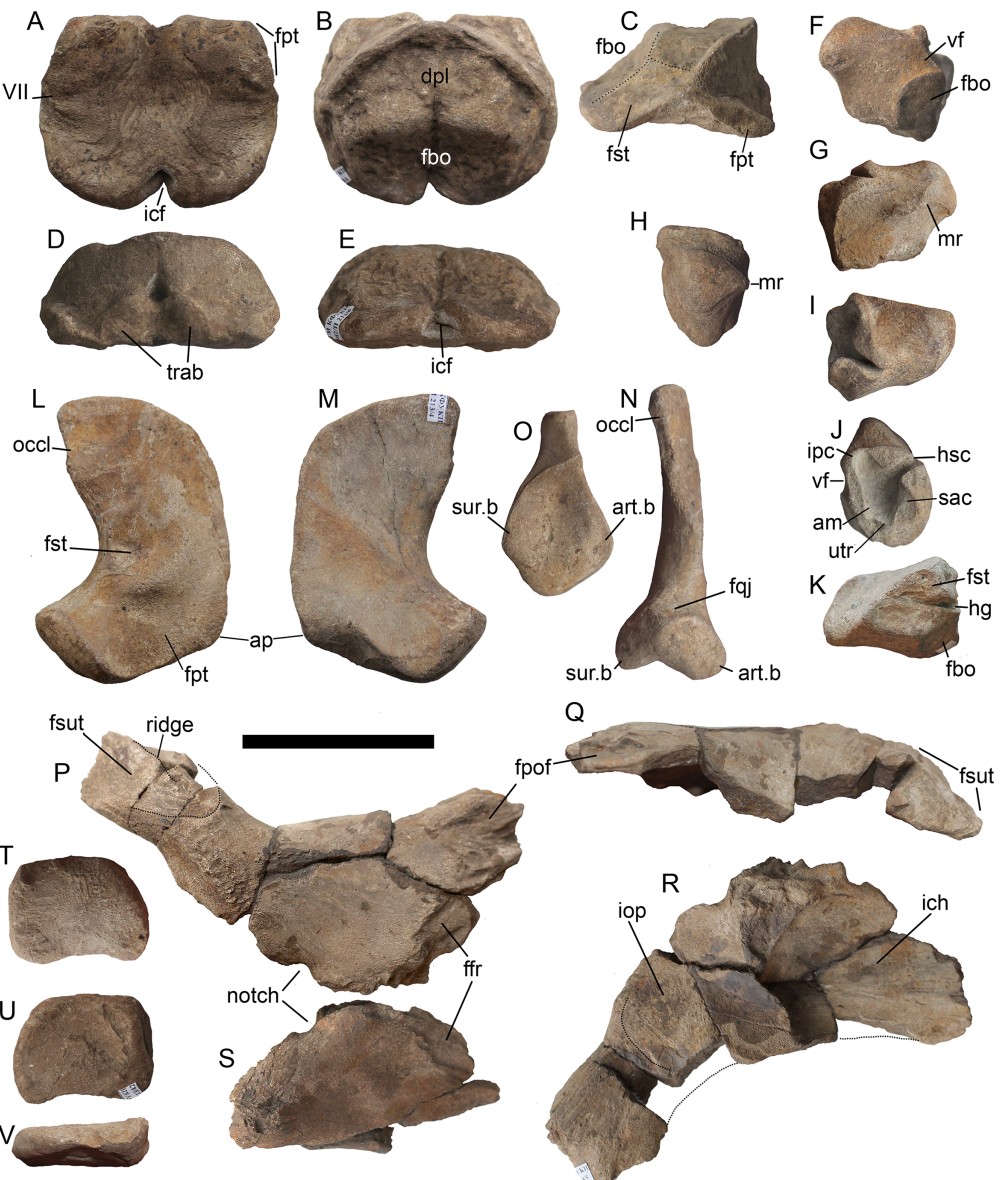

**Figure 16 Cranial elements of *Arthropterygius volgensis* KSU 982/P-213.** (A–E) Basisphenoid in ventral (A), dorsal (B), lateral (C), anterior (D), and posterior (E) views. (F, G, I, J) Left opisthotic in posterior (F), anterior (G), dorsal (I), and medial (J) views. (H, K) Right opisthotic in lateral (H) and ventral (K) views. (L–N) Left quadrate in posteromedial (L), anterolateral (M), and posterolateral (N) views. (O) Ventral view of the right quadrate. (P–R) Left parietal in dorsal (P), lateral (Q), and ventral (R) views; (S) partial right parietal in dorsal view. (T–V) Right articular in medial (T), lateral (U), and dorsal (V) views. Abbreviations: am, ampulla; art.b, articular boss; dpl, dorsal plateau of the basisphenoid; fbo, facet for the basioccipital; ffr, facet for the frontal; fpof, facet for the postfrontal; fpt, facet for the pterygoid; fqj, facet for the quadratojugal; fst, facet for the stapes; hg, groove for transmission of hyomandibular branch of facial (VII) or glossopharyngeal (XI) nerve; hsc, impression of horizontal semicircular canal; icf, foramen for the internal carotid arteries; ich, impression of the cerebral hemisphere; iop, impression of the optic lobe; ipc, impression of posterior vertical semicircular canal; fsut, facet for the supratemporal; mr, muscular ridge on the opisthotic; occl, occipital lamella; sac, sacculus; sur.b, surangular boss; trab, facets for cartilaginous continuation of the *cristae trabeculares*; ut, utriculus; vf, vagus foramen; VII, groove of the palatine ramus of facial (VII) nerve. Scale bar represents five cm.

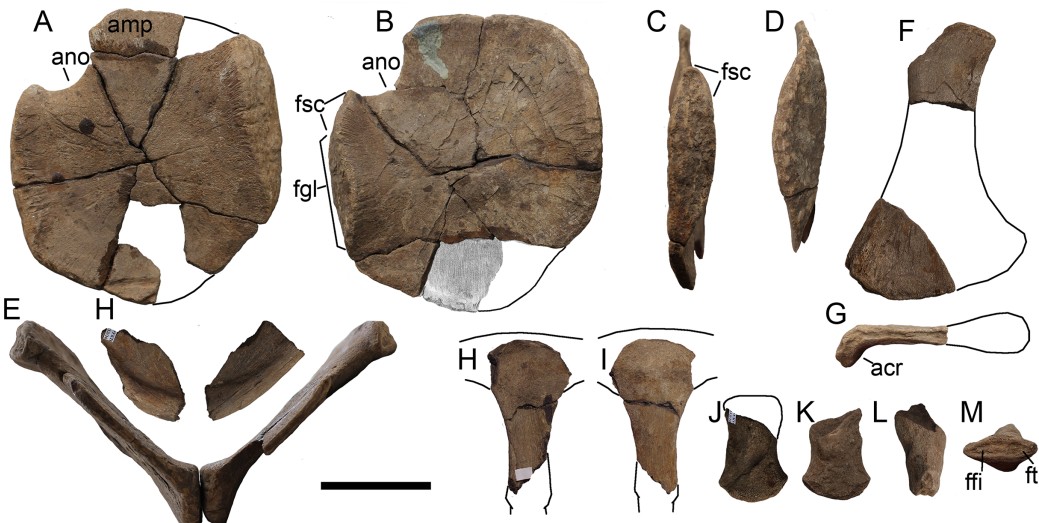

**Figure 17 Pectoral girdle elements and femur of *Arthropterygius volgensis* KSU 982/P-213.** (A) Left coracoid in dorsal view. (B–D) Right coracoid in ventral (B), dorsolateral (C), and ventromedial (D) views. (E) Articulated coracoids in anterior view. (H) Fragmentary clavicles. (F, G) Fragmentary right scapula in mediall (F) and proximal (G) views. (H, I) Interclavicle in dorsal (H) and ventral (I) views. (J–M) Right femur in ventral (J), dorsal (K), anterior (L), and distal (M) views. A portion of the right coracoid, that is, currently missing (B) is modified from *Kasansky (1903*, Tab. II, fig. 6*)*. Abbreviations: acr, acromial process; amp, anteromedial process of the coracoid; ano, anterior notch; ffi, facet for the fibula; fgl, glenoid contribution; fsc, facet for the scapula; fti, facet for the tibia. Scale bar represents five cm.

canals of the otic capsule is V-shaped (Figs. 10G and 10H). Both impressions of the horizontal semicircular canal and posterior vertical semicircular canal are nearly equal in length, unlike in *Undorosaurus gorodischensis* and *Acamptonectes densus*, in which horizontal semicircular canal impression is markedly longer (*Fischer et al., 2012*; *Zverkov & Efimov, 2019*). The impression housing the posterior ampulla, utriculus, and the sacculus is expanded (Fig. 10G).

**Exoccipital.** Both exoccipitals are preserved in SVB 1451, however, right element was identified as left and figured upside down in the original description (*Druckenmiller et al., 2012*). The statement that "there is no evidence of any foramina for cranial nerves perforating the element" (*Druckenmiller et al., 2012*: 331) is likely a result of the state of preservation, as was also suggested by *Delsett et al. (2018*: 23*)*. At least one hypoglossal foramen could be seen on the lateral side of the left exoccipital, although, indeed, columnar morphology with the reduced base of the occipital foot make the reduction of a number of hypoglossal foramina expected.

**Stapes.** The hyoid process of the stapes is relatively well developed and helps for correct spatial orientation of the element (Fig. 10L). The basisphenoid and basioccipital facets are clearly demarcated; dorsal to them there is an extensive facet for the opisthotic (Fig. 10J). Given that the stapedial facet of the basioccipital is directed anteriorly, and that there is some extent of stapedial curvature, the stapes, when articulated, was strongly rotated anteroventrally (Fig. 10A). This condition is very unusual for ophthalmosaurids

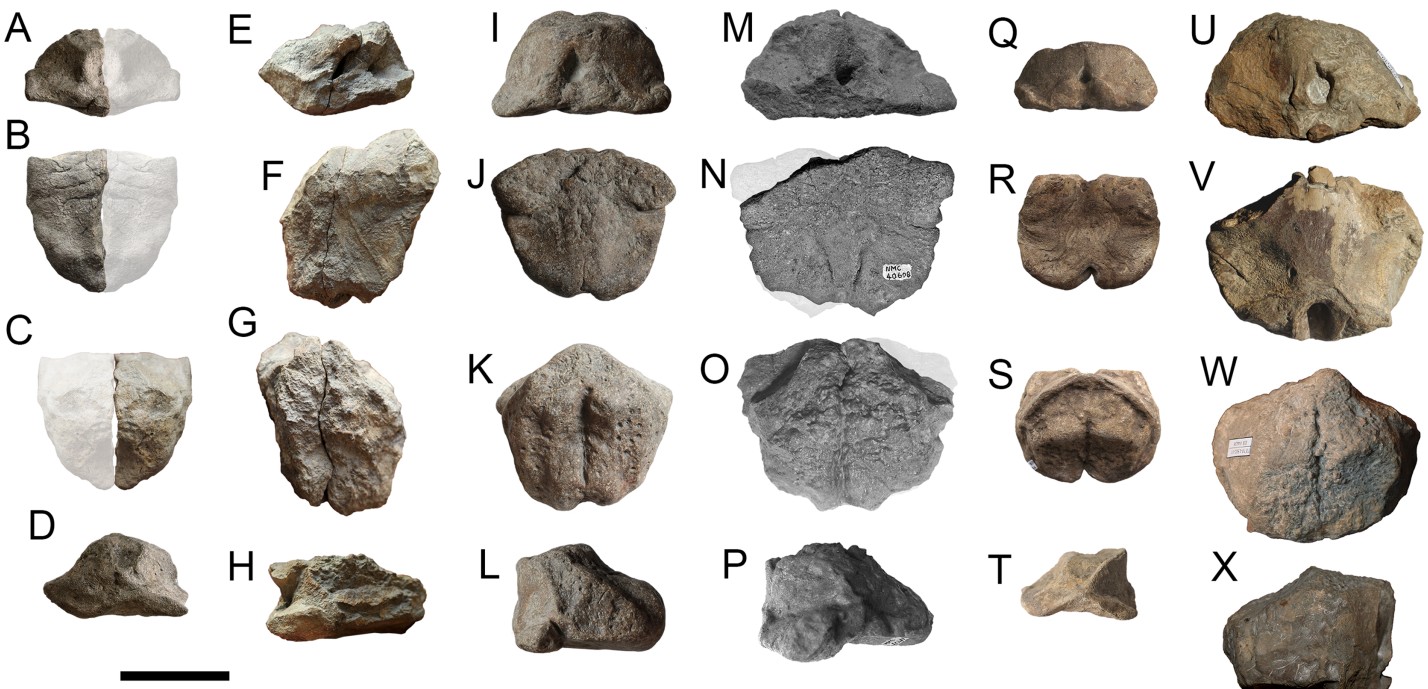

**Figure 18 Comparison of basisphenoids of Arthropterygius.** (A–D) juvenile of *A. chrisorum* CCMGE 3–16/13328. (E–H) young adult of *A. chrisorum* CCMGE 17–44/13328 (take into consideration strong deformation of this specimen). (I–L) young adult of *Arthropterygius* cf. *A. chrisorum* SGM 1743-2 (basipterygoid processes are slightly eroded). (M–P) mature individual of *A. chrisorum* CMN 40608. (Q–T) juvenile of *A. volgensis* KSU 982/P-213. (U–X) mature individual of *A. lundi* SGM 1502. Respective views are in rows from the top to down: anterior view; ventral view; dorsal view; lateral view. N and O are modified from (*Maxwell, 2010*, fig. 2), M and P are provided by E. Maxwell and J. Mallon (personal communication, 2015). Scale bar represents five cm.

but probably was typical for ichthyosaurs of *Arthropterygius* clade, as all of them have anteriorly directed stapedial facet of the basioccipital. The configuration of the articulated occipital region of *A. hoybergeti* was strongly protruding posterodorsally, somewhat "vaulted," which is probably a result of a strong reduction of the postorbital region.

**Dentition.** The teeth of *A. hoybergeti* are relatively large. The crowns are robust, conical, ranging from straight to slightly recurved. The enamel ornamentation is composed of numerous tightly packed ridges, which are semicircular in cross-section (Fig. 11A). The ridges seem to extend to the apex of the crown and arranged around its entire circumference. The apicobasal length of the largest crown is *c.* 14 mm in apicobasal length and nine mm in diameter at the base.

**Vertebral column.** No line of fusion of atlas and axis could be observed and is unlikely to present, (cf. *Druckenmiller et al., 2012*: 334). An incomplete anterior presacral ("cervical") centrum is preserved and has characteristic oval outline slightly tapering ventrally (Fig. S5).

**Clavicle.** The right clavicle is nearly complete but badly preserved (12A), it is very robust and similar to those of *A. chrisorum* and *A. lundi*, thus typical of the genus.

**Scapula.** The preserved scapular dorsal rami are slightly curved and mediolaterally compressed having an oval cross-section of the shaft (Figs. 12B and 12C).

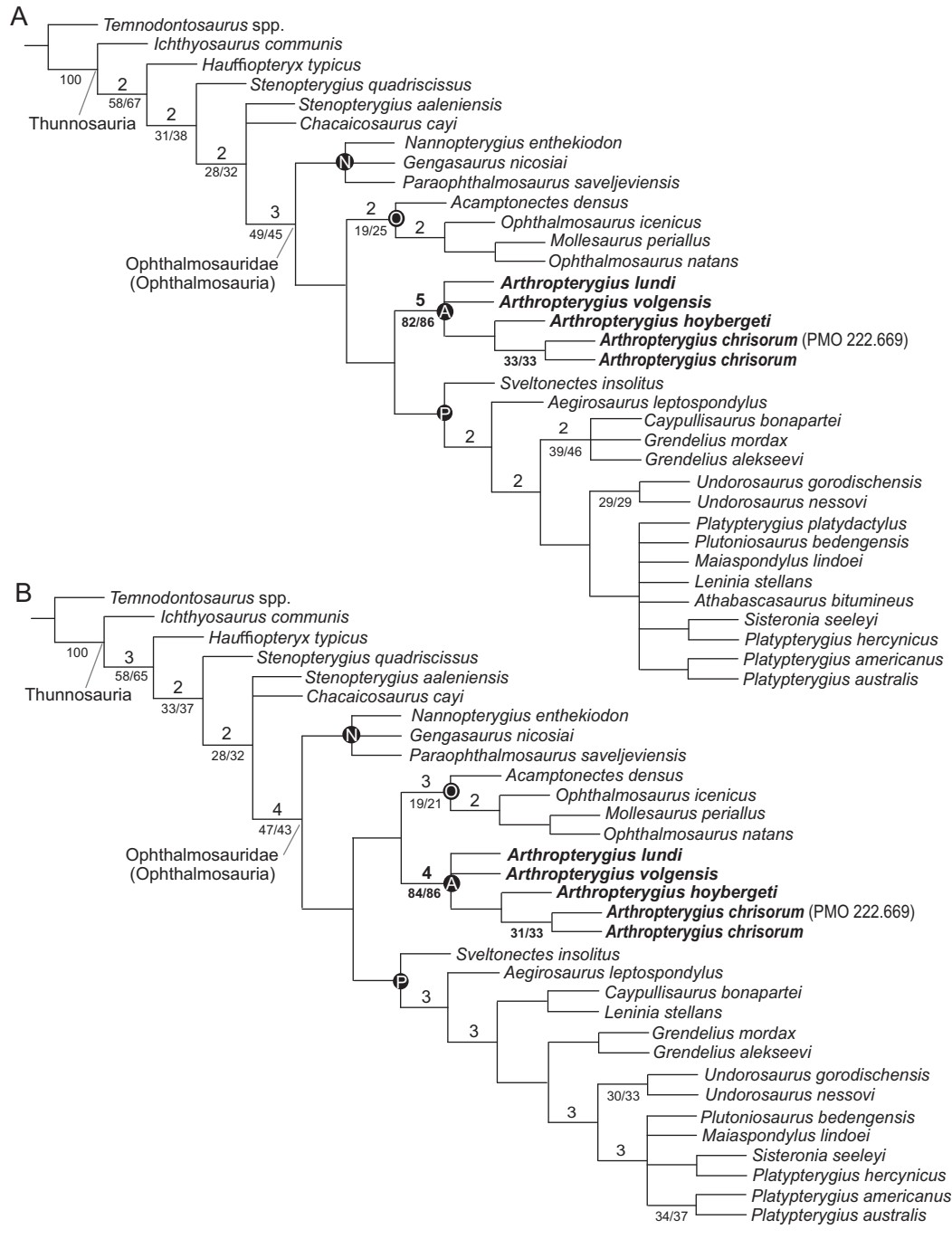

**Figure 19 Phylogenetic position of *Arthropterygius*.** (A) Strict consensus recovered from the analysis of the full dataset. (B) Strict consensus recovered from the analysis of the reduced dataset. Bremer support values >1 are shown above the branches; bootstrap/jackknife support values of greater than 20 are indicated below the branches. Abbreviations: A, *Arthropterygius* clade; N, *Nannopterygius* clade; O, Ophthalmosaurinae; P, Platypterygiinae.

**Humerus.** A number of fragments of the humerus are preserved. Most of them are from the right humerus, however, some could belong to the left humerus. Most important are proximal and distal portions. The shape of the preserved proximal portion of the right

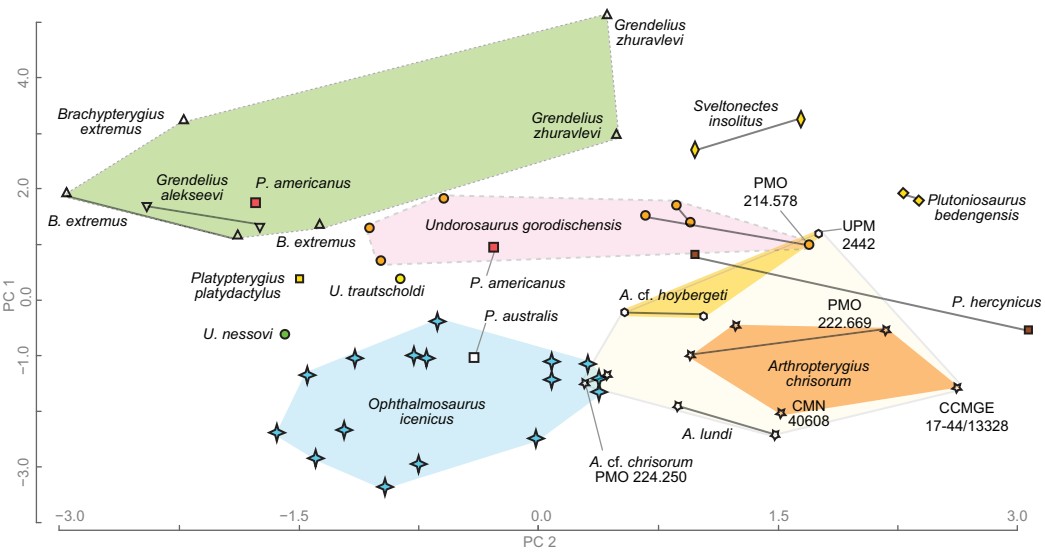

**Figure 20 Results of principal component analysis of ophthalmosaurid humeral morphology.** Humeri belonging to the same individual are connected by the solid line.

humerus indicates that it was anteroposteriorly elongate and has a pronounced plate-like deltopectoral crest (Figs. 12D and 12E). The anterodistal fragment of the humerus demonstrates that it was dorsoventrally thick distally. A facet for the anterior accessory element is relatively small and triangular in outline (Fig. 12F; Fig. S7A).

**Epipodial and autopodial elements.** Several epipodial and autopodial elements are preserved in SVB 1451, including the complete anterior accessory epipodial element, radius, and intermedium as well as fragmental ulna (Fig. 12F). While not included in the original description (*Druckenmiller et al., 2012*), the elements were recently mentioned and figured by *Delsett et al. (2018)*, however, we propose some different identifications: we interpret as an anterior accessory epipodial element the elements identified as a pisiform by *Delsett et al. (2018)*; we can not find arguments for identification of the distal carpal 3, and we interpret as the radiale, although uncertainly, the element that was considered to be a metacarpal. The anterior accessory epipodial element present in SVB 1451 is semicircular in dorsal view, it strongly tapers along the anterior margin, which is nearly straight, but still not involved in perichondral ossification (Fig. 12F). This morphology is most similar to other known anterior accessory epipodial elements available in the examined *Arthropterygius* material (in: PMO 222.669, PMO 222.654; CCMGE 17–44/13328; SGM 1731-01–15). The pisiform is known for PMO 222.669 that was originally referred to as *Palvennia hoybergeti* (*Delsett et al., 2018*) but referred to as *Arthropterygius chrisorum* herein (see above); this is a slender lunate element with its posterior edge involved in perichondral ossification, markedly dissimilar to the morphology seen in the element described above. In this regard, it is hardly possible to identify the discussed element of SVB 1451 as a pisiform. The radius is typically pentagonal in dorsal view and has a strongly convex proximal articular surface. The intermedium is somewhat diamond-shaped in dorsal

**Figure 21 Overlapping skeletal elements of *Arthropterygius chrisorum*, "*Keilhauia nui*," "*Palvennia*" *hoybergeti*, "*Janusaurus*" *lundi*, and "*Ichthyosaurus*" *volgensis* compared to those of other well-known contemporary.** Reconstructed regions are shown in white with dotted boundaries. White elements in the column of *Grendelius alekseevi* (parietal, jugal, articular) are based on observations on other congeneric taxa *G. mordax* and *G. pseudoscythicus*. Scale bars for cranial elements = five cm, for postcranial elements = 10 cm.

view, wedging between the radius and ulna and bearing two distal facets, evidently for distal carpals three and four (Fig. 12F). The autopodial elements are circular in outline and were loosely arranged in the limb as in *Ophthalmosaurus icenicus* (see *Moon & Kirton, 2016*) and other species of *Arthropterygius*.

### *Arthropterygius* cf. *hoybergeti*

(Figs. 12G–12Q)

**Referred specimens:** UPM 2442 and YKM 63548, see Table 1.

The specimens referred herein to as *A.* cf. *hoybergeti* lack cranial remains, whereas the holotype of *A. hoybergeti* lacks most of the postcranium resulting in poor overlap between these specimens. This could call into question our decision to refer UPM 2442 and YKM 63548 to as *A.* cf. *hoybergeti*, however, we suggest that this is a reasonable assumption. Despite the minute difference in size, the humeri of UPM 2442 and YKM 63548 are very similar one to another and bear diagnostic features of *Arthropterygius*: three concave distal articular facets for the preaxial accessory element, radius, and ulna; ulnar facet: radial facet dorsoventral width ratio *c.* 0.8; dorsoventrally compressed posterior edge of the humerus (Figs. 12G–12Q). Furthermore, YKM 63548 preserves epipodial elements and intermedium that are greatly consistent with those of other *Arthropterygius* species: ulna is larger than radius and lacks the posterior perichondral ossification; intermedium bears two nearly equal distal facets (Fig. 12Q). At the same time, the described humeri are distinct from humeri of *A. chrisorum* and *A. lundi* in absence of pronounced ventral skew between the radial and ulnar facet and in relatively small size of the facet for the anterior accessory epipodial element. Thus, based on these observation, UPM 2442 and YKM 63548 belong to *Arthropterygius*, but represent a species different from *A. chrisorum* and *A. lundi*. Although the humerus of *A. hoybergeti* is fragmented it also demonstrates relatively small and triangular in outline facet for anterior accessory epipodial element and well-developed plate-like deltopectoral crest, not characteristic for other species of *Arthropterygius*, hence we consider UPM 2442 and YKM 63548 as *A.* cf. *hoybergeti*.

Among other morphological characteristics of the discussed humeri are well developed and plate-like deltopectoral crest and dorsal process (Figs. 12H, 12I, 12K–12M and 12Q). The radial facet is the thickest part of the distal humerus, which gradually flattens posteriorly to more elongated ulnar facet (Figs. 12J and 12N). There is no marked ventral skew between the radial and ulnar facets compared to that in *A. chrisorum* and *A. lundi*, however, the decrease in the dorsoventral thickness between the radial and ulnar facets is apparent (Fig. 12J).

### *Arthropterygius lundi* (*Roberts et al., 2014*) comb. nov.

(Figs. 11B, 11C and 13–15)

v*2014 *Janusaurus lundi Roberts et al.*: 4, figs. 3–14.

v.2015 *Arthropterygius sp.*, *Zverkov, Arkhangelsky, Pardo Pérez, Beznosov*: 84, figs. 3–7.

2016 *Janusaurus lundi* Roberts et al.; *Delsett et al.*: figs. 6B, 9 and 10B–10D.

*2017 Janusaurus lundi* Roberts et al.; *Delsett et al.*: figs. 12J and 12K.

**Holotype:** PMO 222.654, an incomplete skeleton (for details see *Roberts et al., 2014*).

**Referred specimens**: SGM 1502 and SGM 1731-01–15.

**Emended diagnosis:** A medium sized ophthalmosaurid (three to four m long) diagnosed relative to other species of *Arthropterygius* by the following unique characters (including autapomorphies, marked with "*") and character combination: extremely gracile and constricted stapedial shaft*; basisphenoid trapezoid in ventral view with widest part in the region of basipterygid processes (unknown for the holotype and based on SGM 1502); posterior foramen for internal carotid arteries opening on the posteroventral edge of the basisphenoid and forming a notch as in *A. hoybergeti* and unlike *A. chrisorum* (based on SGM 1502); large basioccipital facet on the opisthotic (reduced in *A. hoybergeti*); small teeth with gracile crowns and poorly pronounced ridges (relatively large teeth with ridged crowns in *A. chrisorum* and *A. hoybergeti*, although teeth are unknown for *A. volgensis*); interclavicle with pointed lateral extremities (blunt in *A. chrisorum*) and deep trough on the dorsal surface of posterior median stem*; isometric proximal end of the humerus with nearly equal dorsoventral and anteroposterior length (as in *Ophthalmosaurus icenicus, Undorosaurus nessovi*, and *U. trautscholdi*); reduced dorsal process and deltopectoral crest; strongly dorsoventally flattened posterior and distal parts of the humerus; anterodistal facet for the anterior accessory epipodial element nearly as long, but not as wide as the radial facet, being thus relatively smaller than that of *A. chrisorum* (this facet is sufficiently smaller in *A. hoybergeti*), ventral skew between the radial and ulnar facets is pronounced in a lesser degree than in the type species, but stronger than in specimens referred to as *A.* cf. *hoybergeti*; strongly expanded dorsal portion of the ilium with distinct anterodorsal process (although this could be a generic feature).

**Occurrence:** *Arthropterygius lundi* is found in the European Russia and Svalbard. Everywhere it is found in the early Middle Volgian: Slottsmøya Member of the Agardhfjellet Formation in Svalbard (type locality), *Pavlovia rugosa* to *Dorsoplanites ilovaiskii* ammonite biozones; Paromes Formation in Timan-Pechora Basin and Promza Formation of the Volga Region, all these finds correspond to *Dorsoplanites panderi* Ammonite Biozone.

## DESCRIPTION

Here, we provide some new observations on the holotype (PMO 222.654) and description of SGM 1731-01–15. Description of SGM 1502 was given in *Zverkov et al. (2015)*.

**Skull.** Several sutures in the holotype skull are reinterpreted herein (Fig. 13). The postfrontal medial contact with the supratemporal was imprecisely traced by *Roberts et al. (2014)* likely because of poor preservation. Based on our observations, similarly to other species of *Arthropterygius* the supratemporal of PMO 222.654 forms an anteromedial tongue covering the postfrontal (Figs. 13A and 13B). The parietal of *A. lundi*

has a typical morphology of *Arthropterygius* with a very short medial symphysis and well-pronounced notch posterior to it (Figs. 13A and 13B). The anterior portion of the parietal is divergent and has contributed to a presumably large parietal foramen that was restricted by the frontals anterolaterally (Fig. 13). A ventral exposure of the parietal allows adding that the supratemporal process is relatively slender and the impression of the optic lobe is quite extensive (Figs. 13C and 13D).

**Squamosal.** A squamosal was "*presumed to have been absent in PMO 222.654*" (*Roberts et al., 2014*: 7), on the basis that "*the region in which this element is usually present is well preserved in the specimen*" (*Roberts et al., 2014*: 7), however, as in case of other specimens from Svalbard this assumption is very unlikely (*Zverkov & Efimov, 2019*). In the postorbital region of PMO 222.654, there is an anteroposteriorly elongated depression along the ventral margin of the supratemporal and continuing anteriorly to postfrontal (Fig. 13A). Furthermore, the surface of the postorbital in this region is roughened. The depression has exact the same configuration as the squamosal facet of *A. hoybergeti* (SVB 1451) and presumably represents the facet of squamosal, thereby we suppose that the squamosal was likely present in *A. lundi*, and was similar in morphology to that of *A. hoybergeti*. As this element is delicate and poorly attached to the rest of postorbital bar, it is not surprising that it could be detached and in some cases lost in a number of specimens from Svalbard, including PMO 222.654.

**Quadratojugal.** Considering the slenderness and small size of the quadratojugal, as well as the configuration of its articulation with the quadrate, it is likely that in life this element was largely obscured in lateral view and exposed mostly posteriorly.

**Quadrate.** Judging from its exposed portions, the quadrate of PMO 222.654 has relatively "weak" condyle and a shallow notch of the quadrate foramen; its occipital lamella presumed to be reduced, although is incompletely preserved (Fig. 13). The dorsoventral height of the quadrate of PMO 222.654 is *c.* 105 mm. The facet for quadratojugal is located on the inner surface of the quadrate as in *A. chrisorum* (Fig. 13 cf. Figs. 3J, 3L, 3O and 3P). Nearly the entire posteromedial surface of the quadrate is occupied by an extensive contact with the pterygoid, and only small region in its dorsal part has contact with the supratemporal (Figs. 13C and 13D). Presumably, there was no supratemporal-stapes contact.

**Basisphenoid.** The basisphenoid of PMO 222.654 is mostly hidden in the matrix and covered by other elements, thereby the only significant observation that could be made to the moment is that the facet for the basioccipital was strongly shifted dorsally, an autapomorphy of *Arthropterygius*. The basisphenoid was described in detail for SGM 1502, that is, here referred to as *A. lundi* (see *Zverkov et al., 2015*).

**Opisthotic.** Although it was not reported by *Roberts et al. (2014)*, the nearly complete right opisthotic is present in the holotype (PMO 222.654). This element was collected from the weathered medial side of the specimen in the occipital region (A.J. Roberts, January
2019, personal communication). The paraoccipital process of the opisthotic is relatively short and very robust. The facet for the supratemporal is triangular in outline (Fig. 14E). The lateral muscular ridge is well developed. The stapedial facet is roughly trapezoidal in outline (Fig. 14D). The facet for the basioccipital is quadrant in outline with convex margin directed dorsolaterally (Fig. 14B); it is as large as the stapedial facet. A V-shaped impression formed by two smooth-floored semicircular canals of the otic capsule is deep. Impressions of the horizontal semicircular canal and posterior vertical semicircular canal are nearly equal in length as in *A. hoybergeti*. The posterior vertical semicircular canal impression is only slightly wider. The impression housing the posterior ampulla, utriculus, and the sacculus is expanded, especially in its anteroventral part, to where sacculus impression continues (Figs. 14F and 14G).

**Dentition.** The dentition of *A. lundi* is weak compared to that of *A. chrisorum* (PMO 222.669) and *A. hoybergeti* (SVB 1451). The crowns are slender and their enamel is subtly ridged (Figs. 11B and 11C). An estimated crown height is less than nine mm in PMO 222.654, as calculated by *Roberts et al. (2014*: 15*)*. The largest crown of SGM 1502 is 10 mm high and has five mm in basal diameter.

**Axial and appendicular skeleton.** Not much could be added to the thorough description of the axial and appendicular skeleton of *A. lundi* made by *Roberts et al. (2014)*. Among the interesting traits not mentioned by the aforementioned authors are: the extensive circular facet on the clavicle that formed a firm articulation with the acromial process of the scapula (Figs. 15D and 15E) and, typical of the genus, pronounced angle close to 90° between the articulated coracoids (Fig. 15Q). A "foramen" located on the ventral surface of the interclavicle of PMO 222.654, is likely an artefact of preservation, thus not an autapomorphic trait as was supposed by *Roberts et al. (2014)*. The interclavicular trough is very deep unlike in other species of *Arthropterygius* and in other ophthalmosaurids in general, thereby we agree with the idea of *Roberts et al. (2014)* that this could be considered as an autapomorphy. A bulge in the middle of the interclavicle posterior median stem is present in PMO 222.654 (Figs. 15F and 15G), supporting our assumption that this is a characteristic trait of *Arthropterygius*.

The well-preserved coracoid and scapula of SGM 1731-01–15 demonstrate a typical morphology of *Arthropterygius* (Figs. 15A–15C). The coracoid is slightly longer anteroposteriorly than wide mediolaterally; it bears a prominent anteromedial process, laterally limited by an extensive anterior notch. The posterior portion of the coracoid is strongly compressed and forms a convex protrusion posteriorly (Fig. 15A). The scapula has a well-developed acromial process, nearly equal coracoid facet and glenoid contribution (the latter is slightly shorter) and typical mediolaterally compressed, oval in cross-section scapular shaft (Figs. 15B and 15C).

**Humerus.** Although coracoid and scapula do not bear any specific traits in *A. lundi*, the humerus does. Having humeri nearly identical to that of PMO 222.654, both SGM 1502 and SGM 1731-01–15 well complement the hypodigm. The humerus of *Arthropterygius lundi* has a characteristic isometric proximal end as high dorsoventrally as long anteroposteriorly, and strongly flattened distal end and posterior portion of the shaft

(Figs. 15H–15L and 15N–15P). The dorsal process and deltopectoral crest of the humerus are relatively poorly developed. The ventral skew between the radial and ulnar facets is pronounced in a lesser degree than in the type species, but stronger than in *A.* cf. *hoybergeti* (YKM 63548 and UPM 2442).

**Epipodial and autopodial elements.** The epipodial and autopodial elements in SGM 1731-01–15 and PMO 222.654 are virtually identical (cf. Figs. 15M and 15N). The anterior accessory epipodial element is circular in dorsal view. The radius has a typical pentagonal shape in dorsal view. The ulna is markedly larger than the radius, it is somewhat hexagonal, lacking a perichondral ossification along its posterior edge. Distally ulna bears three nearly equal facets for the intermedium, ulnare and the pisiform (Figs. 15M and 15N). The intermedium is diamond-shaped in dorsal view, having equal contacts with the radius and ulna and bearing two distal facets, evidently for distal carpals three and four. The autopodial elements are mostly circular in outline and were loosely packed as in *Ophthalmosaurus icenicus* (*Moon & Kirton, 2016*) and other species of *Arthropterygius*. Of certain interest are two small ossicles that are semicircular in outline, having perichondral ossification along one of the edges (Fig. 15N). These are probably the pisiform and an element of a postaxial accessory sixth digit.

*Arthropterygius volgensis* (*Kasansky, 1903*) comb. nov.

(Figs. 16 and 17)

v*1903 *Ichthyosaurus volgensis* *Kasansky*: 29, Tables I and II.

1910 *Ophthalmosaurus* sp.; *Bogolubov*: 472 [*pars*].

2000 *Otschevia? volgensis*; *Arkhangelsky*: 550.

2000? *Ophthalmosaurus* sp.; *Storrs, Arkhangel'skii & Efimov*: 197 [*pars*].

2003 *Ichthyosaurus volgensis* *Kasansky, 1903* nomen dubium: *McGowan & Motani*: 134

2008 Undorosaurinae gen. indet.; *Arkhangelsky*: 253 [*pars*].

**Holotype and only specimen:** KSU 982/P-213, see Table 1.

**Diagnosis:** *Arthropterygius volgensis* differs from the other species of *Arthropterygius* by the following characters: gracile articular condyle of the quadrate, less high dorsoventrally and less obtuse posteriorly, do not forming a pronounced ventral angle; and square ventral outline of the basisphenoid with posterior end of the element mediolaterally wider than the anterior end, due to a pronounced reduction of the basipterygoid processes.

**Occurrence:** *Arthropterygius volgensis* is known from only the type locality to the moment: the mouth of the Berezoviy Dol Ravine near Novaya Racheika Village, Syzran District, Samara Region. Upper Jurassic, Middle Volgian, *Dorsoplanites panderi* Ammonite Biozone.

## DESCRIPTION
### Skull

**Supratemporal.** A posterodorsal portion of the right supratemporal is preserved (for the figure see *Kasansky, 1903*, Table 1; Fig. 10). The medial ramus is massive and mediolaterally short, it bears a concave facet for articulation with the parietal.

**Parietal.** The parietal is well preserved and similar to that of other *Arthropterygius* species. It possess a relatively elongated and slender supratemporal process (Fig. 16P). The posterodorsal surface of the supratemporal processes is rugose with the central ridge that contributed to a somewhat peg-and-socket articulation with the supratemporal (Fig. 16P). The medial articular facet is anteroposteriorly shortened; its surface is deeply ridged for a strong interdigitating articulation with the contralateral parietal. Posterior to the facet is a pronounced notch of finished ossification (Figs. 16P and 16S). Anteriorly, the parietal bears rugose facets for articulation with the frontal and postfrontal. The frontal facet completely occupies the anterior edge of the parietal, thus, the parietal unlikely contributed to the posterior border of the parietal foramen. Ventral surface of the element is divided into two areas: the deep and extensive impression of the cerebral hemisphere occupy more than a half of the anterior ventral surface (Fig. 16R, ich); posteriorly situated optic lobe impression, which is roughly circular in outline, occupies the rest of the element (Figs. 16R, iop). The dorsal surface of the parietal is convex and nearly horizontal along the midline in lateral view. There is no sagittal eminence.

**Quadrate.** The articular condyle of the quadrate is relatively reduced and dorsoventrally low compared to that of *A. chrisorum*. The articular and surangular bosses of the condyle are nearly equal in size (Figs. 16O and 16N). The articular boss is only slightly more pronounced ventrally, however, its ventral margin is gradually curved, but not V-shaped as in *A. chrisorum*. The facet for the quadratojugal is a small depression on the dorsal surface of the condyle (Fig. 16N). The quadrate foramen is shallow due to a reduction of the articular condyle and the occipital lamella (Fig. 16L). The occipital and pterygoid lamellae are slightly demarcated one from another forming an angle of *c*. 145°. A circular depression of the stapedial facet is located in the middle of the medial surface (Fig. 16L).

**Basisphenoid.** The basisphenoid is square in ventral view: its posterior and anterior ends are nearly equal in mediolateral length (Fig. 16A). The basipterygoid processes are reduced and faced anterolaterally. The basioccipital facet is a broad hexagonal irregularly pitted surface that faces posterodorsally. A pentagonal dorsal plateau is mediolaterally wide. The stapedial facet is oblique and relatively small (Fig. 16C). The anterior wall is high and vertical, even on the lateral sides. The dorsum sellae, located in the middle of the anterior surface, is smoothly bordered from the rest of the anterior wall (Fig. 16D). The impressions of a cartilaginous continuation of the crista trabecularis are well-pronounced (Fig. 16D). The posterior foramen for the internal carotid arteries opens posteroventrally, forming a medial notch of the posteroventral edge of the basisphenoid, as is CCMGE 3–16/13328, which may be due to the immaturity of these individuals.

**Opisthotic.** The paraoccipital process of the opisthotic is shortened and robust, however, this could be regarded as an immature condition as was discussed by *Kear & Zammit (2014)*. The facet for the supratemporal is triangular in outline (Fig. 16H). The lateral muscular ridge is well pronounced. The trapezoid in outline stapedial facet is larger than the facet for the basioccipital, which is quadrant in outline with convex margin

directed dorsolaterally (Figs. 16F and 16K). The stapedial facet bears a deep straight mediolateral groove either for VII or for IX nerve in its middle (Fig. 16K). The impressions for the semicircular canals of the otic capsule are deep and nearly equal in length as in other species of *Arthropterygius*. The impression of the posterior vertical semicircular canal is wider than that of the horizontal semicircular canal. The impression housing the posterior ampulla and the sacculus is expanded (Fig. 16J).

**Articular.** The articular is anteroposteriorly elongated and trapezoid in outline (Figs. 16T and 16U). It is similar to that of *Arthropterygius lundi* (*Roberts et al., 2014*) and PMO 222.669, the referred specimen of *A. chrisorum*, being more anteroposteriorly elongated than in holotypes of *A. chrisorum* and *A. hoybergeti* (cf. Fig. S4; *Maxwell, 2010*; *Delsett et al., 2018*).

**Axial skeleton.** The detailed description and measurements of the vertebral column (which is nowadays lost) were provided by *Kasansky (1903)*.

**Pectoral girdle.** The preserved middle fragments of clavicles (Fig. 17H) have a morphology common of ophthalmosaurids: these are anteroposteriorly thin and dorsoventrally high elements, curving in dorsolateral direction. The clavicles are dorsoventrally high as in other species of *Arthropterygius*. The interclavicle (Figs. 17H and 17I) is a relatively large element, being approximately 2/3 of the coracoid length. Its posterior median stem is shaft-like, ventrally convex, and dorsally bearing a shallow trough. The scapula is incompletely preserved in two fragments. The acromial process of the scapula is large and flattened, anteroventrally curving at the anterior edge (Fig. 17G). The scapular shaft is mediolaterally compressed, as in other species of *Arthropterygius* and ophthalmosaurines *Ophthalmosaurus icenicus* and *Acamptonectes densus* (*Fischer et al., 2012*; *Moon & Kirton, 2016*). Both coracoids are well preserved, they are rounded in general outlines; however, their anteroposterior length slightly exceeds mediolateral width. The ventral surface of the element is slightly saddle-shaped (Fig. 17B), whereas the dorsal surface is nearly flat (Fig. 17A). The scapular facet is demarcated by an obtuse angle (160°) from the glenoid contribution. The medial symphysis is dorsoventrally thin, extending along anterior two-thirds of the coracoid, as in *A. chrisorum* and *A. lundi* (*Roberts et al., 2014*). The angle between articulated coracoids is close to 90° (Fig. 17E).

**Femur.** The only distal portion of the right femur is preserved (Figs. 17J–17M). Its distal facets are poorly ossified and slightly demarcated, thus it is even hard to say, whether two or three distal facets are present (Figs. 17J, 17K and 17M). The ventral process, located in the middle of the ventral surface is more prominent than the anteriorly shifted dorsal process (Fig. 17L).

**Remarks.** Kasansky originally identified the femur as a humerus, at the same time two broken pedicles of the neural arches were identified as femora (*Kasansky, 1903*).

The holotype and only known specimen KSU 982/P-213 is a juvenile individual, thereby the value of features used as diagnostic could be questioned. Indeed, a number of observed traits could be interpreted as juvenile conditions: reduced occipital lamella of the
quadrate, minimally developed basipterygoid processes, and short paroccipital process of the opisthotic (see *Kear & Zammit, 2014*). However, a series of specimens of presumably different age classes available now for *A. chrisorum* allows supporting some of our conclusions. Although the relative development of the basipterygoid processes of the basisphenoid during the ontogeny is supported by our observations, we state that the general ventral (or dorsal) outline of the basisphenoid is stable between all the age classes. Kear & Zammit stated that in the in utero *P. australis* "the basipterygoid processes are minimally developed, giving the basisphenoid a much narrower anterior profile when compared with those of adults" (*Kear & Zammit, 2014*: 77). Based on this, they concluded that for characters dealing with a shape of basipterygoid processes, that is, *Maxwell (2010*: char. 11*)* and *Fischer et al. (2011*: char. 17 and 2012: char. 16*)*, foetal individual scores differently than mature ones. However, this is a subjective observation, as both foetal and mature *P. australis*, regardless the state of development of basipterygoid processes, preserve generally "pentagonal" (or, it is better to say, trapezoidal) ventral outline of the basisphenoid with anterior region markedly wider than the posterior part. This is clearly seen from the Fig. 5M of *Kear & Zammit (2014)*. In contrast, taxa with "square" ventral outline of the basisphenoid always have the same width of anterior and posterior basisphenoid (Nikolay G. Zverkov, 2015–2018, personal observation). All specimens of *Arthropterygius chrisorum* have basisphenoid, that is, mediolaterally wider anteriorly than posteriorly. Indeed, the small (juvenile) CCMGE 3–16/13328 has narrower anterior profile when compared with those of large individuals CCMGE 17–44/13328 and CMN 40608 (Fig. 18), supporting the observation of *Kear & Zammit (2014)*; still the anterior region of the basisphenoid of small individual CCMGE 3–16/13328 is wider than the posterior region (Fig. 18B). In contrast, the posterior region of the basisphenoid of KSU 982/P-213 is wider than the anterior region (Fig. 18R); although CCMGE 3–16/13328 and KSU 982/P-213 represent presumably close ontogenetic stages (basisphenoid and quadrate of KSU 982/P-213 are slightly smaller, whereas coracoid is bigger than those of CCMGE 3–16/13328). Another marked difference of CCMGE 3–16/13328 and KSU 982/P-213 is the shape of the condyle of their quadrates. Whereas CCMGE 3–16/13328, CCMGE 17–44/13328, and PMO 222.669, regardless differences in size, have similar shape of the condyle, KSU 982/P-213 differs in having less dorsoventrally high condyle with gradually curving (not V-shaped) ventral margin. This suggests that the shape of the quadrate could also be regarded as interspecifically and ontogenetically stable feature. Thereby we conclude that at the current state of knowledge, *A. volgensis* should be regarded as a distinct valid species of *Arthropterygius* rather than a synonym of other known species of the genus or a nomen dubium.

**Measurements.** See *Kasansky (1903)* and Table S4.

## Phylogenetic analysis

Our analysis of the full dataset recovered ten most parsimonious trees of 310 steps with the consistency index (CI) = 0.416 and retention index (RI) = 0.662. The strict consensus (length of 321 steps; CI = 0.402; RI = 0.642) is relatively well resolved, however,

supports for relationships within Ophthalmosauridae are still low (Fig. 19). Despite the modifications of the original matrix, the recovered topology is nearly identical to that of *Zverkov & Efimov (2019)*, except for minute changes in relations of derived-most platypterygiines that are even more badly resolved. A clade that includes species of *Arthropterygius* ("A" in Fig. 19) is recovered as the sister group to Platypterygiinae. Sister relations of *Arthropterygius* and platypterygiines are supported by two synapomorphies: "T"-shaped prootic osseous labyrinth (49.0→49.1) and absence of the obturator foramen in the ischiopubis (98.1→98.2).

Only two most parsimonious trees (length of 300 steps, CI = 430, RI = 662) were recovered by the pruned analysis. In the strict consensus tree (length of 302 steps, CI = 425, RI = 656; Fig. 19B), Platypterygiinae is relatively better resolved. Surprisingly, *Caypullisaurus* is found as a sister, not to *Grendelius*, but to *Leninia* (based on two non-unique synapomorphies: presence of prefrontal dorsomedial expansion (16.0→16.1), and squared sqamosal (34.1→34.0). However, the relations of derived platypterygiines is not a focus of the current paper. Concerning this manuscript, *Arthropterygius* clade is recovered as a sister group to ophthalmosaurines. These two groups from a clade with low support, and share three synapomorphies (presence of the lateral "wing" of the nasal (14.0→14.1); absence of supratemporal-postorbital contact (27.1→27.0); and circular shape of the basioccipital condyle (43.1→43.0).

The *Arthropterygius* clade is supported by nine autapomorphies: posterior position of the foramen for internal carotid arteries (unique, 40.1→40.2); dorsally facing basioccipital facet of the basisphenoid (non-unique, 41.0→41.1); raised opisthotic facet of the basioccipital (non-unique, 46.0→46.1); anteriorly shifted stapedial and opisthotic facets of the basioccipital (unique, 47.0→47.1); gracile stapedial shaft (non-unique, 52.0→52.1); robust clavicles (unique, 78.0→78.1), ulnar facet/radial facet ratio less than 0.83 (unique, 84.0→84.1); weak quadrate condyle (non-unique, 110.0→110.1); angle between the articulated coracoids less than 110° (unique, 111.0/1→111.2).

In both the full and pruned analyses the *Arthropterygius* clade has a comparatively high Bremer support values (four and five), Bootstrap and Jackknife (more than 80), thus being the best supported clade in our analyses (Fig. 19). The result further augment our taxonomic decision, leaving no substantial reasons to consider taxa within the *Arthropterygius* clade as representatives of separate genera.

## Multivariate analysis of ophthalmosaurid humeral morphology

The first two axes describe over 60% of the total variance (38.12% and 21.9%, respectively). All variables show low loadings on PC1 ($\geq$0.50; <0.50); among them better pronounced are humeral distal expansion (variable 2: −0.46), humeral proximodistal proportionality (variable 4: 0.43), humeral distal compression (variable 6: −0.4), relative size of faae (variable 7: −0.42), as well as relative dorsoventral width of ulnar and radial facets (variable 9: 0.32) and an angle between these facets (variable 10: 0.28). For the PC2 highest positive loadings are shown by variables 1 (0.58), 4 (0.36), 8 (0.37) and negative loadings by variables 9 (−0.33) and 10 (−0.42). Thereby PC2 characterize humeral proximal expansion, humeral proximodistal proportionality, relative anteroposterior (v8) and dorsoventral (v9)

width of ulnar, and radial facets as well as angle between these facets (v10). The distribution of variable loadings could be found in Table S9.

Considering low sampling for some of the taxa in our analysis, it is hard to say with confidence if the absence of marked morphospace overlap between the Jurassic ophthalmosaurid taxa is a true condition, or it is biased by the sampling. Whether or not, it is clear that most of the Late Jurassic ophthalmosaurid genera are well separated, forming four clusters: *Brachypterygius–Grendelius* cluster (high values on PC1 and low values on PC2), *Ophthalmosaurus* cluster (low values on both PC1 and PC2), *Arthropterygius* cluster (low values on PC1, high values on PC2) and *Undorosaurus* cluster (moderate values on both the axes, so that its morphospace is between the others) see Fig. 20.

Our PCA demonstrate a relatively wide morphospace occupation for species of *Arthropterygius*, which is mostly due to *Arthropterygius* cf. *hoybergeti*, having humeri that are morphologically closer to "moderate" ophthalmosaurid condition and thereby falling closer to *Undorosaurus gorodischensis* and *Platypterygius hercynicus* (Fig. 21). *A. lundi* falls close to *A. chrisorum* (Fig. 20).

*Undorosaurus gorodischensis* morphospace is separated from other species of *Undorosaurus* by the first principal component axis, as *U. nessovi* and *U. trautscholdi* demonstrate low positive and negative values on PC1, and more pronounced negative value on PC2 in case of *U. nessovi*. In general morphology, *U. gorodischensis* have anteroposteriorly elongated humeral proximal end, that is, of roughly oval outline, whereas *U. nessovi* and *U. trautscholdi* are characterized by a nearly circular outline of the humeral proximal end, which is depicted by PC1 partially responsible for humeral proximodistal proportionality, in this regard *U. nessovi* is closer to *Ophthalmosaurus icenicus* morphospace (Fig. 20).

Several derived Cretaceous platypterygiines, added to our analysis, occupy different parts of the morphospace also demonstrating the potential of humeral morphology for distinguishing Cretaceous ichthyosaurs. At the same time, they occur in the morphospace of Middle to Late Jurassic ophthalmosaurids. *Platypterygius australis* is found within the morphospace of *O. icenicus*, *P. americanus* is found in morphospace of *Brachypterygius–Grendelius* and *Undorosaurus* and *P. hercynicus* shows high positive values on PC2 and cross the morphospace of *Undorosaurus* and *Arthropterygius*. Only *Plutoniosaurus* and *Sveltonectes* occupy the new morphospace with no overlap with Jurassic ophthalmosaurids.

The interesting result of our analysis is that in some ophthtalmosaurid individuals left and right humeri can fall wider to each other than to humeri of other specimens of the species and even to other species and genera, indicating the presence of a pronounced humeral asymmetry in ophthalmosaurids. The most outstanding specimen with humeral asymmetry in our analysis is *Platypterygius hercynicus*. Although the asymmetry is a normal condition in all bilaterally symmetric organisms, and a marked example of natural humeral asymmetry was recently described for *Undorosaurus* (*Zverkov & Efimov, 2019*), it is more likely that pronounced asymmetry, like that in *P. hercynicus*, could be explained by artefacts of preservation and/or pathologies. This assumption is consistent

with the recent study of the effect of taphonomy on asymmetry in fossil organisms (*Hedrick et al., in press*).

## DISCUSSION

### New concept of *Arthropterygius*: discussion of taxonomic decisions

New observations and results of our phylogenetic analysis, in our opinion, do not allow to further support the generic validity of ichthyosaurs from *Arthropterygius* clade. Within the clade, *A. volgensis* and *A. lundi* are recovered in a polytomy at the base. No autapomorphies are found for *A. volgensis* and a single non-unique autapomorphy, long anterior process of the maxilla ($10.0 \rightarrow 10.1$), characterises *A. lundi*. The smaller clade of *A. hoybergeti* and *A. chrisorum* is characterized by two non-unique autapomorphies: pronounced striation of the crown ($1.1 \rightarrow 1.0$) and well-developed occipital lamella of the quadrate ($36.1 \rightarrow 36.0$). No autapomorphies are found for *A. hoybergeti*, whereas there are two for *A. chrisorum*: reduction of the angular process of the quadrate ($109.0 \rightarrow 109.1$) and a very large anterior accessory epipodial facet of the humerus ($112.1 \rightarrow 112.2$). In our opinion, none of the autapomorphies characterising the taxa within the discussed clade is sufficient for applying an alternative taxonomic context with *Janusaurus* and *Palvennia* being regarded as valid genera, sister to a more derived *Arthropterygius*.

The results of PCA shows that the morphospace occupied by ichthyosaurs of *Arthropterygius* clade is only slightly larger than the morphospace occupied by *Ophthalmosaurus icenicus* solely. At the same time, no sufficient overlap with other ophthalmosaurid clusters is observed, which could also be considered as support of the new taxonomic context for *Arthropterygius*.

On Fig. 21, we summarized the importanl overlapping skeletal elements of *A. chrisorum* (holotype, CMN 40609, and the referred specimens from Franz Joseph land), "*Keilhauia nui*" (holotype, PMO 222.655), "*Palvennia*" *hoybergeti* (holotype, SVB 1451, and referred specimen PMO 222.669), "*Janusaurus*" *lundi* (PMO 222.654), and "*Ichthyosaurus*" *volgensis* (KSU 982/P-213) and compared them with other well-known contemporary taxa. Following is consideration of taxonomic attribution of specimens referred to as *Arthropterygius* in the present contribution.

In the original description of *Palvennia hoybergeti* (*Druckenmiller et al., 2012*), the only considered overlapping elements with the holotype of *A. chrisorum* (CMN 40608) were the basioccipital, atlas-axis complex and the anterodistal fragment of the humerus. Only the basioccipital was compared with that of *A. chrisorum* and their similarity was noted: "*in posterior articular view the basioccipital of* Palvennia *is nearly oval in outline with only the condyle visible, more similar to* Arthropterygius (*Maxwell, 2010*) *and* Platypterygius australis (*Kear, 2005*)" (*Druckenmiller et al., 2012*: 337). As a difference between the two, the following feature was proposed: "*in* Arthropterygius, *the anterior face of the basioccipital possesses a notochordal pit and a distinct basioccipital peg, both of which are absent in* Palvennia" (*Druckenmiller et al., 2012*: 337). Based on our observations on the holotype of *Palvennia hoybergeti* (SVB 1451) we are unable to support the latter conclusion. An anterior protrusion of the basioccipital under the floor of the foramen magnum interpreted by *Maxwell (2010)* as an "*incipient basioccipital peg*" is also present in

*P. hoybergeti* (SVB 1451) and *J. lundi* (PMO 222.654) although in a slightly lesser degree (Nikolay G. Zverkov, 2017, personal observation). This anterior protrusion was reported for some other ophthalmosaurids (see *Moon & Kirton, 2016*: 41). Although this structure is a vestige of a basioccipital peg, the condition observed in *Arthropterygius* could not be considered as a plesiomorphic state, as was supposed and coded in some previous works (*Fischer et al., 2011*, *2012*). When present, the basioccipital peg is large and tapered, terminating in a pointed tip, but not in a flattened pitted facet, as it does in *Arthropterygius* and some other ophthalmosaurids (see Discussion of this structure in *Moon & Kirton (2016*: 41*)*). It is hard to verify the absence or presence of a notochordal pit on the anterior surface of the discussed protrusion in SVB 1451. It was described as absent in this specimen by *Druckenmiller et al. (2012*: 330*)*, however, we were unable to verify this because of preservation of the anterior region in the latter. Furthermore, the value of this character (i.e., presence/absence of the anterior notochordal pit) is somewhat uncertain, as for the taxa where more than a couple of specimens is know, for example, *Ophthalmosaurus icenicus*, both the conditions could be observed (Nikolay G. Zverkov, 2018, personal observation on *O. icenicus* specimens in CAMSM and NHMUK).

The extracondylar area of the basioccipital of *A. chrisorum* CMN 40608 is extremely reduced and completely unseen in posterior view, as in *P. hoybergeti* (SVB 1451) and *Janusaurus lundi* (*Roberts et al., 2014*; although this element in the latter is poorly preserved). However, the extracondylar area in these specimens is relatively anteroposteriorly wide in lateral view, unlike that of *Grendelius* spp. (Fig. 21; *McGowan, 1976*; *Zverkov, Arkhangelsky & Stenshin, 2015*). *Maxwell (2010)* has interpreted a part of the extracondylar area as a stapedial facet, probably due to poor preservation of CMN 40608. The stapedial facet of CMN 4060 faces anteriorly and is practically unseen in lateral view (cf. *Maxwell, 2010*, fig. 2D). This feature is present in all the specimens referred to as *Arthropterygius* in current contribution, for which the basioccipital is available (Fig. 21; CMN 40608, SVB 1451, PMO 222.654, PMO 222.669) and is currently unknow for any other ichthyosaurs. Thus, we consider this an autapomorphy of *Arthropterygius*.

Recently referred to as *Palvennia hoybergeti* PMO 222.669 shares all features of *A. chrisorum*. However, *Delsett et al. (2018)* provided a brief comparison of PMO 222.669 and *A. chrisorum* (holotype CMN 40608), mainly in their emended differential diagnosis of *P. hoybergeti*. According to those comparisons, *P. hoybergeti* differs from *A. chrisorum* in the following features: anterior face of basioccipital lacks notochordal pit and basioccipital peg (see Comments above); rectangular in outline articular with a slight constriction at the anteroposterior midpoint; longer and narrower anterior notch of the coracoid; proximodistally shorter dorsal process of the humerus (this could be explained by ontogenetic variation, see Discussion of ontogenetic changes below); not as convex articular faces of epipodial elements (also could be due to ontogenetic and interspecific variation; see below). We suppose that none of these differences is sufficient enough to distinguish the species.

The overlapping elements between PMO 222.669 and the holotype of *A. chrisorum* (CMN 40608) are the basioccipital, basisphenoid, articular, atlas-axis complex, and other

vertebrae, pectoral girdle elements, and forelimbs. We suppose that this is sufficient overlap to consider the potential affinity of the specimen. For the comparison of these specimens especially valuable are basicranial elements and the humerus that bear a number of autapomorphic traits (Fig. 21). Basioccipitals of PMO 222.669 and CMN 40608 are very similar and have a feature not observed in most of the other ophthalmosaurids: stapedial and opisthotic facets of the basioccipital shifted anteriorly and poorly visible in lateral view (this condition is shared only with type specimens of *Palvennia hoybergeti* (SVB 1451) and *Janusaurus lundi* (PMO 222.654), Nikolay G. Zverkov, 2017, personal observation). The basisphenoids of these specimens (i.e., PMO 222.669 and CMN 40608) have a foramen for the internal carotid arteries opening posteriorly and not visible in ventral view, being separated from the ventral surface by a thin shelf. This condition is known exclusively for *A. chrisorum* and is autapomorphic. The humeri of PMO 222.669 and CMN 40608 bear a strong ventral skew between the radial and ulnar facets with the dorsoventral thickness of the radial facet greatly exceeding that of the ulnar facet (autapomorphic feature of *A. chrisorum*). The proximal ends of the discussed humeri are strongly anteroposteriorly elongated with reduced deltopectoral crest shifted to its anterior edge (among other ophthalmosaurids, similar condition is found in *Undorosaurus gorodischensis*; *Zverkov & Efimov, 2019*). The facet for the anterior accessory epipodial element of the humerus is comparatively large, as wide as, and close in size to the radial facet (this is a non-unique autapomorphy of *A. chrisorum*, wich also occurs in mid-Cretaceous "*Ophthalmosaurus*" *cantabrigiensis* *Lydekker, 1888*, *Maiaspondylus lindoei* *Maxwell & Caldwell, 2006*, and "*Platypterygius ochevi*" *Arkhangelsky et al., 2008*, but not in any other ophthalmosaurid, including the holotype of *Palvennia hoybergeti*).

Indeed, PMO 222.669 and the holotype of *Palvennia hoybergeti* (SVB 1451) have a very similar morphology of the dermatocranium, including one feature that was considered as an autapomorphy of *P. hoybergeti*—"a very large pineal foramen" (*Delsett et al., 2018*: 11). However, the frontal region and consequently the shape of the parietal foramen is still unknown for a number of Late Jurassic ichthyosaurs, including *Brachypterygius extremus* (*Boulenger, 1904*), *Nannopterygius enthekiodon* (*Hulke, 1871*)*, Grendelius* spp. (*Zverkov, Arkhangelsky & Stenshin, 2015*), *Undorosaurus* spp. (*Zverkov & Efimov, 2019*), and *A. chrisorum* (i.e., its holotype). We cannot exclude the presence of this trait in any of the listed taxa. In this regard, the diagnostic value of this feature is reduced. At the same time, there is a number of differences between PMO 222.669 and the holotype of *P. hoybergeti* (SVB 1451). These are: basisphenoid posterior foramen for the internal carotid arteries separated from the ventral surface by a thin shelf in PMO 222.669 (shifted to the posteroventral edge of the element in SVB 1451, see Description above); extremely shortened and robust paraoccipital process of the opisthotic (relatively elongated and dorsoventrally compressed in SVB 1451; see Description); slender distal stapedial process (expanded in SVB 1451, see Description above); markedly less expanded mediolaterally anteromedial tongue of the supratemporal (see Description); reduced deltopectoral crest of the humerus shifted to its anterior edge (well pronounced in SVB 1451, see Description); large semicircular facet for the anterior accessory epipodial element of the humerus

(comparatively small and anteriorly tapered in SVB 1451; see Fig. S7A); large and rounded in outline anterior accessory epipodial element (this element is relatively small in SVB 1451, semicircular in outline, with a nearly straight anterior margin, see Description above).

From our observations on PMO 222.669 (NGZ), we have not found any additional differences in overlapping material with both the holotype of *A. chrisorum* (CMN 40608) and the other specimens referred to as *A. chrisorum* herein (especially, CCMGE 3–16/13328 and 17–44/13328). Thus, we refer PMO 222.669 to as *Arthropteryguis chrisorum*.

The specimens from Franz Joseph Land (CCMGE 3–16/13328 and 17–44/13328) are referred to as *A. chrisorum* based on exact the same grounds as PMO 222.669 (see above), except for the features of the basioccipital, that is, lost in both. In addition to this, a femur, present in CCMGE 17–44/13328, is well consistent with that of the holotype of *A. chrisorum* (see Description above).

The humeral morphology of SGM 1573 is typical of *A. chrisorum* with the presence of all autapomorphic traits (listed above). Thus, we consider this sufficient for the referral of SGM 1573 to *A. chrisorum*.

The referred specimens of *A. chrisorum* provide additional overlapping elements: supratemporal, postfrontal, jugal, parietal, quadrate, opisthotic, stapes, interclavicle, clavicle, and scapula. Some of these elements also bear autapomorphic traits.

The supratemporal of PMO 222.669 produce a wide anterior projection—a tongue covering the postfrontal with the formation of a marked facet (this could be observed in the isolated postfrontal of CCMGE 17–44/13328). Among ophthalmosaurids, this condition is found only in *Athabascasaurus bitumineus*, "*Palvennia*" *hoybergeti* and "*Janusaurus*" *lundi* (see Description above).

The jugal is strongly bowed ventrally in CCMGE 17–44/13328 and in PMO 222.669. Among ophthalmosaurids, this condition is shared exclusively with "*Palvennia*" *hoybergeti* and "*Janusaurus*" *lundi* (Fig. 21).

The parietal of PMO 222.669 is characterized by the extremely anteroposteriorly shortened medial symphysis, that is, posteriorly restricted by a pronounced excavation and notch. This morphology became clear after the additional preparation that was performed by the enquiry of NGZ, as in the original study of the specimen (*Delsett et al., 2018*) the interparietal region was described before a preparation. The revealed morphology of the interparietal contact is uniquely shared by PMO 222.669 and holotypes of "*Ichthyosaurus*" *volgensis*, "*Palvennia*" *hoybergeti*, and "*Janusaurus*" *lundi* (Fig. 21). We consider this feature as an autapomorhy of *Arthropterygius* (sensu herein).

The stapes of PMO 222.669 is characterized by a very slender lateral process. Among other ophthalmosaurids, this condition is observed only in "*Palvennia*" *hoybergeti*, "*Janusaurus*" *lundi*, and *Acamptonectes Fischer et al., 2012*. There is a number of differences between PMO 222.669 and *Acamptonectes* both cranial and postcranial (cf. *Fischer et al., 2012*), still, the presence of this trait in PMO 222.669 (referred to as *A. chrisorum* herein) and in "*Palvennia*" *hoybergeti* and "*Janusaurus*" *lundi* could also be used as a non-unique autapomorphy of *Arthropterygius* (in its sense in the current paper).

The complete interclavicles are known for SGM 1573 and the holotype of
"*Janusaurus*" *lundi*. Although there are differences between the two (see Description
above), both bear a bulge in the middle of the interclavicle posterior median stem
(see Description). This feature is unique for ophthalmosaurids and considered as
autapomorhy of *Arthropterygius* herein.

The clavicles are known for PMO 222.669, CCMGE 17–44/13328, "*Palvennia*"
*hoybergeti* (SVB 1451) and "*Janusaurus*" *lundi* (PMO 222.654). In all of these specimens
the clavicle is very dorsoventrally high (medial dorsoventral height to anteroposterior
length ratio greater than 0.2) and extremely robust compared to any other Jurassic
ophthalmosaurid (Fig. 21). We consider this feature as an autapomorhy of *Arthropterygius*
(in its sense in the current paper).

The coracoids in all the specimens, where available (see Table 1), are comparatively
large (the scapular proximodistal length is less than the coracoid anteroposterior
length). In their general shape, the coracoids are similar to those of *Ophthalmosurus*
(*Moon & Kirton, 2016*). Although it is not assessable in all specimens (due to
preservation), when visible, a significant angle close to 90–100° could be observed
between the articulated coracoids (CCMGE 3–16/1332, PMO 222.654, KSU 982/P-213).
This configuration is unknown for other ophthalmosaurids and thus considered
autapomorhic of *Arthropterygius*.

Recently erected from the Berriassian of Svalbard *Keilhauia nui* is also referable
to *Arthropterygius*, however, only in open nomenclature. The holotype and only known
specimen of this taxon is a poorly preserved skeleton of a small individual that was
considered to be of "*late juvenile to adult ontogenetic stage*" (*Delsett et al., 2017*: 14).
From our observations of the holotype (PMO 222.655), we are unable to confirm any of the
evidences proposed by *Delsett et al. (2017)* as supporting maturity of PMO 222.655.
The proximal portion of the humerus of PMO 222.655 is heavily weathered and its
posterior portion is broken so that it is impossible to say something regarding its natural
shape and its value for identification of maturity. The same concerns a texture of the
humeral shaft, which along with other skeletal elements of PMO 222.655 is poorly
preserved, weathered, and partially covered by matrix along with products of pyrite decay.
It is unclear what *Delsett et al. (2017)* meant under the degree of ossification that
"(*when it is possible to observe*) *resembles mature finished bone*," as, in our opinion, all the
available surfaces are too pooly preserved. Furthermore, the facets of appendicular
elements are poorly demarcated from each other, the epipodial and authopodial elements
are rounded and very loosely arranged and the ventral margin of the ischiopubis bears
an excavation along its ventral margin, which could be a result of an extensive cartilaginous
continuation of the element. The listed traits could be considered as arguments for the
immaturity of PMO 222.655, although could be alternatively interpreted as a result of poor
preservation. A natural shape of the ischiopubis is unclear because its proximal portion
is partially eroded and diagenetically compressed. PMO 222.655 is generally similar
to CCMGE 3–16/13328 being only slightly smaller in size, and has a number of features
that are diagnostic of *Arthropterygius*: the humerus of PMO 222.655 has a ventral
skew between the radial and ulnar facets, its ulnar facet:radial facet dorsoventral width

ratio is less than 0.8; the facet for anterior accessory element is nearly as large as the radial facet (a diagnostic feature of *A. chrisorum*); the clavicle of PMO 222.655 is relatively large and robust. Judging from the field photographs (J. Hurum, September 2017, personal communication), the coracoid was originally longer anteroposteriorly than mediolaterally wide and extremely similar to that of CCMGE 3–16/13328, its current shape is likely due to conservation processes. The ischiopubis of PMO 222.655 is plate-like and lacks an obturator foramen similarly to the condition observed in *J. lundi* (PMO 222.654) and several indeterminate ophthalmosaurids from Svalbard (*Delsett et al., 2017*), and unlike any other known ophthalmosaurid. What concerns the ilium of 222.655, its expanded dorsal portion is an important character that probably demonstrates a juvenile condition of what in *A. lundi* (PMO 222.654) developed in an "anteromedial process" and posteriorly curved end (*Roberts et al., 2014*). Thus, expanded dorsal portion of the ilium and plate-like ischiopubis with no obturator foramen could also be generic features of *Arthropterygius* (in its sense in the current paper).

Taking into account all the arguments above, we consider *"Keilhauia nui"* as a *nomen dubium* and identify its type specimen as *Arthropterygius* sp. juv. cf. *A. chrisorum*.

## Ontogenetic changes, intra- and interspecific variation in *Arthropterygius*

Owing to new specimens of *A. chrisorum*, we can now make some observations on the ontogenetic changes and variation in this taxon.

In general, changes in morphological proportions during growth of *A. chrisorum* are consistent with those observed in other ichthyosaurs (*Huene, 1922*; *McGowan, 1973b*; *Deeming et al., 1993*). Having largely incomplete specimens (Table 1; Fig. 22), we are unable to assess the growth of the whole skull and the whole body. Thereby we compared selected cranial and postcranial elements (Fig. 23). The growth of elements of the skull base and occiput of *A. chrisorum* is more or less isometric compared to each other. The same concerns the growth of elements of the appendicular skeleton (Fig. 23A). At the same time, the growth rates differ between the skeletal regions.

Relative anteroposterior length of the basisphenoid and the humerus is among the few ratios that could be calculated for *A. chrisorum* in order to compare the growth of the cranial and postcranial skeleton. In juvenile CCMGE 3–16/13328 this ratio is 0.58, in young adult CCMG E 17–44/13328—0.42, and in mature individual CMN 40608—0.35; thus we observe typical negative allometry. It is not surprising that the growth of the cranial elements is negatively allometric relative to the growth of the appendicular elements. Interesting is that growth of the appendicular skeleton is somewhat positively allometric relative to that of the axial skeleton (Fig. 23A), whereas for *Ichthyosaurus* and *Stenopterygius* this reported as being isometric (*McGowan, 1973b*).

Judging from the available cranial elements, the general morphology and proportions of the occipital region have not undergone dramatic changes with age. Despite differences in size CCMGE 3–16/13328, CCMGE 17–44/13328, and PMO 222,669 have a peculiar shape of the quadrate condyle: it is dorsoventrally high with a V-shaped ventral margin of the articular boss. Furthermore, the quadrate do not develop the anterior

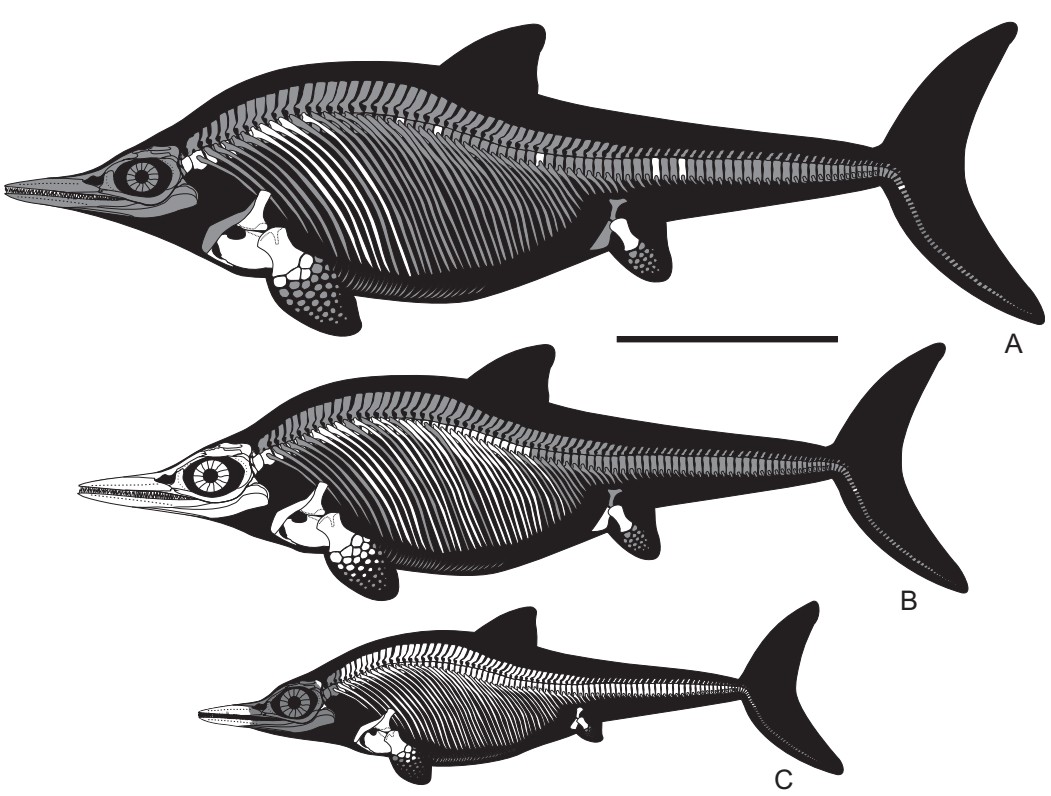

**Figure 22  Skeletal reconstructions of *Arthropterygius chrisorum* old adult based on CMN 40608 and SGM 1573 (A) young adult based on CCMGE 17–44/13328 and PMO 222.669 (B), and juvenile based on CCMGE 3–16/13328 and PMO 222.655 (C).** Unknown skeletal regions are shown in gray. Scale bar equals one m.

protrusion with age. In all specimens of *A. chrisorum*, the basisphenoid is trapezoidal in ventral outline, being mediolaterally wider anteriorly than posteriorly. The juvenile CCMGE 3–16/13328 has a narrower anterior profile when compared to those of adults CCMGE 17–44/13328, PMO 222,669, and CMN 40608 (Figs. 23B–23E), supporting observations of *Kear & Zammit (2014)* on *Platypterygius australis*. The only marked difference of the basisphenoids is the relative position of the posterior foramen for the internal carotid arteries, which is still exposed ventrally in juvenile CCMGE 3–16/13328, but already separated by a grown shelf in young adults PMO 222,669 and CCMGE 17–44/13328 (Figs. 23B–23D).

 All the specimens of *A. chrisorum* have concave humeral distal facets and convex proximal articular facets of the epipodial element. A tendency for deepening of humeral distal facets with age could be observed, however, it is non-uniform. Although the old adult CMN 40608 has very deeply concave facets (*Maxwell, 2010*), comparable in size SGM 1502 has less concave facets and consequently should have had less convex proximal surfaces of the epipodial elements. Considering this variation and the fact that after the publication of *Maxwell (2010)* humerus-epipodial peg-and-socket articulation was reported for other ophthalmosaurids (*Zverkov et al., 2015*), we suggest that "*proximal surface of zeugopodial elements angular in outline for articulation with humerus*" (*Maxwell, 2010*: 404) cannot be further considered as a diagnostic character of *Arthropterygius*.

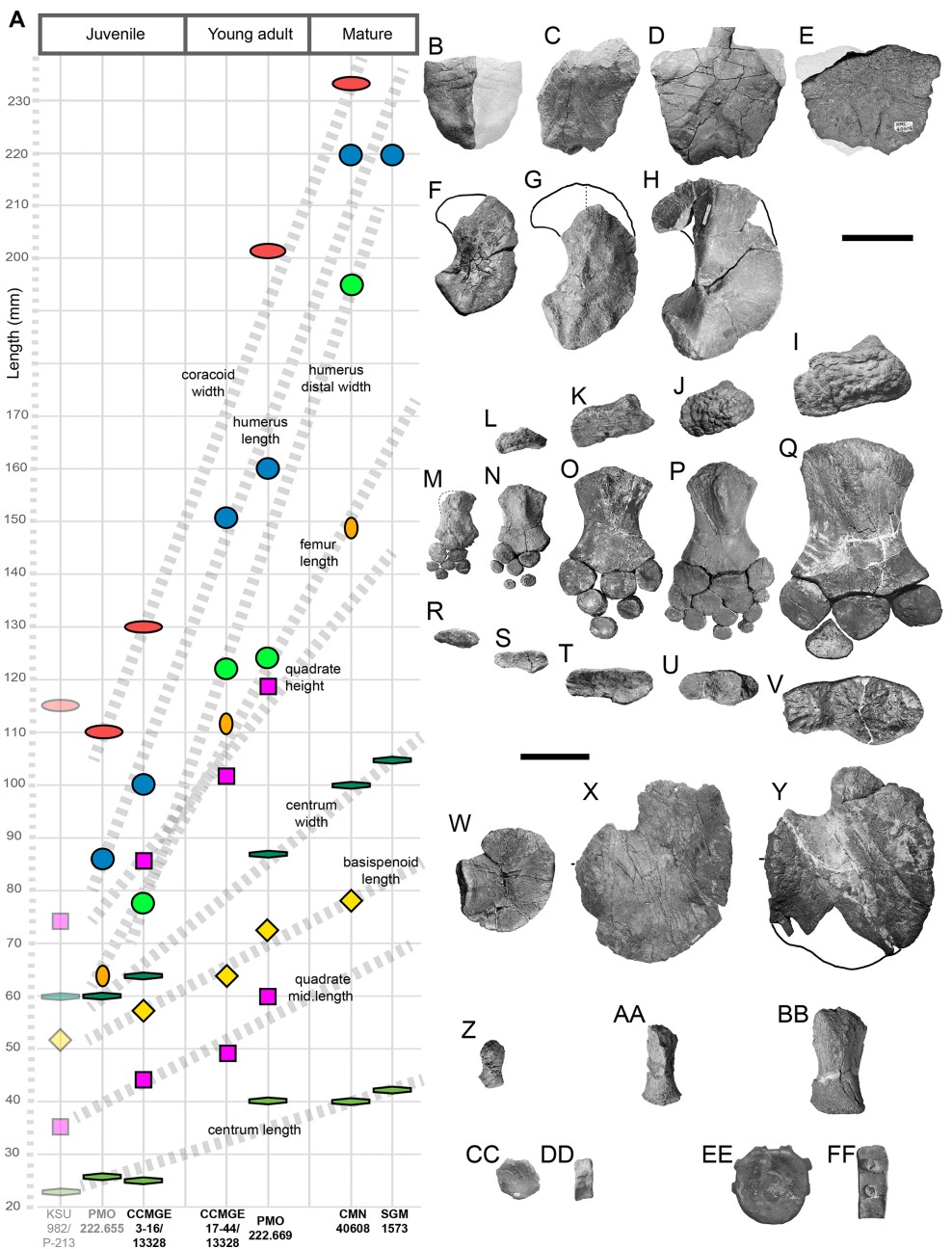

**Figure 23 Cranial and postcranial ontogeny of *Arthropterygius chrisorum*.** (A) simplified "plot" showing measurements of various cranial and postcranial elements versus hypothesized ontogenetic stage of the specimen (specimens are arranged equidistantly to each other and divided onto three ontogenetic categories: juveniles, young adults and mature). (B–C) ontogenetic series of selected skeletal elements of *A. chrisorum*, from top to bottom: basisphenoids in ventral view; quadrates in posteromedial view; humeri in proximal wiew; forelimbs in dorsal view; humeri in distal view; coracoids in ventral view; femora in ventral view; posterior presacral vertebrae in articular and lateral views. Specimens: juvenile of *A. chrisorum* CCMGE 3–16/13328 (B, F, L, N, S, W, CC, DD); young adults of *A. chrisorum* CCMGE 17–44/13328 (C, G, K, O, T, AA) and PMO 222.669 (D, H, J, P, U, X); large mature individuals of *A. chrisorum* CMN 40608 (E, Q, V, Y, BB, EE, FF) and SGM 1573 (I, EE, FF); juvenile of *Arthropterygius* sp. juv. cf. *A. chrisorum* PMO 222,655 (M, R, Z); (E, Q, V, Y, BB) are modified from *Maxwell (2010)*. Scale bars for (B–H) equal five cm, for (I–FF) 10 cm.

It is interesting that there are no marked differences in humeral morphology between the juvenile and adults. The marked change is the angle between the radial facet and facet for the anterior accessory epipodial element that became less pronounced with age (Figs. 23M–23Q). The same concerns the angle between the ulnar and radial facets. The absence of marked ontogenetic changes in relative size and shape of the humeral distal facets supports their diagnostic value. Thus, the features related to humeral distal facets could likely be used for diagnosing specimens of *Arthropterygius* irrespective of their maturity.

As in case of *Undorosaurus* (Zverkov & Efimov, 2019) and *Grendelius* (Zverkov, Arkhangelsky & Stenshin, 2015), species of *Arthropterygius* could be potentially distinguished based exclusively on humeral morphology, which was already demonstrated above by the results of PCA. Especially valuable is the outline of the humeral proximal end—each of these genera has species with anteroposteriorly elongated humeral proximal ends (*Grendelius zhuravlevi*, *Undorosaurus gorodischensis*, *A. chrisorum*) and those with isometric proximal ends (*G. alexeevi*, *U. nessovi*, *U. trautscholdi*, *A. lundi*). We cannot exclude the possibility that some of these species may actually represent males and females, thus demonstrating sexual dimorphism, differing in limb morphology in a way, similar to that hypothesized for Triassic ichthyopterygians *Chaohusaurus* and *Shastasaurus* (Shang & Li, 2013; Motani et al., 2018). However, given other existing differences (especially cranial) between the discussed species, and considering that in some genera more than one species with either elongated or isometric humeral proximal end could be present, it is impossible to say, which of the species are representing sexual morphs of the same species and which of them are morphs of other species. Thereby, in the current state of knowledge, we prefer to retain all the "morphs" as separate species.

## Palaeobiogeographic implication of *Arthropterygus*

After the discovery of *Arthropterygius* in Argentina (Fernández & Maxwell, 2012), this taxon, even being known from a couple of specimens, has already raised a question regarding the cosmopolitan distribution of ichthyosaurs (Fernández & Maxwell, 2012; Zverkov et al., 2015). New discoveries further support the idea that many ophthalmosaurids have had a widespread distribution. For more details on palaeobiogeography of Boreal ichthyosaurs, we direct the reader to (Zverkov et al., 2015; Zverkov & Efimov, 2019; Arkhangelsky et al., in press).

Given the new data, *Arthropterygius* seem to be very common ichthyosaurs for the Volgian (especially middle Volgian): *A. chrisorum* is found in Arctic Canada, Svalbard, and Volga Region, thus indicating a wide distribution of this species across the Arctic basins and Middle Russian Sea. The same concerns *Arthropterygius hoybergeti* and *A. lundi*, which are both known from Svalbard and Volga Region (Fig. 24A). Additionally *A. lundi* is known from the Timan-Pechora, thus unambiguously demonstrating that the Mezen-Pechora Strait connecting the Middle Russian Sea with Arctic Basins was used as a passage during this time.

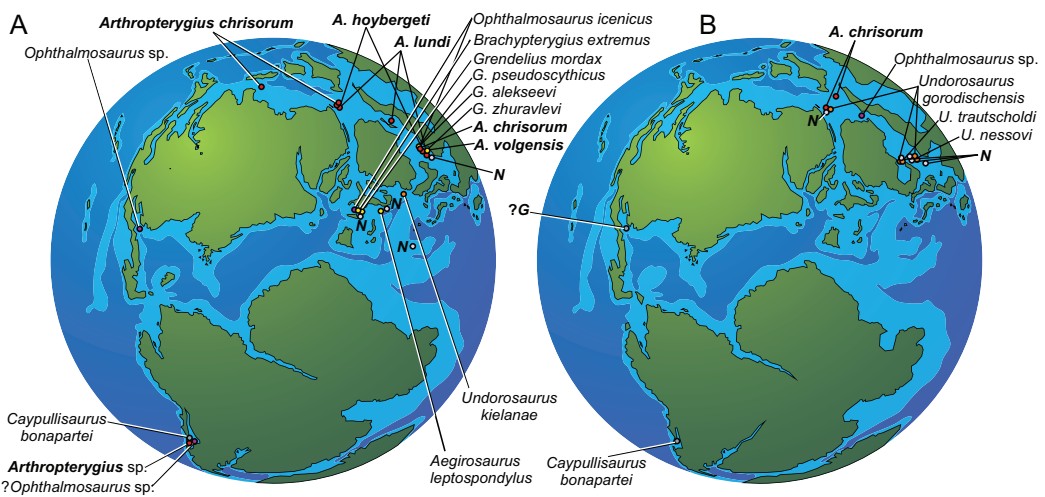

**Figure 24 Distribution of ophthalmosaurids in the Kimmeridgian—early Middle Volgian (*Dorsoplanites panderi* Chron) time interval (A), and Middle Volgian (*Virgatites virgatus* Chron)—Late Volgian (late Tithonian to early Berriassian time interval) (B).** Abbreviations: N, representatives of *Nannopterygius* clade; G, possible *Grendelius*. Reconstruction is modified from *Zakharov et al. (2014).*

From the Middle Volgian *Virgatitus virgatus* Chron the unifying element of the Middle Russian Sea and Arctic basins is *Undorosaurus gorodischensis* (*Zverkov & Efimov, 2019*), whereas *Arthropterygius* are currently unknown in the Middle Russian Sea from this time interval (Fig. 24B), but they still existed at high latitudes during the late Volgian and Ryazanian (latest Tithonian and Berriassian), thus being among the few ichthyosaur taxa that are recognized in the Berriassian.

## Significance of the new finds and further perspectives in the study of ophthalmosaurids

The Beriassian fossil record of marine tetrapods is scarce and patterns of faunal turnover during the Jurassic–Cretaceous transitional interval are non-uniform (*Benson et al., 2010*, *2013*; *Fischer et al., 2012*; *Benson & Druckenmiller, 2014*; *Tennant et al., 2017*; *Zverkov et al., 2018*). It has already been suggested that ichthyosaurs survived the Jurassic–Cretaceous transition relatively unscathed (*Fischer et al., 2012*, *2013*). However, Berriassian ichthyosaur record is still poor (*Fernández & Aguirre-Urreta, 2005*; *Fernández, 2007a*; *Ensom et al., 2009*; *Fischer et al., 2012*; *Green & Lomax, 2014*; *Delsett et al., 2017*). As was discussed above, it is better to consider "*Keilhauia nui*" from the Berriassian of Svalbard as a *nomen dubium*. In this regard, only one Berriassian ichthyosaur, *Caypullisaurus bonapartei* from the Neuquen Basin of Argentina, is hitherto to be recognized at the species level (*Fernández, 2007a*), demonstrating that this Tithonian species sucsessfully crossed the Jurassic–Cretaceous boundary. Discovery of *A. chrisorum* in the Berriassian of Franz Joseph Land provides the second ophthalmosaurid species that unambiguously crossed the Jurassic–Cretaceous boundary, further argument that this transition had minimal (if some) effect on ichthyosaurs.

A discrete character of the fossil record of ophthalmosaurids (*Cleary et al., 2015*) has led to certain problems in the study of this group. The only more or less thoroughly investigated ophthalmosaurids to date are Callovian *Ophthalmosaurus icenicus* (*Andrews, 1910*; *Appleby, 1956*; *Kirton, 1983*; *Moon & Kirton, 2016*) and Albian *Platypterygius australus* (*Wade, 1984*, *1990*; *Kear, 2005*; *Zammit, Norris & Kear, 2010*; *Kear & Zammit, 2014*). Other ophthalmosaurids are incomparably poorly known either due to a small sample size or because of fragmented and/or poor preservation. In such conditions, it is hardly possible to develop a strong phylogenetic hypothesis for ophthalmosaurids. The continuing replenishment of the ophthalmosaurid taxon list by new poorly known and difficult to compare (but having withal a number of autapomorphies) taxa do not make this task easier. The fair attempt to consider all the known ophthalmosaurid taxa and all the proposed phylogenetic characters results in the extremely poorly resolved Ophthalmosauridae (*Moon, 2019*).

Recently *Massare & Lomax (2018)* demonstrated the effect of large sample sizes on the identification of taxonomically distinct morphological characters in *Ichthyosaurus*. This is what is actually needed for ophthalmosaurids. In this regard, Late Jurassic to Early Cretaceous formations of Arctic, considering the abundance and exceptional preservation of marine reptiles (*Delsett et al., 2016*; Nikolay G. Zverkov, 2015–2016, personal observation), have great perspectives for collection of a large sample size, comparable to those of the Lias Group and Posidonia Shale lagerstätten of Western Europe.

## INSTITUTIONAL ABBREVIATIONS

**BRSMG**    Bristol City Museum and Art Gallery, UK
**CAMSM**    Sedgwick Museum of Earth Sciences, Cambridge, UK
**CCMGE**    Chernyshev's Central Museum of Geological Exploration, Saint Petersburg, Russia
**CMN**    Canadian Museum of Nature, Ottawa, Canada
**GIN**    Geological Institute of the Russian Academy of Sciences, Moscow, Russia
**KSU**    A.A. Shtukenberg Museum of Geology and Mineralogy of Kazan State University, Kazan, Russia
**MOZ**    Museo Prof. J. Olsacher, Dirección Provincial de Minería, Zapala, Argentina
**NHMUK**    Natural History Museum, London, UK
**OUMNH**    Oxford University Museum of Natural History, UK
**PMO**    Natural History Museum, University of Oslo (Palaeontological collection), Oslo, Norway
**SGM**    V.I. Vernadsky State Geological Museum of the Russian Academy of Sciences, Moscow, Russia
**SVB**    Svalbard Museum, Longyearbyen, Norway
**UPM**    Undory Palaeontological museum, Undory, Ulyanovsk Region, Russia
**VSEGEI**    A.P. Karpinsky Russian Geological Research Institute, St. Petersburg, Russia
**YKM**    I.A. Goncharov Ulyanovsk Regional Museum, Ulyanovsk, Russia.

## ACKNOWLEDGEMENTS

We thank organizers and participants of the expedition in Franz Joseph Land: N.N. Sobolev, E.A. Korago, E.O. Petrov, S.V. Yudin, P.V. Rekant, A.V. Shmanyak, P.O. Sobolev (all from VSEGEI), N. Yu. Matushkin and N.E. Mikhaltsov (A.A. Trofimuk Institute of Petroleum Geology and Geophysics SB RAS), D.E. Cherepanov (Rosneft Oil Company), very special thanks to V.B. Ershova (Saint Petersburg State University) and A.V. Prokopiev (Diamond and Precious Metal Geology Institute SB RAS), and V.A. Nikishin (Rosneft Oil Company). Many thanks to M.A. Rogov and V.A. Zakharov (GIN) who managed to second NGZ to Franz-Joseph Land and provided valuable consultations on the stratigraphy and palaeobiogeography of the Boreal Upper Jurassic and Lower Cretaceous. Thanks a lot to T. Poulton for consultations on stratigraphy of Arctic Canada and discussion on the stratigraphic position of CMN 40608. Erin Maxwell (Staatliches Museum für Naturkunde Stuttgart) is thanked for discussion on *Arthropterygius* and for providing valuable photographs of CMN 40608 and some other ophthalmosaurids. Jordan Mallon (CMN) is thanked for providing additional photographs of CMN 40608. J.H. Hurum, M.-L. Knudsen Funke, B. Funke, V.S. Engelschiøn, and L.L. Delsett are thanked for hospitality and valuable assistance during NGZ work with PMO collections 27–30, September 2017 and 7–8, November 2018. We thank I.A. Starodubtseva (SGM), V.V. Silantiev and M.N. Urazaeva (KSU), I.M. Stenshin (UPM), O.V. Borodina, D.A. Korepova and S.A. Stryukov (YKM), M. Riley (CAMSM), H. Ketchum (OUMNH), D. Hutchinson (BRSMG), and S. Chapman (NHMUK), for the opportunity to study materials under their care and kind assistance during NGZ visits. Thanks to the technical support of the Artec 3D company, our research is provided with high-quality 3D models. We thank the Willi Hennig Society for their sponsorship making TNT available for all researchers free of cost. The thorough and thoughtful reviews of Aubrey Roberts (NHMUK), Valentin Fischer (Université de Liège) and Davide Foffa (The University of Edinburgh) were beneficial for this work resulting in numerous modifications, but any errors are our own.

### Funding

The expedition in Franz Joseph Land was sponsored by Rosneft Oil Company. During the work on this contribution, Nikolay G. Zverkov was partially supported by the State program no. 0135-2018-0035, programs of RAS no. 19 0135-2018-0042 and no. 17 0135-2018-0050; and primarily by the Russian Foundation for Basic Research, project no. 18-35-00221. The work of Natalya E. Prilepskaya was supported by the Russian Foundation for Basic Research, project no. 18-04-01301-a. There was no additional external funding received for this study. The funders had no role in study design, data collection and analysis, decision to publish, or preparation of the manuscript.

## Grant Disclosures

The following grant information was disclosed by the authors:

Rosneft Oil Company.

State program: 0135-2018-0035.

Programs of RAS: 19 0135-2018-0042 and 17 0135-2018-0050.

Russian Foundation for Basic Research: 18-35-00221.

Russian Foundation for Basic Research: 18-04-01301-a.

## Competing Interests

The authors declare that they have no competing interests.

## Author Contributions

- Nikolay G. Zverkov conceived and designed the experiments, performed the experiments, analyzed the data, contributed reagents/materials/analysis tools, prepared figures and/or tables, authored or reviewed drafts of the paper, approved the final draft.
- Natalya E. Prilepskaya analyzed the data, contributed reagents/materials/analysis tools, approved the final draft, 3D scanned the specimens.

## Data Availability

Zverkov, Nikolay; Prilepskaya, Natalya (2019): A prevalence of Arthropterygius in the Late Jurassic–earliest Cretaceous of the Boreal Realm: character-taxon matrix. figshare. Dataset. DOI 10.6084/m9.figshare.7406522.v1.

Zverkov, Nikolay; Prilepskaya, Natalya (2019): 3d scan of Arthropterygius chrisorum CCMGE 3–16/13328. figshare. Fileset. DOI 10.6084/m9.figshare.7406606.v1.

Zverkov, Nikolay; Prilepskaya, Natalya (2019): 3d scans of Arthropterygius chrisorum CCMGE 17–44/13328. figshare. Fileset. DOI 10.6084/m9.figshare.7406696.v1.

## Supplemental Information

Supplemental information for this article can be found online at http://dx.doi.org/10.7717/peerj.6799#supplemental-information.

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
