# Peer review of "A prevalence of Arthropterygius (Ichthyosauria: Ophthalmosauridae) in the Late Jurassic—earliest Cretaceous of the Boreal Realm"

_PeerJ, doi:10.7717/peerj.6799_

## Round 0.1 · original submission · Major Revisions

Dear authors,

I apologise for the delay in getting this back to you. I had some trouble getting reviewers (but luckily two accepted at similar times).

All three reviewers made the same recommendation. I strongly suggest reading all three carefully, but the changes suggested I feel are all fair.

The issue of referring to ‘in press’ papers. Can the authors ensure they are available for the reviewers to look over at the next stage? That would greatly help them.

I look forward to receiving your revised manuscript.

·

Basic reporting

Dear editor, dear authors,

I consider this manuscript as a real tour de force, bringing an impressive amount of novel data from a series of spatiotemporal settings, to better understand the early evolution of ophthalmosaurids ichthyosaurs. I really like the thoroughness of the morphological analyses, the expanded phylogenetic dataset and the idea to use PCA. I am in certain this will make a great paper.

I am thus totally favorable for publication of this work in PeerJ, but after some revision. Indeed, I feel that the present version of the manuscript is problematic in its (i) tone, (ii) structuration/argumentation (Basic Reporting section) and (iii) taxonomic decisions (Validity of the findings section). I detail these three points in each section. I have indicated in General Comments, a series of line-by-line minor comments. I am confident that the authors can work these issues out and so I suggest that the paper requires a *moderate revision*. This is not an option in PeerJ, so because "minor" requires no additional revision, I had to select "major".

(i) Tone
I think that the text is generally unnecessarily critical of some colleagues, using words and verbs suggesting that most of them were quite profoundly mistaken: “misinterpretation”, “misleading”, etc. (e.g. L66 and in Materials, L244, L324-336). By reading the text, it appears that nearly all authors who have been working on similar taxa were mistaken: e.g. Delsett (many instances), Roberts (many instances), Maxwell (L575), Druckenmiller (L683), Kasansky (L1048). Only one scientist, the Russian geologist and palaeontologist A.P. Pavlov is regarded as “outstanding” (L100). While this might be the case and that our knowledge is filled with a series of errors and mistakes, remember that in most cases, you are providing re-interpretations (which, indeed are probably very good!) rather than the absolute truth. I think this helps putting things in perspective. I am totally certain that this is not intentional.

Similarly, the burden of proof is on your side; you cannot mention current knowledge as "alleged" prior to detailing your results. I would suggest to carefully re-read the text and rephrase some sentences, first sticking to what is currently known (or believed) and then acknowledging that you have a different interpretation than previous authors.


(ii) Structuration of the text and argumentation
There are two things here, the first being that in the present version of the paper, it is stated upfront that profound changes in the taxonomy are made, but the reasons are disseminated in the entire (fairly long) text. Given that important taxonomic changes proposed, I think it would be better to first describe the new remains and re-interpretations, then discuss the taxonomic possibilities, and then place a Systematic Palaeontology section, towards the end of the paper.

The second issue is that the text appears to frequently mix facts, interpretations, and opinions (I have indicated several instances in the line-by-line comments below). This has the (surely undesired) effect of confusing the reader. Similarly, in some instances, results are presented without any argumentation or sensible/tangible data (e.g. L20-21, L150 and after and to a lesser extent the whole biostratigraphic data, L166-167, L237-238, L324-336, L418-421, L451-453), making them opinions rather than testable results and trying to get certitude out of incertitude. It is difficult to propose a clear-cut solution here, but rather a thorough re-read of all the paper’ sections.

Experimental design

I do not see obvious flaws here as methods have been carried on with great care, but I have a couple of questions on PCA, which I discuss on the next section?

Validity of the findings

(iii) Taxonomy
There seem to be a distinction in the taxonomic relevance of morphological features throughout the manuscript. Many features listed by previous authors as differing between Arthropterygius and other genera are said to be misinterpretations (nearly all authors who have been working on similar taxa were seemingly mistaken, as mentioned above in these comments). Other differing features are dismissed right away as variability (e.g. L504).

But at the same time, all previously established species are retained, sometimes based on rather minute differences (e.g. A. volgensis) and no variation is invoked in these cases. This difference in the taxonomic importance (generic VS specific VS population) of morphological features appears at times puzzling from the reader’s perspective and would benefit from a more rigorous treatment throughout the paper.

Indeed, no clear arguments is provided for or against another and simpler possibility: that Arthropterygius, Janusaurus and Palvennia are valid genera belonging to a well-defined clade of ophthalmosaurids. The results of the morphospace analyses suggest the same, with Arthropterygius spp. having a wider spread (and thus disparity) than entire ophthalmosaurids subfamilies, with a much larger time span. Don’t get me wrong, the simplicity of this solution does not mean it is the correct one, but at least it should thoroughly and fairly evaluate, along with more taxonomically complex solutions.

The inference (L1253) that a series of ichthyosaurs had assymetric limbs appears quite unlikely and most probably an effect of diagenesis. From my experience, all the humeri I have examined were very similar left-right (not totally similar not certainly not massively dissimilar); this result actually call into question the influence of diagenesis on the results, which is not discussed in the text. Indeed, the authors state that humeri of Arthropterygius can be recognized quite easily, but at the same time, mention that there are large in vivo intra-individual differences (added to the result mentioned above that the disparity of the humeri of Arthropterygius exceeds those of Ophthalmosaurines and Platypterygiines).

Additional comments

Line-by-line minor comments

L14 and L53: “heretofore” might be a bit awkward here, how about “hitherto”?

L15-16: you should briefly argue why (i.e. phylogenetic position with respect to ophthalmosaurid subfamillies)

L49: this might be a strecht to say Aegirosaurus is *well* known... Same for Undorosaurus. Why do you omit Ophthalmosaurus? Do you specifically refer to the taxa abundant in the Kimmeridgian-Tithonian instead of the whole Late Jurassic?

L54: Awkward sentence construction; could you please rephrase?

L65: poorly known “in” instead of "for"

L73: “allow us *to* substantially”

L100: Consider removing 'outstanding', it is an opinion

L108-110: I would suggest to remove this sentence, as the main message is explained in the next sentences.

L128: Geological setting. I would suggest to produce a couple of figures with the specimens mentioned that have a stratigraphic importance. Indeed, this is scientific data that should be analyzed, commented and criticized by others just like in other data in the paper. This might be especially important in future, notably regarding the hot debates surrounding the Jurassic-Cretaceous boundary.

L150= "almost" and "unambiguous" do not go well together most of the time.... I would suggest to rephrase and to provide arguments for this (e.g. only X m below the start of such biozone etc.)

L195: synapomorphies are usually applied to suprageneric clades; I would suggest to use "autapomorphies" to avoid confusion.

L224: I suggest to make a table with the different features and their presence/absence/NA in the species merged into Arthropterygius. That would be a good companion to the text.

L237-238 and throughout: such sentence, somehow forcing the reader into an opinion, should be avoided; I suggest to delete it.

L244: "misleading" is not appropriate here; I would suggest "questionable"

L246: consider replacing "say something"

L264: lacks *an* obturator foramen

L254-266: the sentence is unclear; could you rephrase?

L324-336: again please watch out for tone and the difference between fact and opinion

L338: Description: would it be possible to add more comparisons with other ophthalmosaurids?

L418-421: Again, an opinion is voiced without any argumentation. This needs to be thoroughly checked in the entire text.

L451-453: I suggest not to estimate jaw length when the entire dentary is missing.

L639: "h" missing in Arthropterygius

L788-790 The meaning of this sentence is unclear; could you rephrase?

L807: too many "th" in Arthropterygius

L1136 a series of "unambiguous" synapomorphies are indicated, some of which being labelled as "non-unique" (thus ambiguous). Could you clarify?

L1247-1249: this is well known (and did not require a PCA to be noted by previous authors ;) )

L1364: I do not understand the meaning of "aerials" in this sentence.

L1398 "lagerstätten"


All the best and great job!

Valentin

·

Basic reporting

General comments

Throughout the manuscript, it is confusing which specimens are referred to which species. Please add a table to make this very clear for readers, that might not be familiar with the previous specimens and publications. When describing a species with multiply referred specimens, please include which specimen you are referring to in the description (this needs to be really clear). This is to avoid confusion for readers in which specimen preserves what. Please fix this throughout the description
In general, the level of English in the paper is high, however there are a few things that should be changed. I have suggested a few wording changes in the document, one of these that should be searched and replaced are the words “thanks to”. This is particularly scientific language and should be changed throughout the document. Examples of words they could be changed to: “due to”, “as consequence to” ect. Other sentences/words that should be rephrased are highlighted.


Figures:
The figures are of a good standard and quality, figure 1 in particular is very detailed and visually appealing. However, I have not seen any evidence of figure captions, where are they? Some of the figures of cranial elements should be edited. The lines bordering the different bone elements follow, in some cases, taphonomic cracking/breakage and do not represent a suture. These should be shown either by a dotted line or pointed out on the figure. On the PCA figure, one of the names seems to be incomplete (just PMO). This should be changed.

Experimental design

I insist on changes made to the emended diagnosis as it is currently confusing and should include which features which are present in which taxa/specimens and which features it shares or differs with other taxa. This will make the diagnosis more useful to other researchers and will avoid misunderstandings on which features are present in which species.

The palaeobiogeography section can be shortened, as much of it just repeats previous work, which is well summarised in Zverkov et al. 2015. Please just refer to this work and add in what the new data tells us about the distribution of Arthropterygius.

Other comments on methods and analyses- see annotated pdf

Validity of the findings

From the text I cannot clearly understand why you have referred PMO 222.669 to A. chrisorum and not A. hoybergeti. This needs to be clarified. There is no evidence of a large pineal foramen in any of the other Arthropterygius specimens referred to here, which was one of the main reasons to referring PMO 222.669 to Palvennia.

The new combination referral of the juvenile KSU 982/P-213 to Arthropterygius volgensis lacks sufficient justification. Basing a species purely on the morphology of the humerus is not acceptable, as mentioned in your own PCA analysis, the is even variation in the morphology of some of the traits in individuals. This specimen should be referred to Arthropterygius indet. This problem is also the case for the referral of UPM 2442 and YKM 63548 to A. hoybergeti.

The section on “Ontogenetic changes and variation in Arthropterygius chrisorum”, requires significant revision. Using a few incomplete specimens of different species (which could vary in size), is not sufficient to estimate ontogenetic change. This either needs to be confirmed using methods such as histology (or μCT scanning), or played down significantly. There are multiple papers by the Sander lab looking into the histology of marine reptiles. I can provide these if the authors need them.

Additional comments

I very much enjoyed reading the paper and I think it fits the angle for the journal sufficiently, I recommend acceptance with major revisions. There are numerous changes that need to be made to the manuscript before publication. These are available in my edited word document and some of them are highlighted in my comments. hope my review of this manuscript will be useful to the authors and editor and I look forward to seeing the paper published after these changes have been made.

·

Basic reporting

No comment

Experimental design

No comment

Validity of the findings

No comment

Additional comments

I commend the authors for their extensive and detailed description, which is the fruit of years of fieldwork, personal observations and detailed analyses of the collected data. The result is an undeniably convincing and well exposed hypothesis that attempts to revise the morphology, taxonomy, systematics and palaeogeography of an important ichthyosaur genus.

The manuscript can be improved in some respects – which I consider marginal compared to the amount of effort and work already done.

The language should be improved to ensure that an international audience can clearly understand your text. The manuscript contains a number of grammatical mistakes that should be addressed. In some cases, these imprecisions make it unclear the meaning of individual sentences - but it rarely affects the overall meaning of the text.
Amongst these:
- There is a recurrent misuse/omission of articles “A” and “THE” (see attached PDF).
- In several cases sentences are unnecessary long, full of subordinates – often exceeding 3-4 lines. They should be simplified to improve the flow and clarity of the text (see attached PDF).
- Use of odd sentences words (e.g. Lines 108-110; 115-116). While the meaning of the sentence is clear by context, it is not immediate. These sentences/words should be replaced simplified. These and other examples can be found in the attached PDFs.
- The whole description should be considerably simplified as it is currently difficult to follow.
- There is no consistency in the verb tenses used through the text.

My biggest concern is that TOO much details are referred to other papers (most worryingly to one that is “in press” and was not provided with the manuscript – thus that information is unverifiable).
Description
Generally, the authors report well supported arguments – in places the authors should tone down the text (e.g. “In our opinion, this is…” rather than “This is”.
In places the authors should instead providing better evidence for their claims, especially in the initial part of the manuscript where they argue against a number of researchers’s observation (e.g. paragraphs before and beginning with line 324).
Unfortunately some of the authors’ arguments are not sufficiently justified and they offer an alternative interpretation without supporting it (a simple reference or observations on other specimens would be enough).
A lot of the statements in the descriptions are referred to “Zverkov & Efimov, in press”. As I do not have access to this manuscript, I cannot assess any of these statements/observations.
In the description section there is very limited comparisons with other ophthalmosaurids (and also put in a phylogenetic context as it is very well done in some paragraphs: e.g. prefrontal (beginning at line 373) and basioccipital (beginning at line 422) and basisphenoid (beginning at line 422). This paragraph should be used as an example for all the other descriptive paragraphs). This issue should be addressed especially for those elements that had not been previously described or for new species.
Finally the authors should more clearly underline the similarities between species they refer to the same genus, especially where the newly available materials show them.
Multivariate analysis of ophthalmosaurid humeral morphology
This whole section is accessory to the manuscript – and the authors may want to consider moving this in the supplementary section. It is not vital – and a really minor point. I understand the desire of the authors to include it in the manuscript.
I would agree if the authors decided to keep it in the main text, but if they do, the relevance of the analysis for the manuscript should be more clearly stated.
Biogeography.
Overall, very interesting section. It would greatly benefit from inclusion of some missing references and comparisons with other groups (see PDF). The importance of the Hispanic corridor is likely dismissed too quickly and the dispersal pattern of other marine reptile groups should at very least be mentioned.
Supplementary information
There is an array of several different fonts and font size in the supplementary material.
Please review it and make it consistent through the text and with the main manuscript.
Figures
Beautiful and very clear figures. Well referenced within the text.
Additional clarifications on the stratigraphy should be probably added.
A comparative plate of overlapping Arthropterygius cranial and postcranial elements should be added (similar to Fig. 17 but with additional diagnostic features and overlapping elements).
Additional details and comments are included in the PDF.

---

## Round 0.2 · Minor Revisions

Dear authors,

I have accepted the decision of the two reviewers (I didnt hear back from from one other reviewer) of 'minor revisions'.

I look forward to receiving your revised manuscript.

·

Basic reporting

The structuration of the paper is still problematic, see general section below.

Experimental design

All good here, but all methods should be put a Methods section (and all results should be the Results section).

Validity of the findings

Nothing major here; see general section .

Additional comments

Dear editor, dear authors,

First of all, let me apologize for the 2 days delay in sending my review!

The authors have made substantial modifications to the paper and have answered many of my comments. I still have some concerns about the organization of the paper. I suggest acceptation after a minor revision.

Main concerns:

L72-79. The ventral skew. This is interesting and indeed quite easy to recognize. However, the universality of this feature as an indicator of Arthropterygius is puzzling, as it is also found in a series of specimens that have been linked to the poorly known taxon Macropterygius trigonus (e.g. Lydekker 1889), which do not have a facet for an AAE element at all (or extremely reduced; e.g. Lydekker 1889; Sauvage 1912), which strongly differs from the condition seen in Arthropterygius.

L195. This addition was made to reply to a series of points raised by reviewers bit I think it appears a bit dodgy and mixed-up. What I suggested in the previous round was to describe the new remains, describe the re-interpretations of existing specimens (using their specimen numbers and mentioning holotype of, say, Palvennia hoybergeti). And only after that, discuss the similar and finally propose a new taxonomic scheme. Here, we actually get the exact opposite, starting the results by a very technical discussion of detailed features, with quotes etc. that comes before the description of the new data. I might be because the authors prefer a different structure, but the current result is still quite complex to understand. So, I propose here two different plans to make the paper clearer:

Option 1:

A. Introduction
B. Material
C. Methods (including phylogenetics and PCA)
D. Results
D.1. Comparative description of the novel remains
D.2. Comparative redescription of Palvennia
D.3. Comparative redescription of Janusaurus
D.4. Comparative redescription of Keilhauia
D.5. Phylogenetic analyses
D.6. PCA Analyses
E. Discussion
E.1. A wider concept of Arthropterygius (should contain all your arguments, including most of what is written in the “Anticipatory” section)
E.2. Systematic palaeontology
E.3. PCA, ontogenetic changes
E.4. Palaeobiogeograhy
F. Conclusions


Option 2 (closer to what you have now):

A. Introduction
B. Material
C. Methods (including phylogenetics and PCA)
D. Results
D.1. Preliminary paragraph briefly indicating in a couple of sentences which are synonymised etc.
D.2. Systematic Palaeontology
D.3. Systematic description of
D.3.1. Arthropterygius chrisorum
D.3.2. Arthropterygius hoybergeti
D.3.3. Arthropterygius lundi
Etc.
D.4. Phylogeny
D.5. PCA
E. Discussion
E.1. A wider concept of Arthropterygius (should contain all your arguments, including most of what is written in the “Anticipatory” section)
E.2. PCA, ontogenetic changes
E.4. Palaeobiogeograhy
F. Conclusions


I would also like to comment the response to my previous comment:

Authors replied: It is more about that, in our opinion, no robust arguments for Janusaurus and Palvennia representing distinct genera, from each other, and from Arthropterygius, was provided in previous works. That the burden of proving this is on our side now is a pity. The retention of all the species as valid based on “rather minute differences” is sort a tribute to previous research.

I think there is no need to blame previous authors; either you have new data and novel interpretations suggesting this and that, either you don’t and so the knowledge remains similar to what has been published in the past. What you can do is to say that differences warrant specific but not generic recognition. But again, it comes back to my original comment asking to provide more arguments on why some features are dismissed right away as variable while others are retained without question. I just want you to pay attention to circular reasoning.


L225. I do not understand the argument on the basioccipital peg. It is said to be a vestige of the peg, but that the feature should not be scored as homologous to a peg. That is a bit weird. Either it is homologous (even if different in shape) and scored accordingly (I.e. peg present), either it is a structure with a distinct evolutionary origin and thus scored as absent.

PCA: could you explain why the figure has been modified? You seem to have increased sample size for Ophthalmosaurus as well. Also, you represent both axes with the same unit length? It is a bit strange to have the data points spread along the second axis, while obviously the first axis represents a larger chunk of the total variance.


Minor points:
L32-33. Consider replacing “most well-“ by “best-“
L62-64. Please considered deleting that new sentence, which appears as a judgment of value. The sentence starting by “Recently” is fine.
L67. […] in the present contribution […]
L71; L210 and throughout. Latin names cannot be used as adjectives ideally (exceptions are made for biostratigraphic zones).


REFS:
Lydekker R. 1889. Catalogue of the fossil Reptilia and Amphibia in British Museum (Natural History). Part II. containing the orders Ichthyopterygia and Sauropterygia. London, Printed by Orders of the Trustees of the British Museum, London, 307pp
Sauvage H-E. 1912. Les Ichthyosauriens des Formations Jurassiques du Boulonnais. Bulletin de la Société Académique du Boulonnais, 9 424-443.

All the best,

Valentin Fischer

·

Basic reporting

Dear Editor and Authors,

I have reviewed this manuscript in the first round, and I am pleased to notice that the authors have addressed a substantial number of the requested changes.
Overall, I believe that the text flows better now, and the authors have effectively replaced numerous issues with more informative/precise/correct sentences. The addition of the new table (the specimens table) helps to keep track of the numerous specimens that are mentioned in the manuscript.
Notably many of my concerns and clarifications I asked have been addressed after the publication of Zverkov and Efimov 2019 manuscript that was in press during the first round of reviews. The publication of that paper has given further context to some of the comparisons that were not particularly clear before.
From a stylistic point of view, I particularly appreciated the rewriting of some parts that in the previous version of the manuscript read overly critical to previous authors; the expansion on the Geological Settings section and reorganization of the manuscript “Anticipatory clarification of taxonomic decisions” in its current form.
The scientific side of the manuscript was also strengthened by the reviews, particularly the modifications to the Description and Discussion are more focused and precise.
There are however some issues that still need to be addressed (see below).

Experimental design

no comment

Validity of the findings

no comment

Additional comments

Although the general tone of the manuscript has considerably improved, there still are a handful of sentences that should be revised or better supported. One example is line 231: “Furthermore, the value of this character (i.e. presence/absence of the anterior notochordal) is somewhat uncertain.” The other reviewers and I pointed out that this kind of statement should be better supported by hard evidence or overall avoided. The authors are free to acknowledge that some characters are more useful than others, but they should clear provide evidence when their characters’ interpretation goes against previous workers interpretations. This issue has been largely addressed from the previous version of the manuscript, but still persists in some parts. Please revise the text or tone down these sentences (e.g. “based on XXXXX we believe that the value of this character is uncertain”).

I do not fully understand why the “Anticipatory clarification of taxonomic decisions” section is placed BEFORE the description of the new material. As Reviewer 1 suggested, I would think that a better place for a summarising section like this would be the discussion, or anyway a place after the descriptions. Indeed, this is the central part of the discussion and main take-away message of the manuscript. As thus, a clarification of taxonomic decision should be the main part of the Discussion pulling together the descriptions but also the results of the PCA (especially in absence of a Conclusion section). I advise the author to revise this section and re-formulate their case for the taxonomic decisions using both morphological descriptions and the PCA results.

Still, concerning this section, I am also sorry to insist on the table/figure point. In different forms all 3 reviewers asked for clarifications and recommended additions to the text: I suggested a comparative figure – which I would argue is still needed (even if time-consuming) at least as supplementary figure; R1 argued for a table “including different features and their presence/absence/NA in the species merged into Arthropterygius”; and R2 asked for clarifications too. These would be a great aid to the reader. I think that in addition to the re-writing at very least a table [and/or comparative plate (if not both]) should be added. Do not get me wrong, I do appreciate the efforts of the authors to craft the new section. This helps a lot, but I feel that a table alongside it would summarise these reasonings in a very immediate way, and it is an easy task for the authors to produce it.
Additional issues:
- As R1 I think that comparisons with other ophthalmosaurids would be beneficial to the manuscript, and the additional references to key taxa to compare systematically important features would not significantly increase the manuscript length. I am aware and recognise that there are extensive comparisons with these taxa in previous papers. However, AT VERY LEAST the authors should mention other ophthalmosaurids when explaining: 1) previously misunderstood features; 2) diagnostic and taxonomically important features (genus defining and group defining characters). The reason for this is that the authors' new taxonomy hypothesis may have changed the distribution of some key characters in the family tree. Thus, explicitly discussing these characters in a broader context and would be an aid for future studies (as much as the authors have benefited from previous extensive comparisons in previous papers).
- I do not see a stratigraphic log of the new localities even if the authors said they added one did the authors uploaded a wrong version of figure 1?
- Genera and species names in italicised quotations should not be italicised too (see: “Anticipatory clarification of taxonomic decisions” section) [lines 212 to 218]
- I do appreciate the work done to revise the diagnosis and I think the authors did a great job to clarify in which specimen the characters are visible. However, I would also rather know which of these features are shared with which other taxa.

I would be favorable to be directly contacted by email (d.foffa@nms.ac.uk) in case the authors wished further clarifications.
Many thanks and congratulations for the work done so far.
Kind regards,
Dr. Davide Foffa

---

## Round 0.3 · accepted · Accept

Dear authors,

Many thanks for your revised manuscript. After reading it, I have accepted it for publication in PeerJ.

Once again, thank you for submitting your manuscript to PeerJ and I hope you will use us again as your publication venue.

If we need to clarify any details required to move the manuscript forward, then our production staff will get in touch with you. Otherwise, a proof will be forthcoming shortly for your review.

Congratulations and thank you for your submission.

#